# Depletion of slow-cycling PDGFRα⁺ADAM12⁺ mesenchymal cells promotes antitumor immunity by restricting macrophage efferocytosis

Selene E. Di Carlo[1], Jerome Raffenne[2], Hugo Varet [3,4], Anais Ode[1], David Cabrerizo Granados[1,6], Merle Stein[5], Rachel Legendre [3,4], Jan Tuckermann [5], Corinne Bousquet[2] & Lucie Peduto [1]✉

The capacity to survive and thrive in conditions of limited resources and high inflammation is a major driver of tumor malignancy. Here we identified slow-cycling ADAM12⁺PDGFRα⁺ mesenchymal stromal cells (MSCs) induced at the tumor margins in mouse models of melanoma, pancreatic cancer and prostate cancer. Using inducible lineage tracing and transcriptomics, we demonstrated that metabolically altered ADAM12⁺ MSCs induced pathological angiogenesis and immunosuppression by promoting macrophage efferocytosis and polarization through overexpression of genes such as *Gas6*, *Lgals3* and *Csf1*. Genetic depletion of ADAM12⁺ cells restored a functional tumor vasculature, reduced hypoxia and acidosis and normalized CAFs, inducing infiltration of effector T cells and growth inhibition of melanomas and pancreatic neuroendocrine cancer, in a process dependent on TGF-β. In human cancer, ADAM12 stratifies patients with high levels of hypoxia and innate resistance mechanisms, as well as factors associated with a poor prognosis and drug resistance such as AXL. Altogether, our data show that depletion of tumor-induced slow-cycling PDGFRα⁺ MSCs through ADAM12 restores antitumor immunity.

Inflammation and hypoxia are hallmarks of invasive tumors and major drivers of tumor progression[1,2]. The tumor microenvironment (TME) is characterized by poorly functional angiogenic vasculature, inflammatory infiltrates and high levels of tissue remodeling. Impaired blood flow further restricts delivery of oxygen, antibodies and drug delivery. The resulting hypoxia promotes immunosuppression by upregulation of transforming growth factor beta (TGF-β), vascular endothelial growth factor (VEGF), tumor-associated macrophages (TAMs), nutrient deprivation,

a switch in tumor metabolism and acidification of the microenvironment[3,4]. Such an immunosuppressive microenvironment promotes adaptive and invasive mechanisms including cancer cell dormancy, a state of low proliferation that is common in stem cells and stem-like cells[5]. This altered metabolic state allows survival in conditions of limited resources, protects from antimitotic drugs and promotes metastasis[6].

Growth of solid tumors induces a stromal reaction due to local damage, particularly at the tumor margin, a transition zone that is

[1]Stroma, Inflammation & Tissue Repair Unit, Institut Pasteur, Université Paris Cité, INSERM U1224, Paris, France. [2]INSERM U1037, Cancer Research Center of Toulouse (CRCT), Toulouse, France. [3]Transcriptome and Epigenome Platform-Biomics Pole, Institut Pasteur, Université Paris Cité, Paris, France. [4]Bioinformatics and Biostatistics Hub, Institut Pasteur, Université Paris Cité, Paris, France. [5]Institute of Comparative Molecular Endocrinology, University of Ulm, Ulm, Germany. [6]Present address: Laboratory for Disease Mechanisms in Cancer, KU Leuven, Leuven, Belgium. ✉e-mail: lucie.peduto@pasteur.fr

**Fig. 1 | Genetic depletion of ADAM12+ MSCs restores tumor immunity.**
**a**, Immunofluorescence staining of PDPN, αSMA, CD3 and collagen in MO5 melanomas. The inset shows CD3+ T cells (green) on PDPN+ stromal cells (red). **b**, Immunofluorescence staining of PDPN, CD31 and ADAM12 (GFP) in MO5 melanomas in ADAM12-GFP mice. The inset shows GFP+ cells (green) close to CD31+ blood vessels (blue; marked with an arrowhead). Scale bars, 100 μm. One representative image from four (**a**) or six (**b**) independent experiments is shown. Right, FACS plot and percentage of CD45−CD31−cells and GFP+ cells in MO5 melanomas 8 d after tumor inoculation. **c**, Percentage of GFP+ cells, measured by FACS, in PDGFRα+ stroma isolated from normal skin (day 0, n = 2) or MO5 tumors (n = 6 for 8–12 d and n = 4 for 14–17 d), and in other populations (n = 3).

**d**,**e**, Tumor growth curves (average tumor volume) from ADAM12-DTR (DTR) and littermate mice (Ctrl) treated with diphtheria toxin (DT) from days 10 to 18 (DTR, n = 10; Ctrl, n = 12) (**d**), or from days 0 to 10 (DTR, n = 5; Ctrl, n = 7) (**e**) after tumor inoculation. The x axis represents days after tumor inoculation. **f**, Percentage of tumor-infiltrating CD3+ T cells (n = 8 for Ctrl, n = 12 for DTR), CD8+ T cells (n = 14 for Ctrl, n = 17 for DTR), IFN-γ+CD8+ T cells (n = 7 for Ctrl, n = 6 for DTR), NK cells (n = 9 for Ctrl, n = 9 for DTR) and IFN-γ+ NK cells (n = 16 for Ctrl, n = 14 for DTR) in mice treated with DT from day 10, measured by FACS in three independent experiments. Statistics were calculated using ordinary two-way analysis of variance (ANOVA) (**d**,**e**) or two-tailed, unpaired Student's t-test (**f**). Quantitative data are presented as means ± s.d. n.s., not significant.

rich in immune cells, blood vessels and mesenchymal cells. Expansion of mesenchymal cells, also called carcinoma-associated fibroblasts (CAFs), around and within a tumor mass is associated with resistance to therapy and poor clinical outcomes[7,8]. Tumor stromal cells express mesenchymal markers such as podoplanin (PDPN) and platelet-derived growth factor receptor alpha (PDGFR-α), and are highly heterogeneous, as has been shown by single cells RNA sequencing (RNA-seq) studies in several tumor types[9–11]. They express, albeit not specifically, a number of factors and pathways associated with recruitment of immune cells, angiogenesis, myofibroblast activation and extracellular matrix (ECM) remodeling, including fibroblast-activation protein (FAP), CXCL12, Lox, TGF-β and Hedgehog pathways, all of which play a role in tumor progression[12–14]. Broad targeting of CAFs or collagen production has led to mixed results, because stromal cells and the ECM are required for tissue homeostasis and to restrain tumor growth[15–21]. As part of the tumor mass, stromal cells also adapt their metabolism, yet the impact on the tumor microenvironment and antitumor immunity in vivo remains unclear.

A disintegrin and metalloprotease 12 (ADAM12) is a membrane-bound metalloprotease that is expressed during organ morphogenesis and is re-induced in mesenchymal cells during repair and fibrosis and in solid tumors, including pancreas, prostate, breast, colon, bladder and liver cancers and melanoma, both in human disease and mouse models[22–27]. ADAM12 overexpression has been associated with resistance to chemotherapy and poor prognosis[26,28–33]; however, the role of ADAM12+ MSCs in tumorigenesis has not been addressed. Here, we use reporter and deleter genetic models to demonstrate that ADAM12+ cells are a subset of slow-cycling PDGFRα+ mesenchymal perivascular cells, distinct from pericytes, that induce angiogenesis and immunosuppression

by promoting TAMs efferocytosis and polarization. Genetic depletion of ADAM12+ cells normalizes the tumor vasculature and decreases hypoxia and acidity, restoring antitumor immunity and blocking tumor growth in mouse models of melanoma and neuroendocrine pancreatic cancer. We further provide direct genetic evidence that TGF-β receptor 2 (TGFBR2) signaling in ADAM12+ cells is required for their pro-tumorigenic function. We propose that ADAM12+ MSCs are part of an evolutionarily conserved mechanism initiated by the cytostatic TGF-β and modulated by inflammation to promote tissue repair, in cooperation with macrophages. In the context of tumorigenesis, such a response represents a major brake for antitumor immunity.

## Results

### Genetic depletion of ADAM12+ MSCs restores tumor immunity

To visualize ADAM12+ cells during tumorigenesis, we subcutaneously inoculated ADAM12-GFP mice)[27] with B16-OVA melanoma cells (MO5). We observed that ADAM12+ MSCs (expressing GFP) localized specifically at the tumor margin, a transition zone enriched in stromal cells expressing various levels of PDPN and smooth muscle actin alpha 2 (αSMA+), collagenous ECM, T cells, CD206+ TAMs and blood vessels (Fig. 1a,b and Extended Data Fig. 1a,b). ADAM12−PDPN+PDGFRα+ cells were already abundant in non-tumoral skin (Extended Data Fig. 1c). ADAM12 was not detected in normal skin or in CD45+ tumor immune cells, CD31+ endothelial cells or PDPN−PDGFRα− stromal cells, but its expression was induced in 2–8% of PDGFRα+PDPN+ cells adjacent to peritumoral blood vessels. The frequencies and absolute numbers of ADAM12+ cells decreased at later tumor stages (Fig. 1b,c and Extended Data Fig. 1d–f). ADAM12+ MSCs were PDGFRβ+, αSMA− and NG2lo or NG2− (a pericyte marker), and were localized outside the ColIV+ vascular

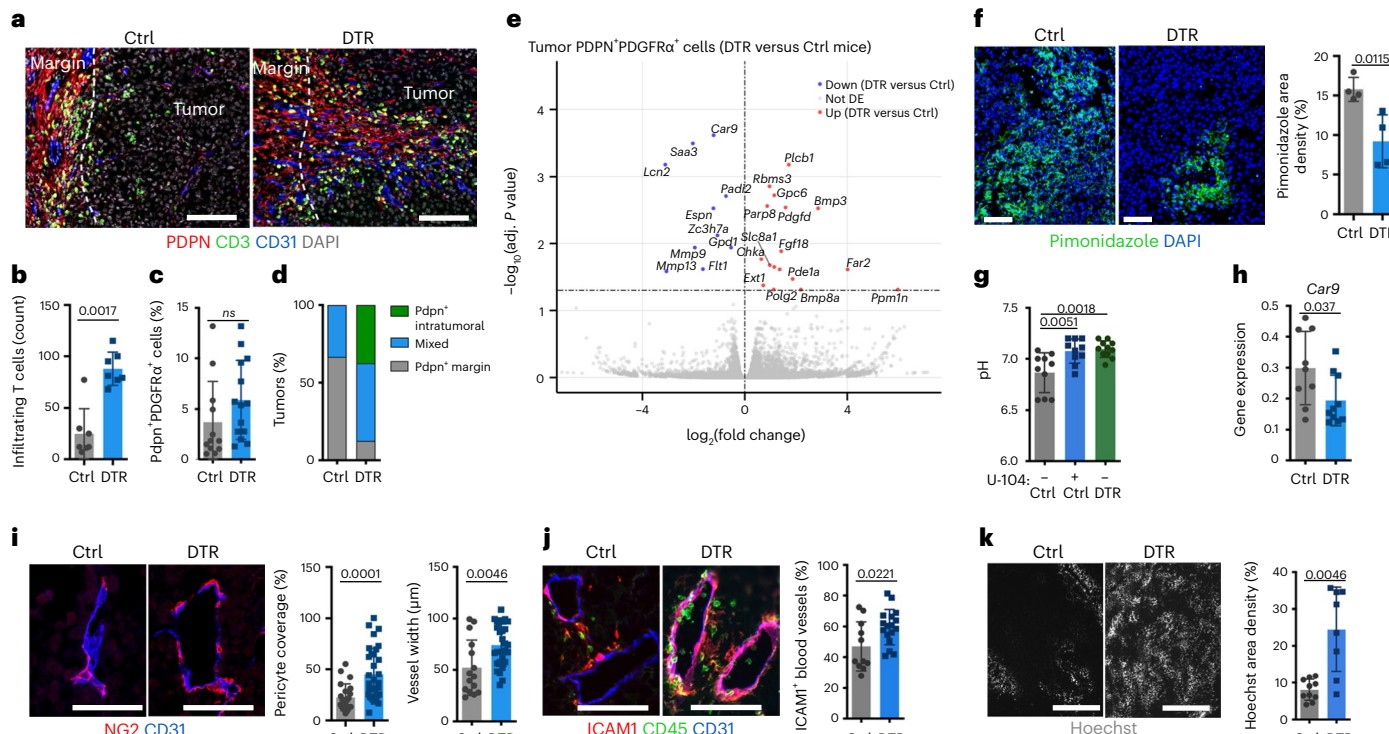

**Fig. 2 | Depletion of ADAM12⁺ MSCs normalizes the stromal/vascular TME.**
**a**, Immunofluorescence staining of PDPN, CD3 and CD31 in MO5 tumors of ADAM12-DTR mice (DTR) and littermates (Ctrl) treated with DT from day 10. One representative image of several independent experiments (*n* = 4) is shown. Scale bars, 100 μm. **b**, Tumor-infiltrating CD3⁺ T cells in mice treated as in **a**; *n* = 7 in 2 independent experiments. **c**, Percentage of PDPN⁺PDGFRα⁺ cells (gated CD45⁻ CD31⁻), measured by FACS, in MO5 tumors of mice treated as in **a**; *n* = 12 (Ctrl)–14 (DTR) in 3 independent experiments. **d**, Percentage of MO5 tumors containing Pdpn⁺ stromal cells, as indicated in mice treated as in **a**. *n* = 6–8 in 4 independent experiments. **e**, Volcano plot showing differentially expressed (DE) genes in PDPN⁺PDGFRα⁺ cells isolated from DTR versus Ctrl tumors. **f**, Tumor hypoxia was assessed by immunofluorescence staining of pimonidazole in tumors treated as in **a** (*n* = 4). Scale bars, 100 μm. **g**, Extracellular pH in tumors growing in mice treated as in **a**, or in Ctrl mice treated with U-104. *n* = 10(Ctrl)–11(DTR) in

3 independent experiments. **h**, Expression of *Car9*, as measured by qRT–PCR, in tumors from mice treated as in **a**; *n* = 9 (Ctrl)–10 (DTR) in 2 independent experiments. **i**, Immunofluorescence staining of NG2⁺ pericytes and CD31⁺ blood vessels in tumor sections from mice treated as in **a**. Left, *n* = 22 (Ctrl)–32 (DTR) and right, n = 15 (Ctrl)–28 (DTR) blood vessels. **j**, Immunofluorescence staining of ICAM1, CD45 and CD31 in tumor sections from mice treated as in **a**; *n* = 11 (Ctrl)–17 (DTR) fields. **k**, Tissue perfusion, as detected by Hoechst 33342 staining, in tumor sections from mice treated as in **a**; *n* = 10 (Ctrl)–8 (DTR). Scale bar, 200 μm. In **i**–**k**, data are representative of several independent experiments (3–5). In **i**–**j**, scale bars, 50 μm. Statistics were calculated using two-tailed, unpaired Student's *t*-test (**f**,**h**,**i**(right),**j**,**k**), ordinary one-way ANOVA (**g**), two-tailed Mann–Whitney test (**b**,**c**,**i**(left)) or two-sided Wald test (DESeq2) (**e**). All quantitative data are presented as means ± s.d.

basement membrane (BM), in contrast to pericytes (Extended Data Fig. 1g–i). To deplete ADAM12⁺ MSCs, we inoculated ADAM12-DTR mice with MO5 melanoma cells; in these mice, the diphtheria toxin receptor (DTR) is expressed under the control of the *Adam12* promotor[27]. Depletion of ADAM12⁺ cells starting 10 d after tumor implantation, when tumors were palpable, resulted in 50% inhibition of tumor growth (Fig. 1d and Extended Data Fig. 1l; efficiency of depletion of ADAM12⁺ cells is shown in ref. 27 and Extended Data Fig. 1j,k). By contrast, depletion of ADAM12⁺ cells in the initial stages of tumorigenesis did not inhibit tumor growth (Fig. 1e and Extended Data Fig. 1m), arguing against an initial feeder role for stromal cells. Tumors lacking ADAM12⁺ cells from day 10 had increased infiltration of interferon-γ (IFN-γ)-producing CD8⁺ T cells and natural killer (NK) cells (Fig. 1f), whereas no difference was observed in infiltration of CD4⁺ T cells, regulatory T cells (T_reg cells), eosinophils, neutrophils, dendritic cells (DCs), myeloid-derived suppressor cells (MDSCs) or total macrophages (Extended Data Fig. 1n). We did not measure significant differences in CD8⁺ T cells or stromal cells in the draining lymph nodes (LNs), consistent with a local effect (Extended Data Fig. 1o,p). Treatment with CD8⁺ T cell-depleting antibodies restored tumor growth in the absence of ADAM12⁺ cells, confirming that ADAM12⁺ cells block antitumor activity of CD8⁺ T cells (Extended Data Fig. 1q).

## Depletion of ADAM12⁺ MSCs normalizes the stromal and vascular TME

Consistent with an active antitumor immune response, T cells infiltrated the center of tumors depleted of ADAM12⁺ cells, in proximity to PDPN⁺ CAFs that had migrated intratumorally and settled near blood vessels (Fig. 2a,b,d). The frequency of T cells in proximity to PDPN⁺ CAFs was similar in depleted and control conditions (Extended Data Fig. 2a), arguing against PDPN⁺ CAFs having an increased capability to recruit T cells[34]. The frequencies and absolute numbers of PDPN⁺ CAFs were similar in both conditions (Fig. 2c and Extended Data Fig. 2b), suggesting that CAFs became permissive to T cells. To identify changes occurring in PDPN⁺ CAFs in tumors infiltrated by T cells, we performed RNA-seq gene expression analysis of PDPN⁺PDGFRα⁺ cells isolated from wild-type (WT) tumors (which were poorly infiltrated) or from tumors lacking ADAM12⁺ cells (which were highly infiltrated) (the gating strategy is provided in Extended Data Fig. 9). Differential gene expression analysis in PDPN⁺PDGFRα⁺ cells in these two conditions identified a few genes, including *Car9*, *Saa3*, *Lcn2*, *Mmp9* and *Mmp13*, that were significantly downregulated in the CAFs of tumors lacking ADAM12⁺ cells (DTR versus control) (Fig. 2e). These genes all have well-recognized protumor roles[35–39], and their expression is commonly induced by hypoxia. Hypoxia-induced *Car9* (carbonic anhydrase

9, CAIX) prevents cytosolic acidification by catalyzing the hydration of carbon dioxide into bicarbonate ions and protons. CAIX expression enhances cell survival in hypoxic conditions and increases acidification of the tumor microenvironment, a major immunosuppressive factor[40,41]. Consistent with a role for ADAM12[+] cells in tumor hypoxia and acidosis, tumors lacking these cells showed decreased hypoxia (Fig. 2f) and increased extracellular pH (Fig. 2g). As oxygen levels were restored in the absence of ADAM12[+] cells, *Car9* expression was significantly decreased in the total tumor (Fig. 2h). To investigate whether normalization of the pH of the tumor microenvironment alone was sufficient to restore tumor immunity, we treated WT mice bearing MO5 melanomas with the CAIX inhibitor U-104. We observed that, although inhibition of CAIX normalized the extracellular tumor pH to levels similar to those achieved by depletion of ADAM12[+] cells (Fig. 2g), antitumor immunity was not restored (Extended Data Fig. 2c). Furthermore, we observed that, in contrast to tumors lacking ADAM12[+] cells, U-104-treated tumors were still hypoxic (Extended Data Fig. 2d). Tumor hypoxia results mainly from poorly functional, collapsed tumor vasculature that lacks pericyte coverage, which is required for proper vessel maturation and function[42]. Consistent with normalization of the vasculature, blood vessels of tumors lacking ADAM12[+] cells had restored levels of NG2[+] pericyte coverage and increased width (Fig. 2i and Extended Data Fig. 2e), as well as increased expression of ICAM1 (Fig. 2j), which is essential for leukocyte adhesion and *trans*-endothelial migration, and improved tumor perfusion (Fig. 2k). Of note, genes upregulated in PDPN[+]PDGFRα[+] cells in tumors lacking ADAM12[+] cells included regulators of PDGF and BMP signaling (Fig. 2e), also involved in vascular maturation and normalization[43,44]. Previous single-cell RNA-seq (scRNA-seq) studies of murine melanomas identified three clusters of CAFs, referred to as immune/inflammatory DPP4[+]CD34[hi] stroma (S1), desmoplastic stroma (S2) and contractile stroma/pericytes (S3)[9]. In tumors depleted of ADAM12[+] cells, we observed a significant decrease in the frequency of S1 cells, whereas S2 and S3 increased (Extended Data Fig. 3d; the fluorescence-activated cell sorting (FACS) gating strategy and expression of marker genes in S1, S2 and S3 subsets are shown in Extended Data Fig. 3a–c). As CAFs in the S3 subset, and those in S2 to a lesser degree, expressed higher levels of the pericyte markers *Rgs5* and *Cspg4* (coding for NG2) than did those in S1 (Extended Data Fig. 3b,c), these data are in line with the increased pericyte coverage observed in tumors depleted of ADAM12[+] cells (Fig. 2i). We further investigated the mechanism. The decrease in S1 CAFs was likely not due to direct ablation of ADAM12[+] cells, as most ADAM12[+] cells had low or no expression of DPP4 and medium or low expression of CD34, consistent with previous scRNA-seq data[9] (Extended Data Fig. 3e,f), and represented a small percentage of total stromal cells (Fig. 1c). Because ablation of ADAM12[+] cells decreased tumor hypoxia (Fig. 2f), we asked whether hypoxia affected CAF differentiation toward S1. Accordingly, we observed that hypoxia induced upregulation of *Cd34*, *Dpp4*, *C3*, *Il6ra* and *Il6st* (marker genes of the S1 stromal population; Extended Data Fig. 3b) in stromal cells in vitro, whereas expression levels of genes coding for broad fibroblast markers, such as *Pdgfra* and *Pdpn*, were unaffected (Extended Data Fig. 3g). These data are in line with previous reports showing that inflammatory CAFs are enriched in tumor hypoxic regions[45,46] and further suggest that depletion of ADAM12[+] MSCs normalizes CAFs by decreasing tumor hypoxia.

## Slow-cycling ADAM12[+] MSCs regulate the TME in a TGF-β-dependent way

These data show that a small subset of ADAM12[+] MSCs developing early at the tumor margins strongly affect the vascular and stromal tumor microenvironment. To investigate the underlying mechanism, we performed a transcriptome analysis of ADAM12[+] cells isolated from MO5 melanomas at day 8. Differential gene expression analysis identified >700 genes that were significantly upregulated in ADAM12[+]PDPN[+]PDGFRα[+] cells compared with ADAM12[−]PDPN[+]PDGFRα[+]

cells (Fig. 3a). Gene set enrichment analysis indicated that among the top enriched pathways for ADAM12[+] cells were terms related to ECM remodeling, cell proliferation and differentiation, and inflammation, including major signaling pathways downstream of growth factors receptors (RTK) and cytokine receptors, such as phosphoinositide 3-kinase (PI3K)–Akt, MAPK, nuclear factor-κB (NF-κB), tumor necrosis factor (TNF) and JAK–STAT signaling pathways (Fig. 3b). Several pathways related to cell metabolism, protein synthesis and DNA replication were downregulated in ADAM12[+] cells compared with ADAM12[−] cells (Fig. 3c). In line tumors with ADAM12[+] cells having a lower metabolism, ADAM12[+] cells downregulated expression of genes involved in cell proliferation, glycolysis and oxidative phosphorylation, such as *Mcm2*, *Cdk2*, *Cdk15*, *Ldhb*, *Ldha* and *Atp5a1*, and upregulated expression of *Gadd45b*, *Gadd45g* and *Gpx3*, which are induced upon cell cycle arrest in response to oxidative or stress damage (Fig. 3d). Further suggesting that there is crosstalk within the TME, ADAM12[+] cells expressed higher levels of *Pdgfra*, *Il1r1* (which encodes an activator of NF-κB), *Osmr* and *Il6st* (gp130), which encodes a signal transducer in a receptor complex involving several cytokines, including oncostatin (OSM) and interleukin-6 (IL-6) (Fig. 3d). They also upregulated *Il6*, genes encoding chemokines including *Cxcl1*, and genes with key roles in monocyte recruitment and macrophage polarization, such as *Ccl2*, *Ccl7*, *Csf1* and *Has2* (Fig. 3d), as well as growth arrest specific 6 (*Gas6*), *Lgals3* and *Pros1*, which promote macrophage efferocytosis through the TAM (TYRO3, AXL and MERTK) tyrosine kinase receptors[47]. In addition, ADAM12[+] cells upregulated *Icam1*, which encodes an adhesion molecule, and protumor components of the ECM such as *Ctgf*, *Lox*, *Hspg2*, *Mmp3* and *Tgfbr3* (Fig. 3d)[48–50], suggesting that these cells have a role in TME remodeling. Compared with ADAM12[−]PDGFRα[+] cells, ADAM12[+]PDGFRα[+] cells upregulated genes overexpressed by mesenchymal progenitors, such as *Ngfr*, *Ly6a* (sca-1), *Vcam1* (CD106) and PDGFRβ, but not *Acta2* (coding for αSMA) or *Lrrc15* (Fig. 3d and Extended Data Figs. 1g and 4a). These data suggest that ADAM12[+] cells are perivascular mesenchymal progenitors that are distinct from αSMA[+] myofibroblasts and Lrrc15[+] CAFs[51].

The majority of ADAM12[+] MSCs did not express Ki-67 or incorporate Edu, in contrast to CAF populations (Fig. 3e and Extended Data Fig. 4b), consistent with the cells being in a slow-cycling state. The absence of senescence markers, such as *Cdkn2a* (p16-INK4A) and *Cdkn1a* (P21), argued against a senescent state, characterized by an irreversible arrest of cell proliferation and growth. Accordingly, and differently from senescent cells, which are unresponsive to mitogenic stimuli, ADAM12[+] MSCs rapidly resumed cell proliferation when given nutrients in vitro (Fig. 3f). TGF-β has a major role in cell cycle regulation[52]. In addition to inducing *Adam12* expression (consistent with previous reports[24,26,27]), TGF-β rapidly induced expression of *Gadd45b* and *Gadd45g*, which encode cycle-arrest proteins, in ADAM12[−]PDGFRα+PDPN+ cells isolated from MO5 tumors, and downregulated cyclins involved in cell proliferation, such the one encoded by *Cdk6* (Fig. 3g). ADAM12[+] cells overexpressed a number of genes encoding receptors for inflammatory cytokines and growth factors (Fig. 3d), including *Il1r1*, *Osmr* and *Il6st*, suggesting that there was additional crosstalk within the TME. Within the TME, TAMs expressed the highest levels of *Il1b*, *Osm* and *Pdgfc* (Extended Data Fig. 4c). Although IL-1β and OSM did not induce *Adam12* expression (Extended Data Fig. 4d), IL-1β induced *Cxcl1*, *Ccl2*, *Il6* and *Mmp3* expression in ADAM12[+] MSCs isolated from MO5 tumors, and OSM rapidly induced expression of *Il6*, *Icam1*, *Has2*, *Il1r1* and *Gadd45g* (Fig. 3h). Additional factors overexpressed by ADAM12[+] cells, such as *Gas6*, were strongly induced by starvation in stromal cells (Extended Data Fig. 4e), overall suggesting that the inflammatory and nutrient-deprived TME further modulated the phenotype of TGF-β-induced ADAM12[+] cells.

*Tgfb1* was upregulated by several cell types within the TME in early tumor stages, and TAMs were major producers of *Tgfb1* in advanced tumors (Extended Data Fig. 4f,g). As ADAM12[+] cells expressed high levels

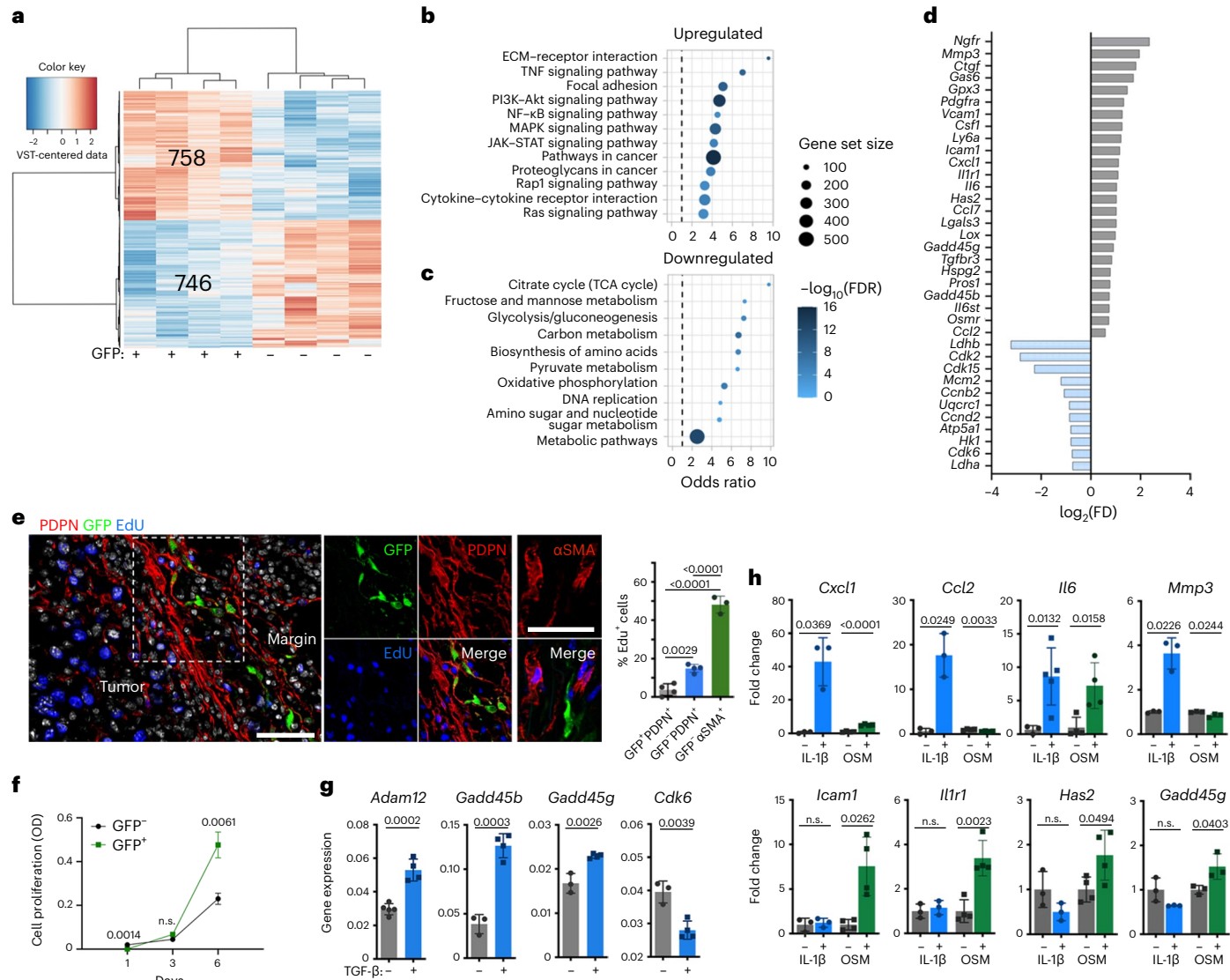

**Fig. 3 | Slow-cycling ADAM12⁺ MSCs promote protumor inflammation and tissue remodeling. a**, Heat map of RNA-seq differential gene expression analysis of ADAM12⁺ PDPN⁺PDGFRα⁺ (GFP⁺) versus ADAM12⁻PDPN⁺PDGFRα⁺ (GFP⁻) CD45⁻CD31⁻ cells isolated by FACS from MO5 tumors growing in ADAM12-GFP mice (n = 4). **b,c**, Pathway enrichment analysis of genes significantly upregulated (**b**) or downregulated (**c**) in GFP⁺ cells, using KEGG annotation. **d**, Differential gene expression analysis (GFP⁺ cells versus GFP⁻ cells). The bar plots represent log₂(fold change). **e**, Left, immunofluorescence staining of the indicated markers in MO5 tumors growing in ADAM12-GFP mice. One representative image of three to four independent experiments is shown. Right, quantification of staining in

the left panel. Scale bars, 50 µm. **f**, Growth curve of GFP⁺ and GFP⁻ cells isolated from MO5 tumors, n = 4 (GFP⁺)–7 (GFP⁻). OD, optical density. **g**, Expression of the indicated transcripts, measured by qRT–PCR, in ADAM12⁻ PDGFRα⁺ cells isolated from MO5 tumors and treated with TGF-β; n = 3 (−TGF-β)–4 (+TGF-β), except for *Adam12*, n = 5 (−TGF-β)–4 (+TGF-β). **h**, Expression of the indicated transcripts, measured by qRT–PCR, in ADAM12⁺PDGFRα⁺ cells treated with IL-1β or OSM (fold change treated versus non treated). +IL-1β, n = 3, except for *Il6*, n = 5; OSM, n = 4, except for *Mmp3* and *Gadd45g*, n = 3. Statistics were calculated using one-way ANOVA (**e**), two-way ANOVA (**f**) or two-tailed, unpaired Student's *t*-test (**g,h**). All quantitative data are presented as means ± s.d.

of *Tgfbr2* (Extended Data Fig. 4h,i), and ADAM12 enhanced TGF-β signaling in vitro[27], we investigated the role of TGFBR2 in ADAM12⁺ cells in MO5 tumors in vivo. To that aim, we generated a tetracycline-regulated ablation model of *Tgfbr2* in ADAM12⁺ cells by crossing ADAM12-tTA mice[27] with tet-controlled Cre (LC-1 mice) and Tgfbr2^loxP/loxP mice (ADAM12-tTA-Cre^Tgfbr2 mice, Extended Data Fig. 4j). As TGF-β expression is induced at early tumor stages (Extended Data Fig. 4f), we started ablating *Tgfbr2* in ADAM12⁺ cells by removing doxycycline at the initiation of tumorigenesis (Extended Data Fig. 4k). We observed significant inhibition of tumor growth, as well as increased infiltration of T cells, in ADAM12-tTA-Cre^Tgfbr2 mice compared with WT littermate mice (Extended Data Fig. 4l,m), showing that TGFBR2 is required for the pro-tumorigenic role of ADAM12⁺ cells. Macrophages isolated from MO5 tumors growing in ADAM12-tTA-Cre^Tgfbr2 mice expressed lower levels of *Vegfa*, a major

inducer of leaky vessels, and tumor vessels had improved pericyte coverage (Extended Data Fig. 4n,o), consistent with blood vessel normalization. Accordingly, stromal cells isolated from MO5 tumors growing in ADAM12-tTA-Cre^Tgfbr2 mice expressed higher levels of *Angpt1* and *Pdgfrb*, which stabilize the vasculature through pericyte–endothelial cell interaction[53], and lower levels of *Car9*, induced in hypoxia[39], compared with stromal cells isolated from WT tumors (Extended Data Fig. 4p). These data show that TGFBR2 signaling in ADAM12⁺ cells is required for their pro-tumorigenic function and TME alterations.

## ADAM12⁺ MSCs promote macrophage efferocytosis and polarization

Further suggesting a crosstalk between ADAM12⁺ MSCs and macrophages, ADAM12⁺ cells accumulated around MO5 tumors close to

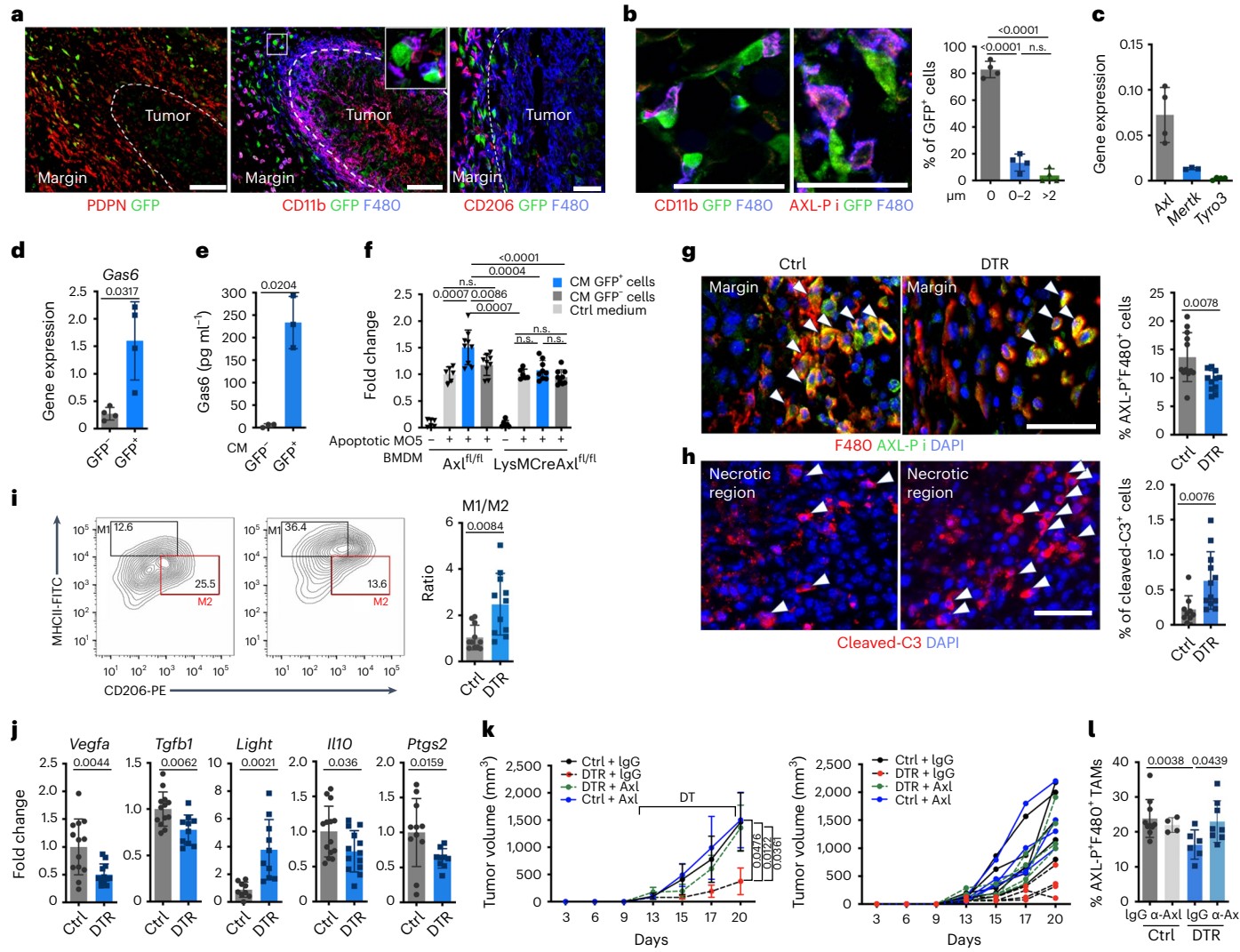

**Fig. 4 | ADAM12⁺ MSCs induce immunosuppressive macrophages by promoting efferocytosis. a,b,** Immunofluorescence staining of the indicated markers in MO5 tumors at 8 d (**a**) and 12 d (**b**) growing in ADAM12-GFP mice. Scale bar in **a**, 100 μm (left, middle) and 50 μm (right); in **b**, 50 μm. In **b**, distance of GFP⁺ cells to AXL-P⁺ macrophages (right). Results are representative of four independent experiments. **c**, Gene expression of *Axl*, *Mertk* and *Tyro3*, measured by qRT–PCR, in macrophages isolated from MO5 tumors (*n* = 4). **d**, Gene expression of *Gas6*, measured by qRT–PCR, in ADAM12⁺ and ADAM12⁻ cells isolated from MO5 tumors; *n* = 4 from independent experiments. **e**, Secretion of Gas6, measured by ELISA (*n* = 3). Results are representative of three independent experiments. **f**, Efferocytosis test in BMDMs isolated from the indicated mice in presence of CM obtained from GFP⁺ or GFP⁻ cells. *n* = 9 (+CM GFP⁺/GFP⁻) and *n* = 6 (Ctrl medium). Results are representative of three independent experiments. **g,h,** Left, immunofluorescence staining of the indicated markers in tumors growing in DTR or Ctrl mice injected with DT. Right, quantification of expression, which was performed on *n* = 12(Ctrl)–11(DTR) in **g** and *n* = 9(Ctrl)–12(DTR) in **h**. Results are representative of three independent experiments. White arrowheads indicate F480⁺AXL-P⁺ macrophages (**g**) and cleaved-C3⁺ apoptotic cells (**h**).

Scale bars, 50 μm. **i**, FACS plot and percentages of macrophages expressing MHCII and CD206 in tumors growing in DTR or Ctrl mice treated with DT (left). Ratio of MCHII^hi^CD206^lo^ (M1) over MCHII^lo^CD206^hi^ (M2) macrophages (right). *n* = 12(Ctrl)–11(DTR) from 2 independent experiments. **j**, Expression of the indicated transcripts, measured by qRT–PCR, in macrophages isolated by FACS from tumors growing in mice treated as in **i**; *Vegfa*, *n* = 13(Ctrl)–12(DTR); *Tgfb1*, *n* = 13(Ctrl)–10(DTR); *Light*, *n* = 9(Ctrl)–10(DTR); *Il10*, *n* = 14(Ctrl)–13(DTR); *Ptgs2*, *n* = 11(Ctrl)–9(DTR). Results are representative of three independent experiments. **k**, Tumor growth curves of DTR and Ctrl mice treated with DT and activating anti-AXL antibodies or isotype control (IgG). Left, average tumor volume; right, growth curves for individual animals; *n* = 4, except for Ctrl + IgG (*n* = 6). The *x* axis represents days after tumor inoculation. **l**, Percentage of AXL-P⁺ macrophages in tumor sections from mice treated as in **k**. Quantifications were performed on *n* = 10 (Ctrl + IgG), *n* = 4 (Ctrl + α-AXL), *n* = 6 (DTR + IgG), *n* = 8 (DTR + α-AXL) fields. Statistics were calculated using one-way ANOVA (**b**), two-tailed, unpaired Student's *t*-test (**d,e,g,i,j**), Mann–Whitney *U* test (**h,j** *Ptgs2*) or two-way ANOVA (**f,k,l**). All quantitative data are presented as means ± s.d.

CD206⁺ TAMs, starting in early tumor stages (Fig. 4a). In larger tumors, ADAM12⁺ cells remained localized at the tumor margin, and a majority (>80%) were in close proximity to macrophages with phosphorylated AXL (AXL-P⁺, Fig. 4b), a receptor that is essential for efferocytosis, and were distant from T cells (Extended Data Fig. 5a). By contrast, only 20–40% of CAFs were adjacent to AXL-P⁺ macrophages (Extended Data Fig. 5b). Engulfment of apoptotic cells by macrophages through

efferocytosis induces anti-inflammatory cytokines such as TGF-β, IL-10 and prostaglandin E2 (PGE2), which promote tumorigenesis[54,55]. Efferocytosis is enhanced by molecules such as Gas6, which bridges phosphatidylserine on apoptotic cells with TAM receptors[54]. Of note, Gas6 has a higher affinity for AXL receptors than for Mertk or Tyro3 receptors, which were expressed at lower levels on tumor macrophages (Fig. 4c). In line with our sequencing data, PDPN⁺PDGFRα⁺ cells had a higher

*Gas6* expression level than did other cell types in the tumors (Extended Data Fig. 5c); ADAM12[+]PDPN[+]PDGFRα[+] cells had particularly high *Gas6* expression levels, both at the RNA and protein level (Fig. 4d,e). Although CAFs did not show efferocytic activity (Extended Data Fig. 5d), the conditioned medium (CM) from ADAM12[+] MSCs (GFP[+] cells) isolated from MO5 tumors increased efferocytosis of bone-marrow-derived macrophages (BMDMs), an effect that was inhibited when BMDMs lacked AXL in myeloid cells (isolated from LysM-CreAXL[fl/fl] mice) (Fig. 4f, AXL[fl/fl] littermates were used as WT controls). Efferocytic BMDMs treated with CM from GFP[+] cells expressed higher levels of *Tgfb1*, *Vegfa* and *Il10* than did BMDMs treated with CM from GFP[−] cells (Extended Data Fig. 5e), consistent with efferocytosis-induced immunosuppression[54,56]. A similar process occurred in vivo: depletion of ADAM12[+] cells induced a significant decrease of AXL-P[+] macrophages (Fig. 4g), whereas the frequencies of apoptotic cells, mostly non-perivascular and non-stromal cells, were increased (Fig. 4h and Extended Data Fig. 5f). Accordingly, the engulfing ability of macrophages isolated from tumors lacking ADAM12[+] cells was decreased compared with that of macrophages isolated from WT tumors (Extended Data Fig. 5g). These data are consistent with decreased expression of *Gas6* in tumors depleted of ADAM12[+] cells compared with WT tumors (Extended Data Fig. 5h), showing that ADAM12[+] cells are a significant source of Gas6 within the TME. In line with a role for ADAM12[+] cells in this process in vivo, the ratio of major histocompatibility complex II (MHCII)[hi]CD206[lo] inflammatory macrophages to MHCII[lo]CD206[hi] TAMs significantly increased in tumors lacking ADAM12[+] MSCs (Fig. 4i). Macrophages isolated from tumors lacking ADAM12[+] cells expressed lower levels of *Tgfb1*, *Il10*, *Ptgs2* (coding for PGE2) and *Vegfa* (Fig. 4j) and upregulated *Light* (*Tnfsf14*), which promotes antitumor immunity by activating NK cells, T cells, stromal cells and restoring blood vessels integrity[57,58]. Accordingly, treatment with anti-AXL activating antibodies[59] restored tumor progression in MO5 tumors lacking ADAM12[+] cells, and increased the AXL-P[+] macrophage level to that in WT mice (Fig. 4k,l), whereas it had no effect on WT tumors. Overall, these data show that ADAM12[+] MSCs are induced at early tumor stages and polarize tumor macrophages toward an immunosuppressive and proangiogenic phenotype by promoting efferocytosis.

## Tumor-induced ADAM12[+] lineage is maintained in advanced stages

To determine whether ADAM12[+] MSCs develop in spontaneously occurring tumors, we crossed ADAM12-GFP mice with Rip-Tag2 mice, a mouse model of neuroendocrine pancreatic tumor, or with TRAMP mice, a mouse model of prostate adenocarcinoma. In these models, expression of the SV40 large T antigen drives multi-step tumor progression with tumor stages similar to those in human cancer. In both tumors, we observed PDPN[+] stromal cells and macrophages accumulating in peritumoral areas (Extended Data Fig. 6a,b). We observed around 1–6% of stromal cells expressing ADAM12 and localized close to blood vessels at the margins of Rip-Tag2 and TRAMP tumors, but not in normal pancreas or prostate (Fig. 5a,b). Similar to what happened in the melanoma, ADAM12[+] cells, in comparison to ADAM12[−] cells, upregulated genes involved in cell cycle arrest, growth factors and cytokine receptors, as well as genes regulating macrophages and the ECM, including *Gas6*, *Csf1*, *Has2*, *Mmp3* and the mesenchymal progenitor markers *Ly6a* and *Vcam1*, and downregulated *Mki67*, *Cdk1*, *Cdkn2a* and *Acta2* (Extended Data Table 1 and Extended Data Fig. 7a). ADAM12[+] cells expressed low to medium levels of PDGFRβ and were localized outside the ColIV[+] vascular BM (Fig. 5a, left, Extended Data Fig. 7a, right). To determine the fate of ADAM12[+] MSCs in vivo during tumor progression, we generated a tetracycline-regulated lineage-tracing system of ADAM12[+] cells by crossing ADAM12-tTA mice[27] with LC-1 mice and Rosa26[STOPfloxYFP] reporter mice (ADAM12-tTA-Cre[YFP] mice, Fig. 5c). After inoculating MO5 tumor cells in ADAM12-tTA-Cre[YFP] mice (maintained on doxycycline until tumor injection), we observed that yellow fluorescent protein (YFP)[+]

cells (progeny of ADAM12[+] cells) constituted about 1% of stromal cells in advanced melanomas. YFP[+] cells were localized at the tumor margins, close to vessels and within the stroma, and expressed varying levels of NG2 and αSMA (Extended Data Fig. 7b). Compared with ADAM12[+] cells, YFP[+] cells downregulated several factors that are essential for immune-cell crosstalk, including *Ccl2*, *Csf1*, *Gas6*, *Lgals3*, *Cxcl12*, *Il1r1*, *Osmr*, *Il6* and *Angpt1*, while upregulating varying levels of *Car9*, *Angpt2*, *Acta2* and *Pdgfrb* (Extended Data Fig. 7c). As *Car9* and *Angpt2* encode proteins that block antitumor immunity[40,41,60], these data suggest that both ADAM12[+] cells and their progeny promote immunosuppression, although through different mechanisms. To determine whether a similar lineage develops in tumors that arise spontaneously, we crossed ADAM12-tTA-Cre[YFP] mice (Fig. 5c) with TRAMP mice or Rip-Tag2 mice. In prostate and pancreatic tumors, we observed YFP[+] cells within the stroma and close to blood vessels that expressed medium to low levels of αSMA and NG2 (Fig. 5e and Extended Data Fig. 7d). Although they were in proximity to blood vessels, YFP[+] cells were localized outside the vascular BM, in contrast to pericytes[61], consistent with detached pericyte-like cells or perivascular αSMA[mid] fibroblastic cells (Extended Data Fig. 7d). By removing doxycycline at different stages of tumorigenesis (Fig. 5d), we observed that ADAM12[+] cells induced at early tumor stages in prostate and pancreatic tumors had increased stromal progenitor potential (Fig. 5f, early, and Extended Data Fig. 7e), compared with ADAM12[+] cells at later tumor stages (Fig. 5f, late, and Extended Data Fig. 7e). After treating with doxycycline 10-week-old TRAMP x ADAM12-tTA-Cre[YFP] mice that had been fate mapped from week 4 (Fig. 5g), we analyzed the fate of ADAM12[+] cells induced specifically at early tumor stages. In this setting, we observed YFP[+]αSMA[−] or YFP[+]αSMA[mid] cells at the tumor margins of large prostate adenocarcinomas in 30-week-old TRAMP mice, demonstrating that ADAM12[+] cells induced in early tumors generate a peritumoral lineage that is maintained during tumor progression (Fig. 5h). In line with the gene expression data (Extended Data Table 1), YFP[+] cells were mostly negative for the proliferation marker Ki-67 (Fig. 5e, right panel). A slow-cycling YFP[+]Ki-67[−] cells were also abundant at the interface of normal tissue and SV40[+] prostate cancer cells that metastasized to the liver and bone in TRAMP × ADAM12-tTA-Cre[YFP] mice (Fig. 5i), further suggesting a role for ADAM12[+] MSCs in the metastatic niche. To investigate the developmental origin of tumor-induced ADAM12[+] MSCs, we performed lineage tracing of ADAM12[+] cells from development, because ADAM12 is expressed during organ morphogenesis[27]. In ADAM12-Cre[YFP] mice, we observed that a majority of stromal cells in healthy skin, prostate and pancreas in adults were generated from fetal ADAM12[+] progenitors (Extended Data Fig. 7f), suggesting that tumorigenesis reactivates a developmental program. Finally, to assess the role of ADAM12[+] MSCs in a spontaneous tumor model, we crossed ADAM12-DTR mice with Rip-Tag2 mice to generate RIP[+]DTR[+] and RIP[+]DTR[−] mice. Depletion of ADAM12[+] cells induced significant growth inhibition of RIP tumors, which displayed increased infiltration of CD3[+] T cells (Extended Data Fig. 7g–i). As observed in the melanoma, PDPN[+]PDGFRα[+] CAFs were reorganized but still present in similar numbers in tumors lacking ADAM12[+] MSCs, and the vasculature displayed increased pericyte coverage and ICAM1 expression. Tumors were better perfused (Extended Data Fig. 7j–n), overall confirming a similar role for ADAM12[+] cells in this tumor model. These data show that slow-cycling ADAM12[+]PDGFRα[+]αSMA[−] perivascular MSCs induced at early stages of tumorigenesis generate a discrete mesenchymal lineage that is maintained and active in advanced carcinomas and metastasis.

## ADAM12 can stratify patients with cancer across tumor types

ADAM12 is overexpressed in several solid tumors, including melanoma, prostate, breast, liver, colorectal and pancreatic tumors[22–26] and is associated with stromal activation and a poor prognosis[23,26,28–33]. As observed in mice models, ADAM12 was preferentially expressed by stromal populations in several human tumors (Extended Data Fig. 8a–e), consistent with previous reports[24,62–64]. In human pancreatic

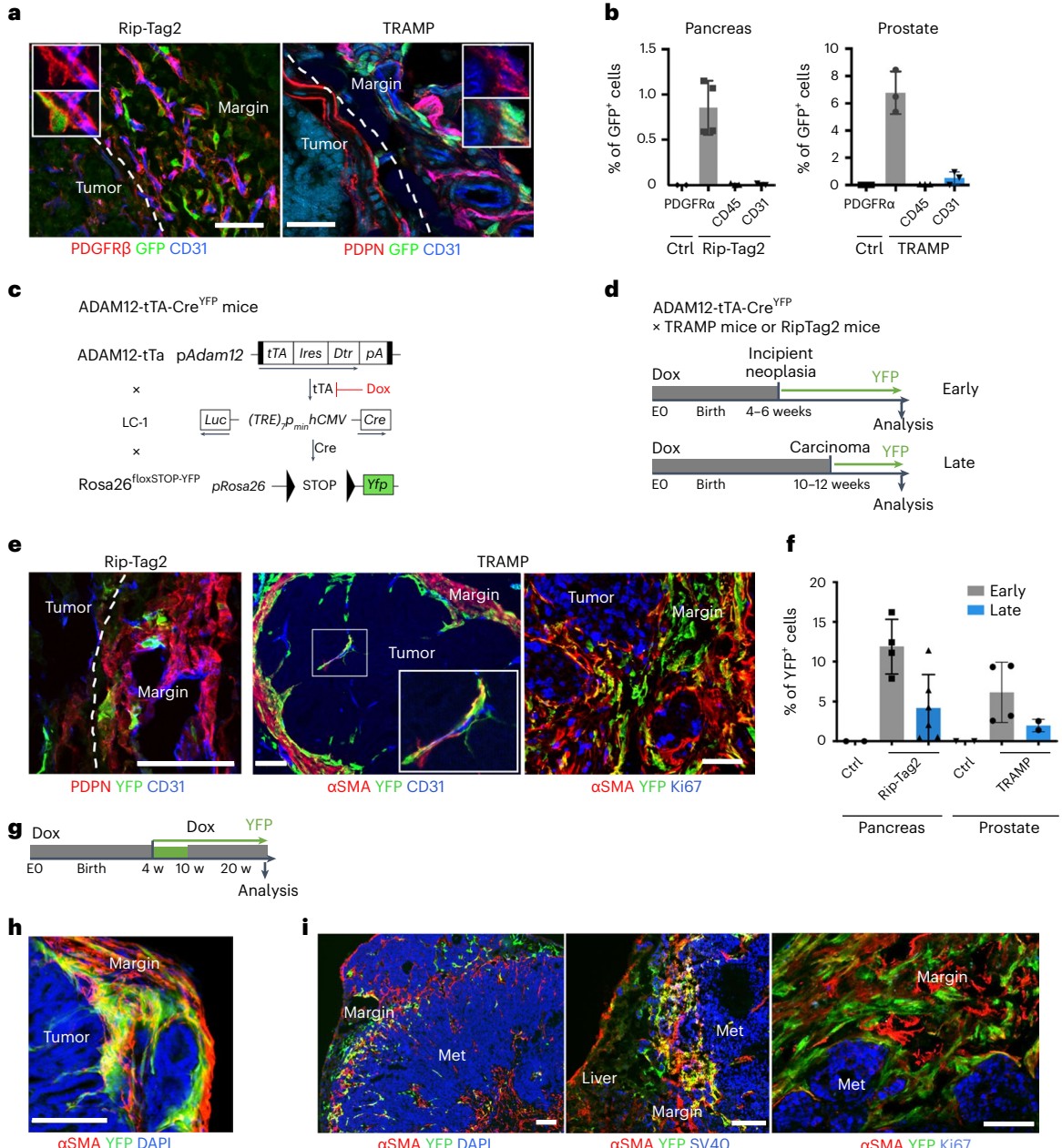

**Fig. 5 | The tumor-induced ADAM12⁺ lineage is maintained in advanced tumor stages. a**, Immunofluorescence staining of ADAM12⁺ cells (GFP), co-stained with the indicated markers, in Rip-Tag2 (left) and TRAMP (right) tumors growing in ADAM12-GFP mice. Scale bars, 50 μm. **b**, Percentage of GFP⁺ cells among the indicated populations in TRAMP tumors ($n = 3$), RIPTag tumors ($n = 4$ for stroma, $n = 3$ for CD45⁺ and CD31⁺ cells), Ctrl prostate ($n = 3$) and Ctrl pancreas ($n = 2$), measured by FACS. Results are representative of two independent experiments. **c**, Strategy for inducible fate mapping of ADAM12⁺ cells. tTA, tetracycline transactivator; Dtr, diphteria toxin receptor; Ires, internal ribosomal entry site; Luc, luciferase; TRE, tet-responsive element; hCMV, human cytomegalovirus; Dox, doxycycline; LC-1, Luciferase_Cre transgenic mice[74]. **d**, Experimental setup for lineage tracing of ADAM12⁺ cells induced de novo at early or late tumor stages in TRAMP or Rip-Tag2 mice. **e**, Immunofluorescence staining of YFP and the indicated markers, in Rip-Tag2 and TRAMP tumors growing in ADAM12-tTA-Cre^YFP mice. Scale bars, 50 μm. **f**, Percentage of YFP⁺ cells among total tumor stromal cells, measured by FACS, from mice treated as in **d**. $n = 4$ (RIP early), $n = 6$ (RIP late), $n = 4$ (TRAMP early), $n = 2$ (TRAMP late), $n = 2$ (Ctrl). **g**, Experimental setup for lineage tracing of ADAM12⁺ cells induced at early stages of tumorigenesis. **h**, Immunofluorescence staining of the indicated markers in TRAMP prostate tumors growing in ADAM12-tTA-Cre^YFP mice treated as in **g**. Scale bar, 100 μm. **i**, Immunofluorescence staining of the indicated markers in TRAMP tumors that metastasized in the liver (left and middle) or bone (right) in ADAM12-tTA-Cre^YFP mice. SV40 stains tumor cells. Scale bars, 50 μm. In **a**, **e**, **h**, and **i**, images are representative of independent experiments ($n = 3$–$6$). Quantitative data are presented as means ± s.d. Met, metastasis. DAPI stains nuclei.

ductal adenocarcinoma, ADAM12 expression was correlated with the 'stroma activated' subtype, which is associated with a severe prognosis[65] (Extended Data Fig. 8d). We further analyzed ADAM12 expression in publicly available datasets from cohorts of patients with skin cutaneous melanoma, pancreatic ductal adenocarcinoma, prostate adenocarcinoma or colon adenocarcinoma. For each cohort, we stratified tumors by ADAM12 expression (ADAM12 expression < median expression versus ADAM12 expression > median expression). Gene Set Enrichment Analysis of pathways using the Hallmark gene sets indicated that, independently of the tumor type, ADAM12^hi tumors were significantly enriched in genes related to the terms 'Inflammatory response', 'Hypoxia' and 'Angiogenesis', and were inversely correlated

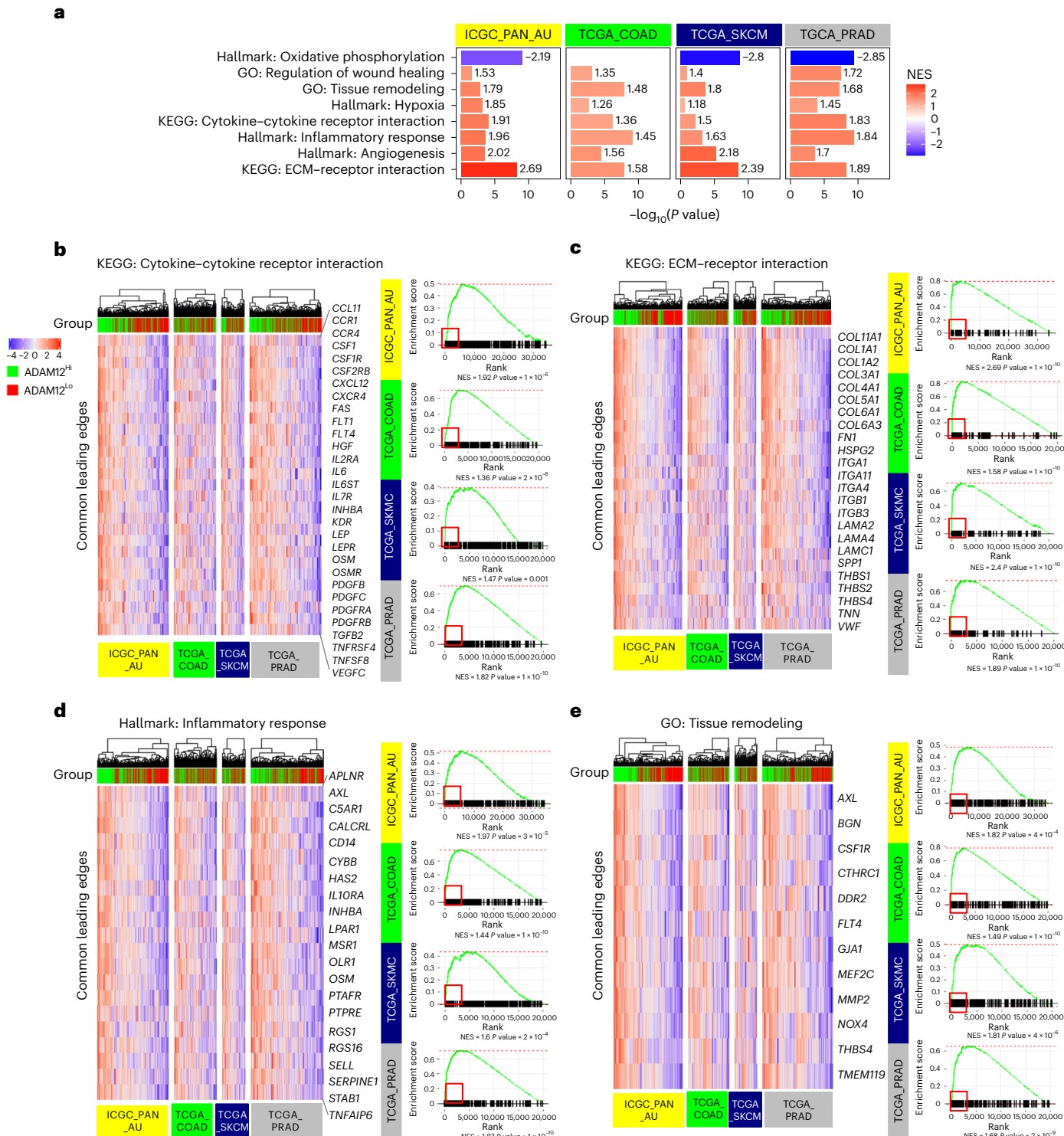

**Fig. 6 | ADAM12 stratifies patients with high levels of hypoxia, inflammation and innate resistance mechanisms across tumor types. a**, GSEA of pathways in ADAM12[hi] versus ADAM12[lo] tumors (median expression) of human pancreatic ductal adenocarcinoma (ICGC_PAN_AU; yellow, $n = 267$), colon carcinoma (TCGA_COAD; green, $n = 450$), melanoma (TCGA_SKCM; navy, $n = 147$) and prostate adenocarcinoma (TCGA_PRAD; gray, $n = 455$) datasets. The bar plots represent the $-\log_{10}(P\text{ value})$ and are colored according to the normalized enrichment score (NES) (red, positive; blue, negative; the value is shown at the end of each bar plot).

**b–e**, Gene set enrichment analysis (GSEA) curves for the indicated pathways for the four tumor datasets analyzed as in **a** (right, red square indicates the leading-edge genes). For each pathway, to the left of the GSEA curves, the heat map shows the expression levels of the genes present in the leading edge and shared between the four datasets. All patients whose data have been analyzed are represented, and whether they have high (green) or low (red) *ADAM12* expression is indicated above the heat map. A Wilcoxon rank-sum test was used for pre-rank GSEA for statistical analysis.

with 'Oxidative phosphorylation' (Fig. 6a), a process that requires oxygen. Pathway analysis using the Gene Ontology and Kyoto Encyclopedia of Genes and Genomes (KEGG) database in these tumors further indicated enrichment for the terms 'Cytokine–cytokine receptor interaction,' 'ECM–receptor interaction,' 'Tissue remodeling' and 'Regulation of wound healing' (Fig. 6a,b). Analysis of the leading edge (i.e., genes accounting for a pathway being defined as enriched) in the pathways 'Cytokine–cytokine receptor interaction', 'ECM–receptor interaction', 'Inflammatory response' and 'Tissue remodeling' further identified a number of shared genes in the four tumor datasets (Fig. 6b–e), suggesting that there is a shared gene expression program across solid tumors grouped by ADAM12 expression. Notably, and similar to the data obtained in mice models, common genes of the leading edge in the four human datasets included genes regulating or expressed by tumor macrophages, such as *AXL*, *CSF1*, *CSF1R*, *CD14* and *MSR1*, and genes encoding cytokines and cytokine receptors of the OSM, IL-6 and PDGF pathways, as well as genes encoding structural and regulatory proteins of the ECM with essential roles in tissue remodeling and angiogenesis, including collagens, laminins, HSPG2, HAS2, KDR and NOX4. Consistent with increased resistance mechanisms, stratification of human prostate cancer by ADAM12 expression further correlated with the Gleason score, which identifies high-grade tumors at high risk of recurrence (Extended Data Fig. 8f). These data show that, in several desmoplastic tumors, including pancreatic, prostate and colon cancer, ADAM12 expression stratifies patients harboring tumors with high levels of hypoxia, inflammation, tissue remodeling and innate resistance mechanisms, as well as factors associated with a poor prognosis and drug resistance such as AXL.

## Discussion

Here we show that tumor-induced stromal cell cycle arrest at the tumor margin coordinates angiogenesis, tissue remodeling and immunosuppression, key drivers of tumor progression. Selective depletion of such metabolically altered mesenchymal stromal subset, identified by expression of ADAM12, normalized the TME and decreased tumor hypoxia and acidosis, inducing infiltration of activated T cells and inhibition of tumor growth. We further provided direct genetic evidence that TGFBR2 signaling in ADAM12+ MSCs is required for their pro-tumorigenic function.

We showed that ADAM12+ MSCs are induced by TGF-β, a major cytostatic factor that is upregulated in early stages of tumorigenesis[5], and are further modulated by IL-1β and OSM, which have pivotal roles in cancer-associated inflammation and tumor initiation[66]. These results are consistent with previous reports showing that the NF-kB and PI3K pathways, activated by IL-1β and OSM, respectively, are associated with alterations of tumor-associated stroma cells and TAMs, vascularization and ECM remodeling[67–69]. Specifically localized in the perivascular niche at the tumor margins, ADAM12+ cells were major producers of factors regulating monocyte recruitment and macrophage function and polarization, including *Ccl2*, *Csf1* and *Has2*, as well as bridging molecules enhancing macrophage efferocytosis, such as *Gas6*, and promoted macrophage efferocytosis in an AXL-dependent way. Efferocytosis is a fundamental mechanism in prevention or resolution of tissue inflammation. In the context of cancer, it favors immunosuppression, angiogenesis and tumor progression[54–56]. Consistent with inhibition of macrophage polarization toward proangiogenic TAMs, depletion of ADAM12+ MSCs induced normalization of the tumor vasculature. ADAM12+ MSCs are restricted to the tumor margin, further suggesting that they spatially coordinate with TAMs to relay signals from the margin.

We show that, in tumors lacking ADAM12+ MSCs, the total number of CAFs was not reduced, but rather these cells became immunopermissive. We did not detect any significant differences in transcripts coding for collagens or chemokines in immunopermissive CAFs, consistent with previous reports showing that structural, vascular and

immune homeostatic functions of CAFs are essential to restrain tumor growth[15–21]. The major transcriptomic changes in immunopermissive CAFs were downregulation of hypoxia-induced genes such as *Car9* (CAIX), which promotes tumor acidification, a strong immunosuppressive factor. Consistent with restoration of normoxia, depletion of ADAM12+ MSCs increased the coverage of blood vessels with NG2+ pericytes, essential for vascular stabilization and function, and the number of inflammatory CAFs was reduced. These data are consistent with previous reports showing that inflammatory CAFs are enriched in tumor hypoxic regions[45,46], and further suggest that hypoxia is a major driver of immunosuppressive functions in CAFs[35–39].

ADAM12+ MSCs were in a 'dormant' slow-cycling state, distinct from senescence, as they were able to resume the cell cycle when provided with nutrients. The capacity to enter a slow-cycling state is a common feature of stem-like cells that promotes resistance to limited resources and antimitotic therapies, evasion from the immune system and dissemination of tumor cells[6]. Our data are consistent with a pro-tumor effect of cytostasis, mediated by TGF-β, on perivascular MSCs, as selective depletion of ADAM12+ MSCs or conditional ablation of *Tgfbr2* in ADAM12+ MSCs was sufficient to inhibit tumor growth. Therefore, TGF-β might promote pro-tumorigenic functions through different mechanisms in MSCs and CAFs[51]. Albeit distinct from pericytes, ADAM12+ MSCs and their progeny remained within the perivascular niches of the TME for months, resembling αSMAmid fibroblastic cells or detached pericyte-like cells. These populations have major roles in TME alterations and immunosuppression, notably by affecting perivascular TAMs and vascular function[42,70]. Finally, these data suggest that antitumor therapies aimed at promoting cell cycle arrest, cell damage or starvation might promote immunosuppression by inducing a low proliferative stem-like state in tumor-associated MSCs.

ADAM12 has been identified as a marker for stromal activation, poor prognosis and resistance to therapy in several human desmoplastic tumors, including pancreatic, liver, colorectal, lung, breast and ovarian cancers[23,26,28–33]. In agreement with a role for ADAM12 in this process, knock out of *Adam12* in a murine model of prostate cancer blocked tumor progression to poorly differentiated advanced stages[22]. We further show that ADAM12 expression stratifies patients with high levels of inflammation, hypoxia, innate resistance mechanisms, and tissue remodeling. These results are in line with increasing evidence that hypoxia and acidosis suppress antitumor immune responses[4,71–73]. Altogether, these data suggest that ADAM12+ MSCs are part of a fundamental repair process that is initiated by cytostatic TGF-β and modulated by inflammation, to promote angiogenesis and anti-inflammatory responses in coordination with macrophages. We propose that the pathological persistence of the ADAM12 lineage in tumors, which is normally eliminated from the tissue when healing is complete[27], plays a key role in inducing and maintaining an immunosuppressive and hypoxic TME. Although hypoxia and acidosis promote tumor malignancy, they are essential components of the repair process, notably for pathogen clearance and restoration of tissue homeostasis, suggesting an evolutionary advantage.

## Online content

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

## Methods

### Mice

We have previously described the generation of BACs (bacterial artificial chromosomes) transgenic mice ADAM12-GFP (*Cre-Ires-GFP*) and ADAM12-DTR (*tTA-Ires-DTR*)[27]. To perform inducible lineage tracing of ADAM12+ cells, we crossed ADAM12-tTA mice with tetracycline-controlled Cre (LC-1 mice[74]) and Rosa26[floxSTOP-YFP] reporter mice (obtained from The Jackson Laboratory) to obtain ADAM12-tTA-Cre[YFP] triple transgenic mice. TGFBR2[flox] mice[75] were obtained from The Jackson Laboratory. All mice were housed in specific-pathogen-free conditions (14 h light/10 h dark, 20–24 °C, 50% humidity). Mice experiments were approved by the committee on animal experimentation of the Institut Pasteur and by the 'Ministère de l'Education Nationale, de l'Enseignement Supérieur et de la Recherche.'

### Tumor models

The B16-OVA (MO5) melanoma cell line (B16 cell line containing the ovalbumin gene[76]) was provided by C. Leclerc (Institut Pasteur). For melanoma studies, $5 \times 10^5$ MO5 cells were implanted subcutaneously in 8–12-week-old transgenic mice and sex-matched non-transgenic littermates, in the C57Bl/6 background, and tumor volume was assessed every 2–4 d. Caliper measurements were used to assess tumor volumes, using the formula (L × W²) / 2 (L, length; W, width). Tumors did not exceed the maximum size permitted by the animal experimentation committee (20 mm). In TRAMP mice, the rat promoter *Probasin* drives expression of SV40 large T antigen, leading to progressive forms of prostate cancer, from intraepithelial hyperplasia to carcinoma with distant-site metastasis[77]. In Rip-Tag2 mice, the rat *Insulin* promoter drives expression of SV40 large T antigen, leading to pancreatic β-cell tumors[78].

### Mice treatment

Depletion and lineage tracing of ADAM12+ cells was performed as we previously described[27]. Briefly, we injected ADAM12-DTR mice intraperitoneally (i.p.) with 100 ng of diptheria toxin (DT) every day for the indicated time to deplete ADAM12+ cells. To stop lineage tracing of ADAM12+ cells, we treated ADAM12-tTA-Cre[YFP] mice with doxycycline (Sigma-Aldrich) at 1 mg ml$^{-1}$ in drinking water containing 5% sucrose. The CAIX inhibitor U-104 (38 mg kg$^{-1}$, Selleck) was injected i.p. every 2 d starting at day 10 following MO5 inoculation. To assess proliferation, tumor-bearing mice were injected i.p. with 80 μl of 10 mM EdU and analyzed 24 h after injection. To inhibit CD8+ T cells, mice were injected i.p. at day 6 and 10 after tumor inoculation with anti-CD8 antibodies (200 μg per mouse, Biolegend no. 100746) or control IgG (Biolegend no. 400544). For analysis of tumor perfusion, the fluorescent stain Hoechst 33342 (H33342) was injected intravenously (i.v.) 1 min before euthanization. To activate AXL, mice were injected i.p. at day 9 and 13 after tumor inoculation with anti-AXL activating antibodies (1 mg kg$^{-1}$, AF854; R&D Systems) or control IgG (AB-108-C; R&D Systems), as previously described[59,79].

### Immunofluorescence

Tissues were processed and stained as previously described[27]. Briefly, 8-μm sections from OCT-frozen tumors were incubated in 10% bovine serum (BS) in PBS containing 0.1% Triton X-100 (PBS-TS), followed by incubation with primary antibodies in PBS containing 1% PBS-TS overnight at 4 °C, washed and incubated for 1 h at 20 °C with secondary antibodies or streptavidin, washed, incubated with DAPI (1 μg ml$^{-1}$) and mounted with Fluoromount-G (Southern Biotechnology Associates). Expression of *Adam12* on tumor sections was detected with an RNAscope ISH assay (Advanced Cell Diagnostics), following the manufacturer's instructions. We examined slides with an AxioImager M1 fluorescence microscope (Zeiss) equipped with a CCD camera and processed images with AxioVision Zen software (Zeiss) or ImageJ software.

### Antibodies

A full list of antibodies used in this study is provided in Supplementary Table 1. We used the following dilutions: anti-GFP polyclonal (rabbit 1:1,000, chicken 1:1,500), anti-CD11b (1:400), anti-MHCII (1:400), anti-NK1.1 (1:200), anti-Foxp3 (1:200), anti-collagen-I polyclonal (1:1,000), anti-collagen-IV (1:1,000), anti-αSMA (1:500), anti-NG2 (1:200), anti-CD31 (1:200), anti-CD3 (1:200), anti-CD4 (1:200), anti-CD45.2 (1:200), anti-Ly6C (1:200), anti-SiglecF (1:500), anti-PDGFRα (1:100), anti-PDGFRβ (1:100), anti-CD8b (1:200), anti-F4/80 (1:200), anti-IFN-γ (1:200), isotype control IgG1 (1:200), anti-ICAM1 (1:400), anti-Ly6G (1:400), anti-CD11C (1:200), anti-CD206 (1:400), streptavidins (1:500), anti-PDPN (1:400), secondary antibodies (1:500), anti-AXL (1:1,000), goat IgG control (1:1,000), anti-cleaved-Caspase3 (1:1,000), anti-P-AXL (1:50), anti-CD26 (1:400), anti-FDC (1:200), anti-MadCAM-1 (1:200), anti-CD34 (1:50), and anti-Ki-67 (1:200).

### Cells isolation and FACS

Tumors were cut into small pieces and processed in a solution composed of DMEM (Gibco), Liberase TL (0.26 Wunsch units mL$^{-1}$; Roche) and DNase I (1 U mL$^{-1}$; Thermo Fisher) for 30 min, with manual dissociation by pipetting every 10 min. Cells were filtered through a 100-μm and a 40-μm mesh, washed and processed for cell staining as previously described[27]. Briefly, we first incubated cells with monoclonal antibody 2.4G2 to block Fcγ receptors, and then with the indicated antibodies in PBS containing 2% bovine serum (PBS-BS), followed by appropriate secondary antibodies or streptavidin when necessary. Cells were incubated with DAPI (Sigma) before analysis to exclude dead cells. For FACS analysis of immune cells, tumors were analyzed at day 14–15 except for T$_{reg}$ cells (days 17–18). For intracellular IFN-γ staining, cells were stimulated in vitro for 4 h with PMA and ionomycin, or with SIINFEKL peptide (OVA), and for 2 h with Brefeldin A. Cells were incubated with the LIVE/DEAD Cell Stain Kit to exclude dead cells (Invitrogen). Cells were processed for intracellular staining using CytoFix/CytoPerm Buffer Kit (Invitrogen), according to the manufacturer's instructions. Cells were analyzed with Fortessa (BD Biosciences) and Flowjo software (Tristar) and sorted with FACS ARIA III (BD Biosciences). For cell sorting of stromal cells, dead cells d doublet cells, hematopoietic (CD45+) and endothelial (CD31+) cells were systematically gated out before selecting for positive stromal cell markers. Tumor macrophages were sorted as CD45+CD3−CD19−Ly6G−SiglecF−CD11b+F480+. Stromal cells of the LN were gated as CD45−CD31− PDPN+ (FRC), which were further selected for expression of MAdCAM-1 (MRC) or FDC-M1 (FDC). TRC were defined as CD45−CD31− PDPN+MAdCAM-1−FDC-M1−.

### Efferocytosis assay

To induce apoptosis, MO5 melanoma cells were serum starved for 2 h, irradiated with UV (UVP Crosslinker at 100 mJ cm$^{-2}$) for 10 min and incubated overnight in complete medium. Apoptotic MO5 cells were incubated with Amine-Reactive pHrodo Dyes (Thermo Fisher) at a concentration of 20 ng mL$^{-1}$ for 30 min at 20 °C in the dark. For the efferocytosis assay, BMDMs were mixed with pHrodo-stained apoptotic MO5 cells for 30 min at 37 °C, at a 1:1 ratio, in the presence of conditioned medium from GFP+ or GFP− cells, stained with antibodies anti-F4/80 and analyzed with Fortessa (BD Biosciences) and Flowjo software (Tristar). The fold changes on the *y* axis were calculated using the percentage of efferocytosis in control medium as the baseline.

### RNA Isolation and qRT–PCR

For total tissue RNA extraction, we used the PureLink RNA Mini Kit (Invitrogen), according to the manufacturer's instructions. To extract RNA from cells, we used FACS to sort cells, or collected cells from culture plates, and placed them directly into vials containing lysis buffer. We isolated RNA using the Single Cell RNA Purification Kit (Norgen), according to the manufacturer's instructions. We assessed the quality

of total RNA using the 2100 Bioanalyzer system (Agilent Technologies) and the quantity using the Qbit RNA HS kit (Thermo Fisher). Total RNA was transcribed into complementary DNAs using SuperScript IV Reverse Transcriptase (Invitrogen). We performed qRT–PCR using RT$^2$ qPCR primer sets (SABiosciences and Bio-Rad Laboratories, a list is provided in Supplementary Table 2) and SYBR-Green master mix (Bio-Rad Laboratories), on a PTC-200 thermocycler equipped with a Chromo4 detector (Bio-Rad Laboratories), and analyzed data using Opticon Monitor software (Bio-Rad Laboratories). Ct values were normalized to the mean of the Ct values obtained for the housekeeping genes *Hsp90ab1*, *Hprt* and *Gapdh*.

## RNA sequencing

Libraries were prepared from total mRNA using the SMARTer Stranded Total RNA-Seq Kit v2-Pico Input Mammalian (Takara Bio), according to the manufacturer's instructions. Library quality and quantity were assessed using the 2100 Bioanalyzer system (Agilent Technologies) and the Qbit dsDNA HS kit (Thermo Fisher). Sequencing was performed using Illumina NextSeq 500/550 High Output kit v2.5. The RNA-seq analysis was performed with Sequana 0.11.0 (https://github.com/sequana/sequana_rnaseq) built on top of Snakemake 6.1.1 (ref. [80]). Briefly, reads were trimmed from adapters using cutadapt 3.4, then mapped to the genome assembly GRCm38 from Ensembl using STAR 2.7.3a. Feature-Counts 2.0.1 was used to produce the count matrix, assigning reads to features using corresponding annotation GRCm38_92 from Ensembl with strand-specificity information. Quality-control statistics were summarized using MultiQC 1.10.1 (ref. [81]). Clustering of transcriptomic profiles was assessed using principal components analysis. Differential expression testing was conducted using DESeq2 library 1.22.2 (ref. [82]). The normalization and dispersion estimation were performed with DESeq2 using the default parameters; statistical tests for differential expression were performed by applying the independent filtering algorithm. A generalized linear model was set in order to test for the differential expression between conditions. Raw *P* values were adjusted for multiple testing according to the Benjamini–Hochberg procedure, and genes with an adjusted *P* lower than 0.05 were considered differentially expressed. Gene set enrichment analysis was performed using Fisher's exact test for the over-representation of upregulated genes.

## Human tumors

We selected non-metastatic samples from three TCGA projects (RNA-seq analyses): skin cutaneous melanoma (SKCM), colon adenocarcinoma (COAD) and prostate adenocarcinoma (PRAD), all three in RKPM (Reads Per Kilobase Million) log$_2$ + 1 transformed. We also selected one International Cancer Genome Consortium (ICGC) project, pancreatic ductal adenocarcinoma (PACA_AU, array-based analysis), retrieved from Bailey et al.[83]. We filtered out genes that had missing values in any of the samples or that were located in the Y chromosome locus. Each dataset was analyzed independently. For each dataset, ADAM12$^{hi}$ and ADAM12$^{lo}$ groups were defined according to their ADAM12 expression, either above or below median ADAM12 expression, respectively. Gene set enrichment analyses (GSEAs) were carried out using the fgsea R package. A Wilcoxon test was used to analyze mean differences in gene expression between the ADAM12$^{hi}$ and ADAM12$^{lo}$ groups, and the Wilcoxon test value was used to make the gene rankings for GSEA. GSEA parameters were were: fgseaMultilevel(pathways = ('h.all.v7.0.symbols.gmt', 'c5.go.mf. v7.2.symbols.gmt', 'c2.cp.kegg.v7.1.symbols.gmt', 'c5.bp.v7.1.symbols. gmt'), stats = 'Gene rank', minSize = 5, maxSize = 500). GSEA results were displayed using the plotEnrichment function with default parameters from the fgsea package for GSEA curves. Genes accounting for a pathway being defined as enriched are refered to as 'leading edges'. These genes were shared by all four datasets. Expression levels of the leading-edge genes in our samples were displayed as a heat map using the ComplexHeatmap R package.

## Single-cell RNA-seq

Analysis of *ADAM12* expression in single-cell RNA-seq was performed on previously published data[84–86], and results were visualized using the Broad Institute's Single Cell Portal (https://singlecell.broadinstitute.org/single_cell). For pancreatic ductal adenocarcinoma tumors, raw data from Peng et al.[86] were analyzed using the Seurat (v.3.2) R package, with all functions ran with default parameters. Low-quality cells (<200 genes per cell, <3 cells per gene and >5% mitochondrial genes) were excluded from further steps. Cell-type identification was done using the SCINA R package, and gene markers from the MCP counter were used for stromal cells[87]. *KRT19*, *CDH1*, *MUC1*, *SOX9* and *EPCAM* were used as marker genes for epithelial cells. Nonlinear dimensional reduction (*t*-SNE) was applied as described in ref. [86]. The 'Activated stroma' signature score, as described by Puleo et al.[65], was calculated for each cell using the default function AddModuleScore from Seurat.

## Treatments and cell culture

MO5 melanoma cells were grown in RPMI medium supplemented with 10% FBS, 1% penicillin–streptomycin, G418 (2 mg mL$^{-1}$) and hygromycin B (0.06 mg mL$^{-1}$). To obtain CM, cells were isolated by FACS and seeded into 96-well flat-bottom plates in DMEM 10% FBS at 37 °C. We collected CM after 24–48 h for further experiments. For in vitro polarization of BMDMs, bone marrow cells were isolated and differentiated as previously described[88], then incubated for 24 h in the indicated conditions. When indicated, BMDMs with specific deletion of AXL in myeloid cells were generated by crossing Axl$^{fl/fl}$ mice[89] with LysMcre mice[90]. Axl$^{fl/fl}$ littermates were used as controls. For stimulation in vitro, OSM (5 ng mL$^{-1}$), IL-1β (10 ng mL$^{-1}$) or TGF-β (2 ng mL$^{-1}$) were added on cells for the indicated times. Cell proliferation was measured with Cell Counting Kit-8 (CCK-8; Dojindo, no. 899650), according to the manufacturer's instructions. To induce hypoxia, stromal cells were cultured in a hypoxic chamber (Whitley H35 Hypoxystation) in 1% oxygen for 72 h. The engulfing ability of macrophages was assessed by isolating the cells from tumors and incubating them for 30 min with Fluoresbrite Red microspheres (no. 18660–5, Polyscience) at 37 °C (or 4 °C for control condition), staining them with antibodies against F4/80 and analyzing them with Fortessa (BD Biosciences) and FlowJo software (Tristar). Gas6 was measured by ELISA, according to the manufacturer's instructions (Mouse Gas6 DuoSet ELISA, R&D, no. DY986). The tumor extracellular pH was assessed using pH Microelectrodes (Fisher Scientific), following the manufacturer's instructions.

## Image quantification

The immunofluorescence signal was quantified using ImageJ and threshold processing on high-resolution tiles that were stiched to create mosaic images of the entire tumor section or on tumor fields, obtained with an AxioImager M1 fluorescence microscope (Zeiss) and ZEN software. Tumor hypoxia was detected using the Hypoxyprobe Plus Kit, following the manufacturer's instructions, and the proportion of pimonidazole signal was measured by pixel quantification on mosaic images. The percentage of tumor perfused area was determined by normalizing Hoechst-positive area to the DAPI-positive area on mosaic images. Pericyte coverage was determined by measuring NG2/CD31 staining ratio by quantifying pixels on mosaic images of the total tumor section (all vessels in mosaic images were measured). Blood vessel width was measured using the Adobe Photoshop measurement tool. The proportion of AXL-P$^+$ and cleaved-Casp3$^+$ cells was measured by pixel quantification, and was normalized to F480- or DAPI-positive area, respectively. The tumor margin was identified as the peritumoral zone with a high density of PDPN$^+$ or αSMA$^+$ stromal cells.

## Statistical analysis

We determined statistical significance using two-tailed unpaired Student's *t*-test (with Welch's correction for unequal s.d.) between two

groups, Mann–Whitney *U* test if data were not normally distributed, and one- or two-way ANOVA or the Kruskal–Wallis test across multiple groups, as indicated in the legends. For one- or two-way ANOVA, we performed the Tukey or Sidak multiple-comparison correction, as determined on GraphPad Prism 9. When indicated, we compared conditions using a global Kruskal–Wallis test and then performed pairwise comparisons using a Wilcoxon test. *P* values from Wilcoxon tests were adjusted for multiple testing using the Bonferroni method. Unless otherwise specified, *n* represents the number of mice. No statistical method was used to predetermine sample size, which was chosen on the basis of prior experience and prior published studies with similar layout. No technical replicates across independent experiments were pooled in the datasets. Investigators were blind to the conditions of the experiments during data collection, and no animals or data points were excluded. Age- and sex-matched non-transgenic littermate mice were used as controls, and mice were randomly assigned in each group. Values are expressed as mean ± s.d. *P* < 0.05 was considered statistically significant.

## Material availability
This study did not generate new unique reagents. Further information and requests for resources and reagents should be directed to the corresponding author, L.P.

## Reporting summary
Further information on research design is available in the Nature Portfolio Reporting Summary linked to this article.

## Data availability
The TCGA_PAAD, TCGA_SKCM, TCGA_COAD and TCGA_PRAD dataset can be accessed at https://portal.gdc.cancer.gov/. The ICGC_PAN_AU dataset can be accessed at: https://dcc.icgc.org/. scRNA-seq data of pancreatic cancer from Peng et al.[86] are available from the Genome Sequence Archive (project PRJCA001063, https://ngdc.cncb.ac.cn/bioproject/browse/PRJCA001063). scRNA-seq data from CRC[84] and SKCM[85] are accessible via the Gene Expression Omnibus (GEO) under accession codes GSE178341 and GSE115978, respectively. The genome GRCm38 (GCA_000001635.9) release 102 is accessible from Ensembl (http://www.ensembl.org/Mus_musculus/Info/Index). RNA-seq data generated in this study are deposited in NCBI's Gene Expression Omnibus under accession codes GSE206794 (Depletion of ADAM12+ cells in MO5 tumors) and GSE206795 (ADAM12+ cells in MO5 tumors). Source data are provided with this paper.

## Code availability
This paper does not report original code.

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

## Acknowledgements
We thank the Cytometry Platform at the Institut Pasteur; the Biomics Olatform (C2RT, Institut Pasteur; supported by France Génomique (ANR-10-INBS-09) and IBISA) and the animal facility. We thank B. Charbit, G. Faucher and A. Lepelletier for excellent technical assistance, and C. Leclerc for the MO5 cell line. S.D.C. was funded by La Ligue contre le Cancer and Institut Pasteur. A.O. was funded by a doctoral fellowship from Le Ministère de l'Enseignement Supérieur et de la Recherche. The lab has received funding from the European Research Council (ERC) Consolidator grant 648428-PERIF (to L.P.), INSERM (to L.P.), l'Institut National du Cancer (INCa #PLBIO21-264) (to L.P.), and Institut Pasteur (to L.P.).

## Author contributions
Conceptualization, methodology, and validation, S.D.C. and L.P.; experiments and data analysis, S.D.C., D.C.G., A.O., M.S., L.P.; RNA-seq analysis, H.V. and R.L.; Bioinformatic analysis of human tumors, J.R. and C.B.; providing material, J.T.; writing—original draft, S.D.C. and L.P.; funding acquisition, L.P.; supervision, L.P.

## Competing interests
The authors declare no competing interests.

## Additional information
**Extended data** is available for this paper at https://doi.org/10.1038/s41590-023-01642-7.

**Correspondence and requests for materials** should be addressed to Lucie Peduto.

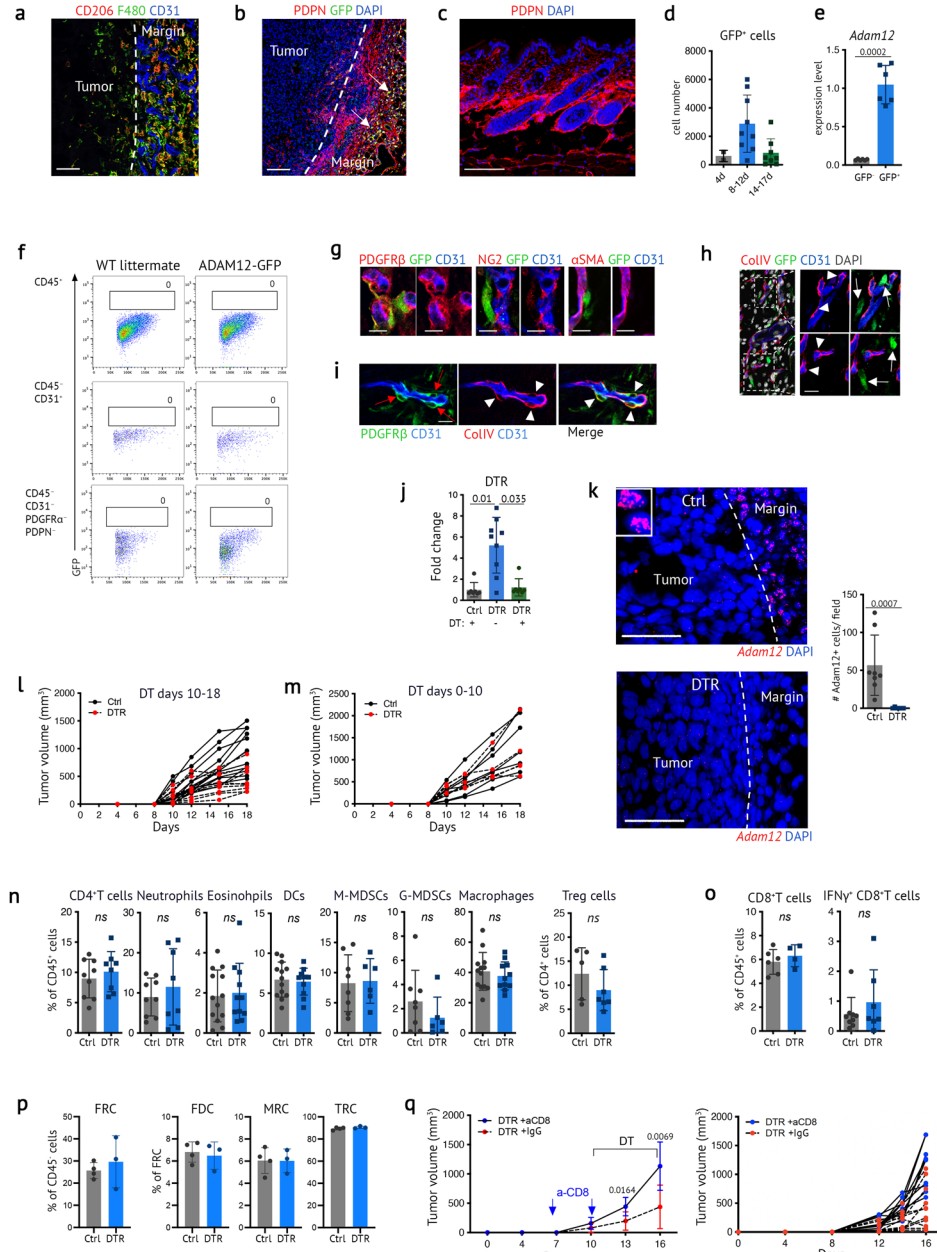

**Extended Data Fig. 1 | Depletion of ADAM12⁺ MSCs restores tumor immunity.**
(**a**) Immunofluorescence staining of the indicated markers in MO5 tumors. Scale bar, 50 μm. (**b**) Immunofluorescence staining of PDPN and GFP in MO5 tumors growing in GFP⁺ mice, as in Fig. 1b. Scale bar, 100 μm. (**c**) Immunofluorescence staining of PDPN⁺ cells in normal skin. Scale bar, 100 μm. (**d**) Absolute numbers of GFP⁺ cells at the indicated days after tumor inoculation, as in Fig. 1c. (**e**) Expression of *Adam12*, measured by qRT-PCR, in the indicated populations isolated by FACS from MO5 tumors; *n* = 6 from 2 independent experiments. (**f**) FACS plot and percentages of GFP⁺ cells in the indicated populations from Fig. 1c. (**g-i**) Immunofluorescence staining of the indicated markers in MO5 tumors growing in GFP⁺ mice (g,h) or GFP⁻ littermates (i). White arrows indicate GFP⁺ cells; red arrows indicate pericytes; white arrowheads indicate the vBM. Scale bar, 10 μm. (**j**) Expression of human *DTR (HBEGF)* measured by qRT-PCR in the indicated conditions; *n* = 7, except for DTR(*n* = 9). (**k**) Expression of *Adam12* detected by RNAscope in tumor sections from mice treated as in Fig. 1d; *n* = 6(DTR)-8(Ctrl) fields from 2 independent experiments. Scale bar, 50 μm. (**l,m**) Individual animal growth curves from Fig. 1d (l) and Fig. 1e (m).

(**n**) Percentage of tumor infiltrating CD4⁺ T cells (*n* = 9 Ctrl, *n* = 8 DTR), neutrophils (*n* = 9 Ctrl, *n* = 8 DTR), eosinophils (*n* = 13 Ctrl, *n* = 11 DTR), DCs (*n* = 13 Ctrl, *n* = 11 DTR), M-MDSCs (*n* = 8 Ctrl, *n* = 6 DTR), G-MDSCs (*n* = 8 Ctrl, *n* = 6 DTR), macrophages (*n* = 13 Ctrl, *n* = 11 DTR), Tregs (*n* = 5 Ctrl, *n* = 7 DTR), measured by FACS. (**o,p**) Percentage of CD8⁺ T cells (*n* = 6 Ctrl, *n* = 4 DTR), IFNγ⁺ CD8⁺cells (*n* = 9 Ctrl, *n* = 7 DTR) (o) and the indicated fibroblastic populations (*n* = 4 Ctrl, *n* = 3 DTR) (p) in the tumor draining lymph node (LN), as measured by FACS. (**q**) Tumor growth curves from the indicated mice (*n* = 11, DTR + IgG; *n* = 9, DTR + anti-CD8), from 2 independent experiments. Left: average tumor volume and Right: individual animal growth curve. In l,m,q, the x axis represents days after tumor inoculation. In a-c,g-i one representative image of at least 3 independent experiments is shown. FRC: fibroblastic reticular cells; FDC: follicular dendritic cells; MRC: marginal reticular cells; TRC: T-zone FRCs. Statistics were calculated using two-tailed, unpaired Student's *t*-test (e,n,o), Kruskal-Wallis test (j), two-tailed Mann-Withey test (k) or two-way ANOVA (q). All quantitative data are presented as mean values +/− SD.

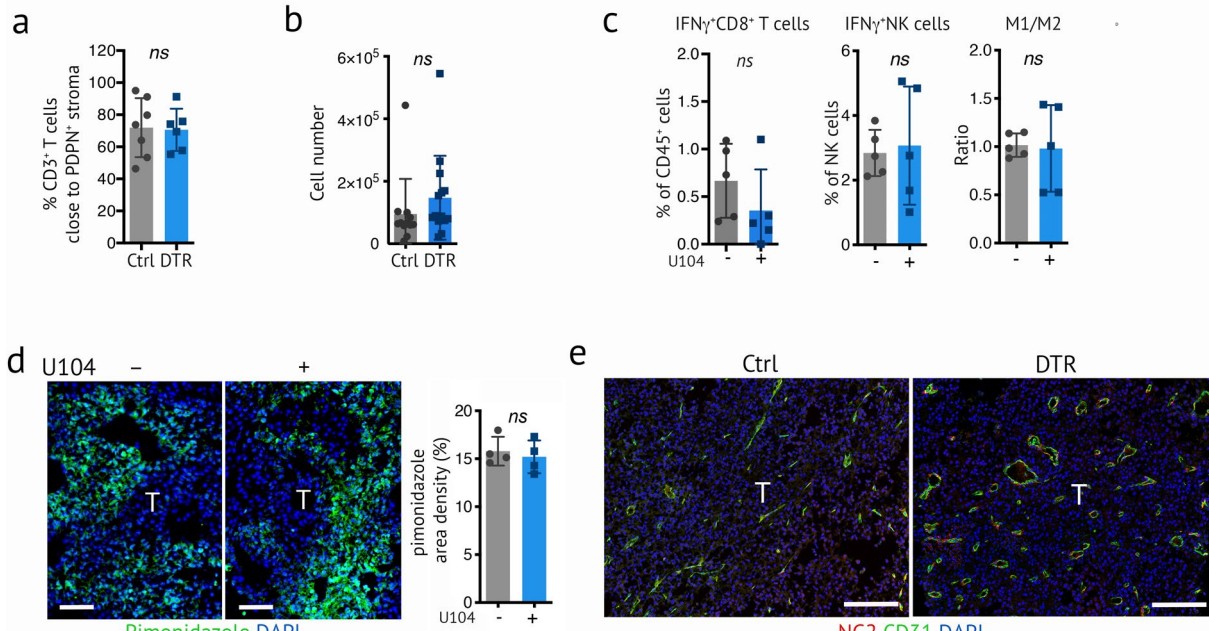

**Extended Data Fig. 2 | Immune and vascular remodeling of the TME.**
(**a**) Percentage of CD3[+] T cells in proximity to PDPN[+] stromal cells in mice treated as in Fig. 2a. Measurements were assessed in 3 fields/tumor from independent experiments ($n$ = 3). (**b**) Absolute numbers of PDPN[+] PDGFRα[+] cells (gated CD45[−]CD31[−]), measured by FACS, in MO5 tumors growing in mice treated as in Fig. 2a; $n$ = 12(Ctrl)−14(DTR) from 3 independent experiments. (**c**) Percentage of the indicated immune populations, measured by FACS, infiltrating MO5 tumors in WT mice treated with U-104 inhibitor; from independent experiments ($n$ = 5).

(**d**) Hypoxic zones (identified by pimonidazole staining) in frozen sections of MO5 tumors treated as in **c** ($n$ = 4); scale bar, 100 μm. (**e**) Immunofluorescence staining of NG2 and CD31 in frozen sections of MO5 tumors from mice treated as in Fig. 2i. Scale bar, 200 μm. T: tumor. In d,e, one representative image of independent experiments is shown. Statistics were calculated using two-tailed, unpaired Student's $t$-test. Data are presented as mean values +/− SD. ns: not significant.

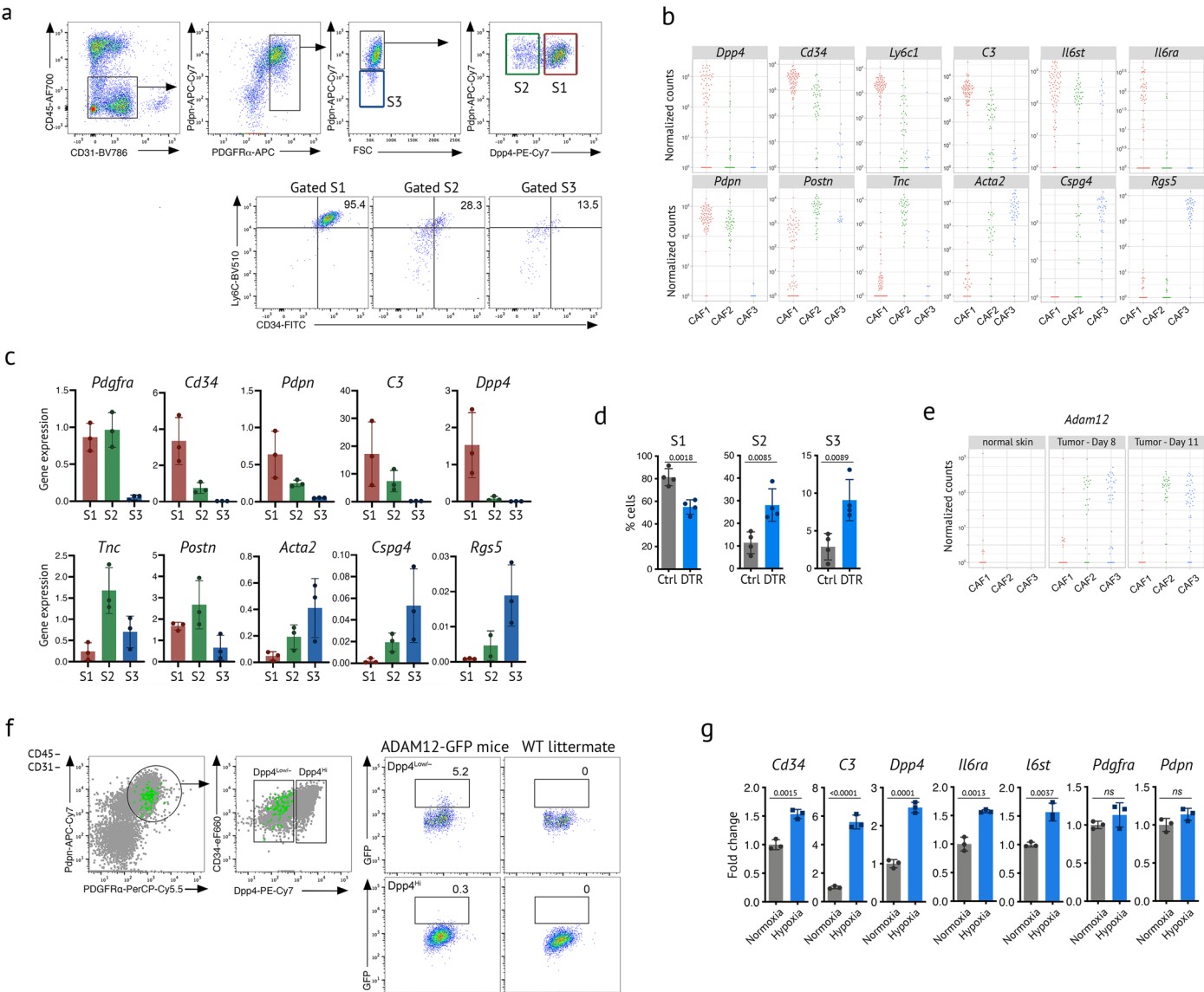

**Extended Data Fig. 3 | Stromal remodeling of the TME. (a)** FACS gating strategy for S1, S2 and S3 stromal populations from MO5 tumors. **(b)** Normalized expression of the indicated genes in stromal populations from melanomas single cells RNAseq[9]. **(c)** Expression of the indicated genes, measured by qRT-PCR, in S1, S2 and S3 isolated from MO5 tumors as shown in a (*n* = 3). **(d)** Percentage of the indicated stromal populations, measured by FACS, in MO5 tumors from ADAM12-DTR⁺ (DTR) and DTR⁻ littermates (Ctrl) treated with DT as in Fig. 2a (*n* = 4). **(e)** Normalized expression of *Adam12* from single cells RNAseq of melanomas and normal skin[9]. **(f)** FACS plot and percentages of GFP⁺ cells in the total stromal fraction from MO5 tumors. One representative experiment from independent experiments is shown (*n* = 7). **(g)** Expression of the indicated genes, measured by qRT-PCR, in stromal cells treated as indicated; *n* = 3. Gating strategy for stromal populations S1: PDGFRα^High PDPN^High CD34^High DPP4^High Ly6c^High; S2: PDGFRα⁺ PDPN⁺ CD34^Mid/Low DPP4^Low/−; S3: PDGFRα^Low PDPN^Low CD34^Low DPP4⁻. Statistics were calculated using two-tailed, unpaired Student's *t*-test (d,g). Data are presented as mean values +/− SD.

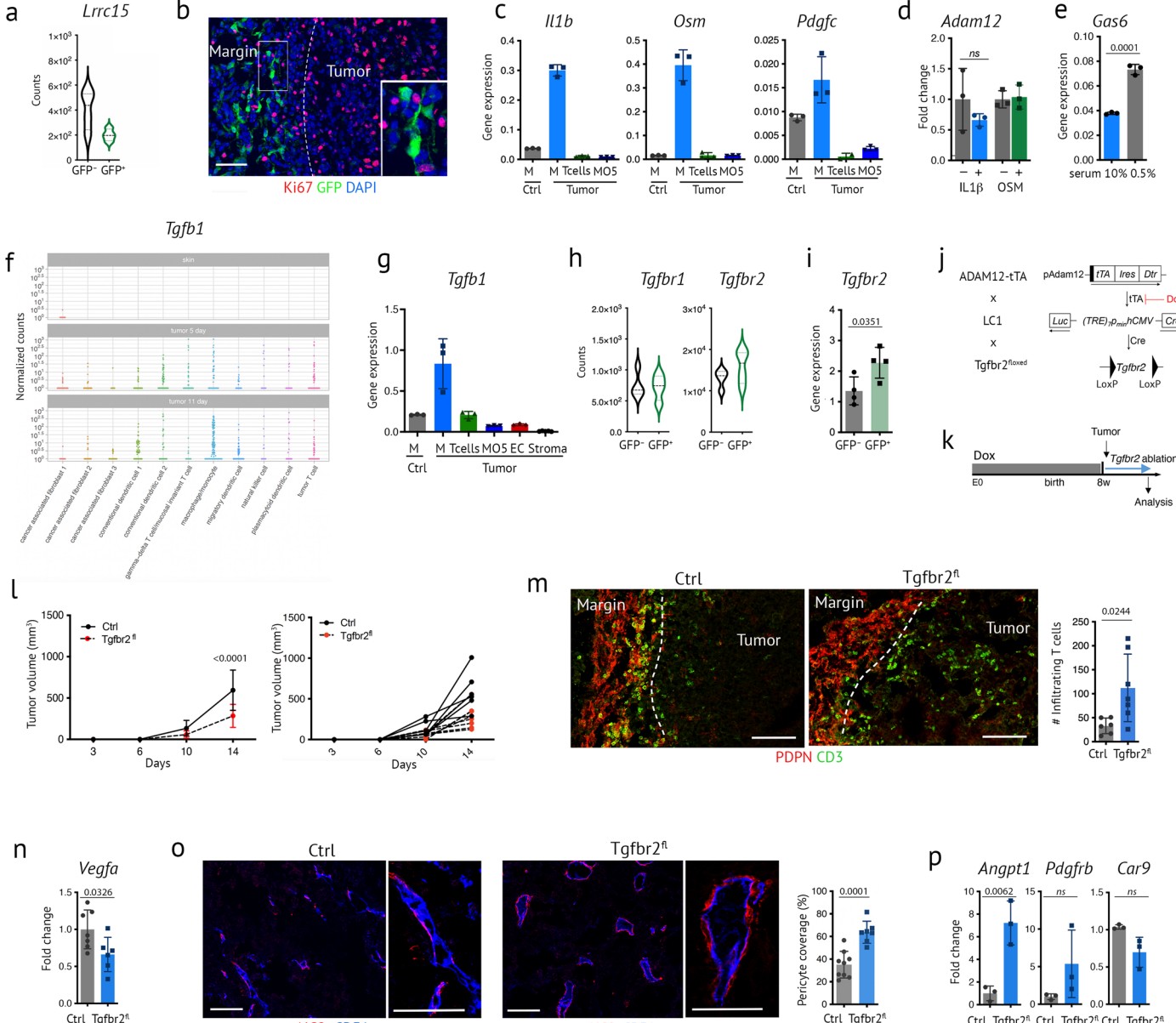

**Extended Data Fig. 4 | Immune-stroma crosstalk and Tgfbr2 signaling in ADAM12⁺ cells.** (**a**) Violin plot of *Lrrc15* from RNASeq data in Fig. 3a (*n* = 4). (**b**) Immunofluorescence staining of Ki67 and GFP in melanomas growing in ADAM12-GFP mice. Scale bar, 50 μm. (**c**) Expression of the indicated genes, measured by qRT-PCR, in the indicated populations isolated by FACS from MO5 tumors at day 12, *n* = 3 from independent experiments. Macrophages (M), tumor cells (MO5), endothelial cells (EC). (**d**) Expression of *Adam12*, measured by qRT-PCR, in ADAM12⁻ PDGFRα⁺ cells isolated from MO5 tumors, treated as indicated (*n* = 3). (**e**) Expression of *Gas6*, measured by qRT-PCR, in stromal cells cultured in indicated conditions (*n* = 3). (**f**) Normalized expression of *Tgfb1* in normal skin and melanomas (from single cells RNAseq dataset[12]). (**g**) Expression of the indicated genes, measured by qRT-PCR, as in **c**. *n* = 3 except *n* = 6 (stroma). (**h**) Violin plots of *Tgfbr1* and *Tgfbr2* from RNASeq data in Fig. 3a (*n* = 4). (**i**) Expression of *Tgfbr2*, measured by qRT-PCR, in GFP⁺ and GFP⁻ stromal cells isolated by FACS from MO5 tumors (*n* = 4). (**j**) Strategy for inducible depletion of *Tgfbr2* in ADAM12⁺ cells. (**k**) Experimental set up for l-p. (**l**) Tumor growth curves

from ADAM12-tTA2-Cre^Tgfbr2 mice (Tgfbr2^fl, *n* = 8) and littermate mice (Ctrl, *n* = 6) treated as indicated in k. Left, average tumor volume, and right, individual animal growth curves, from 2 independent experiments. The *x* axis represents days after tumor inoculation. (**m**) Immunofluorescence staining of PDPN and CD3 in tumor sections in the indicated conditions. Right, quantification of tumor infiltrating T cells was performed on *n* = 6(Ctrl)– 7(Tgfbr2^fl) fields. Scale bars, 100 μm. (**n**) Expression of *Vegfa*, measured by qRT-PCR, in macrophages isolated from tumors. *n* = 7(Ctrl)–6(Tgfbr2^fl) from 2 independent experiments. (**o**) Immunofluorescence staining of NG2 and CD31 in tumor sections from mice treated as in k; *n* = 7(Tgfbr2^fl)–9(Ctrl) fields. Scale bars, 100 μm. (**p**) Expression of the indicated transcripts, measured by qRT-PCR, in tumor stromal cells isolated by FACS. *n* = 3 from independent experiments. In b,e, one representative experiment out of 3 is shown. Statistics were calculated using two-tailed, unpaired Student's *t*-test (d,e,i,m-p) or two-way ANOVA (l). Data are presented as mean values +/– SD.

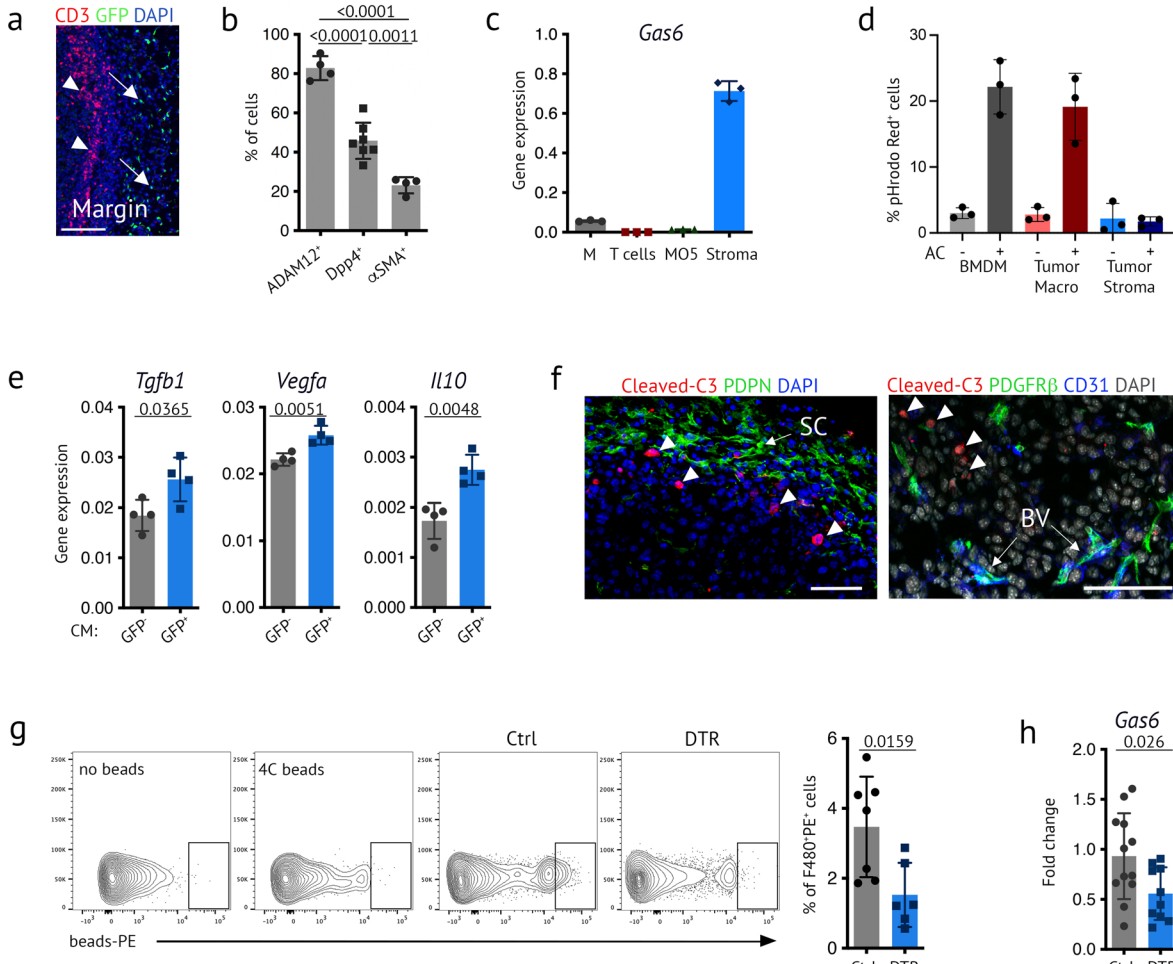

**Extended Data Fig. 5 | ADAM12⁺ cells promote efferocytosis.**
(**a**) Immunofluorescence staining of CD3⁺ T cells and ADAM12⁺ cells in MO5 tumors margin. Arrowheads indicate T cells infiltrated zones, and arrows indicate ADAM12⁺ cells. Scale bar, 100 µm. (**b**) Percentage of tumor stromal cells expressing ADAM12, DPP4 or αSMA in close proximity to AXL-P⁺ macrophages, as in Fig. 4b ($n = 4$, except for DPP4⁺ cells, $n = 7$), from 2 independent experiments. (**c**) Expression of *Gas6*, measured by qRT-PCR, in macrophages (M), T cells, tumor cells (MO5) and stromal cells isolated from MO5 tumors, $n = 3$. (**d**) Efferocytosis assay in BMDM, and macrophages or PDPN⁺PDGFRα⁺ stromal cells isolated from MO5 tumors ($n = 3$). AC= apoptotic cells (MO5). (**e**) Expression of the indicated genes, measured by qRT-PCR in BMDM following efferocytosis ($n = 4$).

(**f**) Immunofluorescence staining of the indicated markers in MO5 tumors depleted from ADAM12⁺ cells. Scale bar, 50 µm. Arrowheads indicate cleaved caspase 3⁺ apoptotic cells. BV= blood vessels; SC= stromal cells. (**g**) Phagocytic capacity of macrophages isolated by FACS from MO5 tumors growing in ADAM12-DTR or Ctrl littermate mice + DT, as in Fig. 4g; $n = 6$(DTR)-7(Ctrl) from two independent experiments. (**h**) Expression of *Gas6*, measured by qRT-PCR in MO5 tumors growing in mice treated as in Fig. 4g; $n = 10$(DTR)-12(Ctrl) from 3 independent experiments. In a,f, one representative image from 3 independent experiments is shown. Statistics were calculated using one-way ANOVA (b) or two-tailed, unpaired Student's *t*-test (e,g,h). All quantitative data are presented as mean values +/− SD.

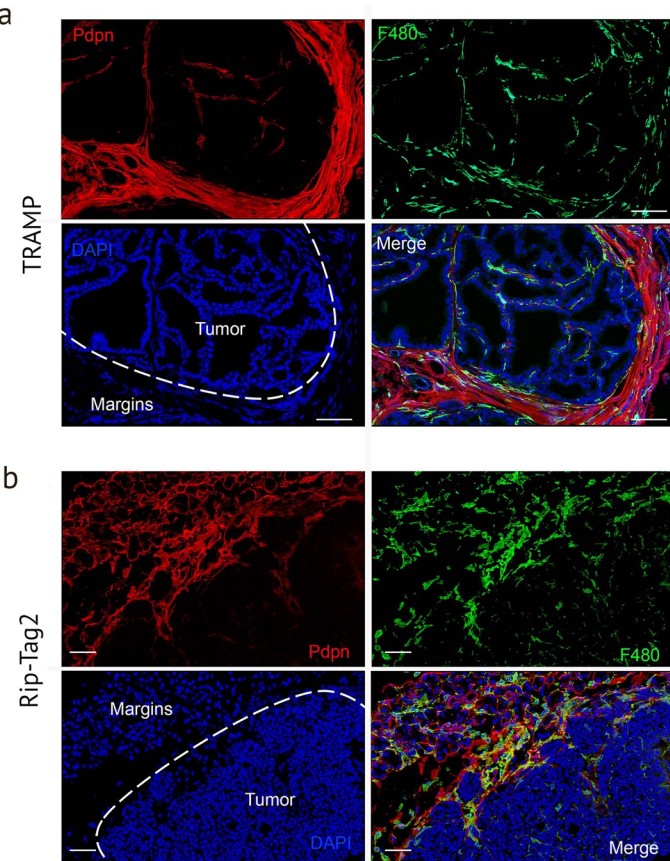

**Extended Data Fig. 6 | PDPN⁺ stroma and macrophages localization.** Immunofluorescence analysis of PDPN⁺ stromal cells (red) and F480⁺ macrophages (green) in frozen sections of tumors isolated from TRAMP mice **(a)** and RipTag2 mice **(b)**. One representative image from 3 independent experiments is shown. Scale bar, 50 μm. DAPI stains nuclei.

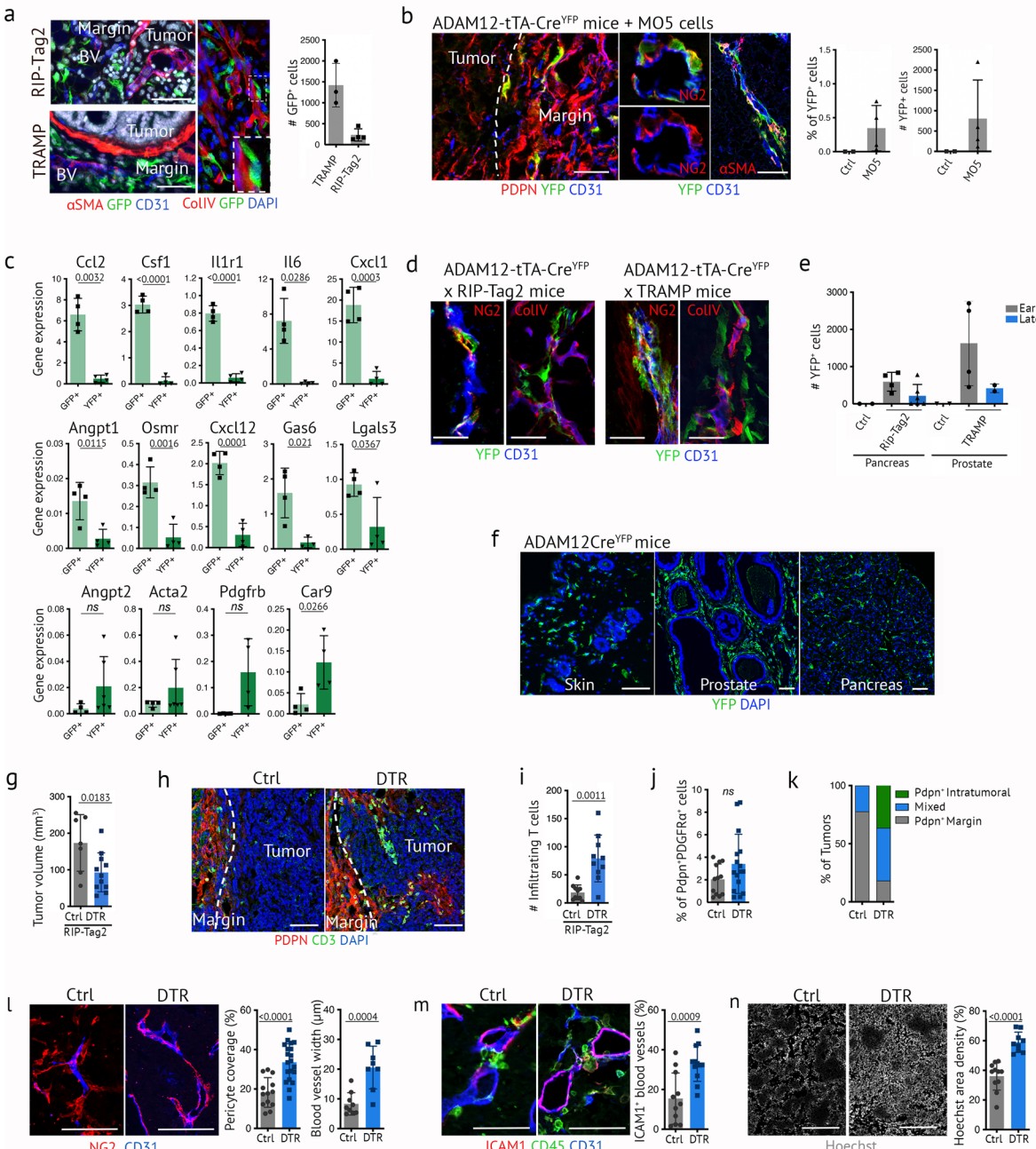

**Extended Data Fig. 7 | The progeny of ADAM12⁺ cells in spontaneous tumor models. (a)** Immunofluorescence staining of GFP and the indicated markers in RipTag (*n* = 4) and TRAMP (*n* = 3) tumors. Right, absolute numbers of GFP⁺ cells measured by FACS. **(b)** Immunofluorescence staining of YFP and the indicated markers in MO5 tumors from ADAM12-tTA-Cre^YFP mice. Right, frequency and absolute numbers of YFP⁺ cells measured by FACS in normal skin (*n* = 2) and tumor (*n* = 4). **(c)** Expression of the indicated transcripts, measured by qRT-PCR, in GFP⁺ and YFP⁺ cells isolated from MO5 tumors growing in ADAM12-GFP mice or ADAM12-tTA-Cre^YFP mice, respectively; *n* = 4, except for *Angpt2* and *Acta2* YFP⁺ (*n* = 6) from independent experiments. **(d)** Immunofluorescence staining of YFP and the indicated markers in prostate and pancreatic tumors from the indicated mice. **(e)** Absolute numbers of YFP⁺ cells measured by FACS on independent mice, as indicated in Fig. 5g. **(f)** Immunofluorescence staining of fetal YFP⁺ cells in adult non-tumoral skin, prostate and pancreas. **(g)** Volume of pancreatic tumors in RIPTag/DTR or RIPTag2 littermate mice (Ctrl) treated with DT from weeks 11 to 14; *n* = 7(Ctrl)–11(DTR) from independent experiments. **(h)** Immunofluorescence staining of PDPN and CD3 in tumor sections from mice

treated as in g. One representative image is shown of independent experiments (*n* = 4). **(i)** Absolute numbers of tumor infiltrating CD3⁺ T cells. Quantifications were performed on *n* = 10 images from 6 mice per group. **(j)** Percentage of PDPN⁺ PDGFRα⁺ cells, measured by FACS, in RIP-Tag2 tumors from mice treated as in g; *n* = 12(Ctrl)–15(DTR) from 6 independent experiments. **(k)** Distribution of PDPN⁺ cells within RIPTag tumors, in mice treated as in g; *n* = 9(Ctrl)-11(DTR) from 3 independent experiments. **(l)** Immunofluorescence staining of the indicated markers in tumor sections of mice treated as in g; quantification left, *n* = 13(Ctrl)–20(DTR) and right, *n* = 9(Ctrl)–8(DTR) fields. **(m)** Immunofluorescence staining of ICAM1, CD45 and CD31 in tumor sections of mice treated as in g; *n* = 11(Ctrl)-12(DTR) fields. **(n)** Tumor perfusion, as detected by Hoechst 33342 staining, in tumor sections from mice treated as in g; *n* = 9(DTR)-11(Ctrl) fields. In a,b,d,f, representative images of at least 3 independent experiments are shown. Statistics were calculated using two-tailed, unpaired Student's *t*-test, except for c (Mann–Whitney test). All quantitative data are presented as mean values +/– SD. Scale bars, 50 μm, 100 μm (f,h), and 200 μm (n).

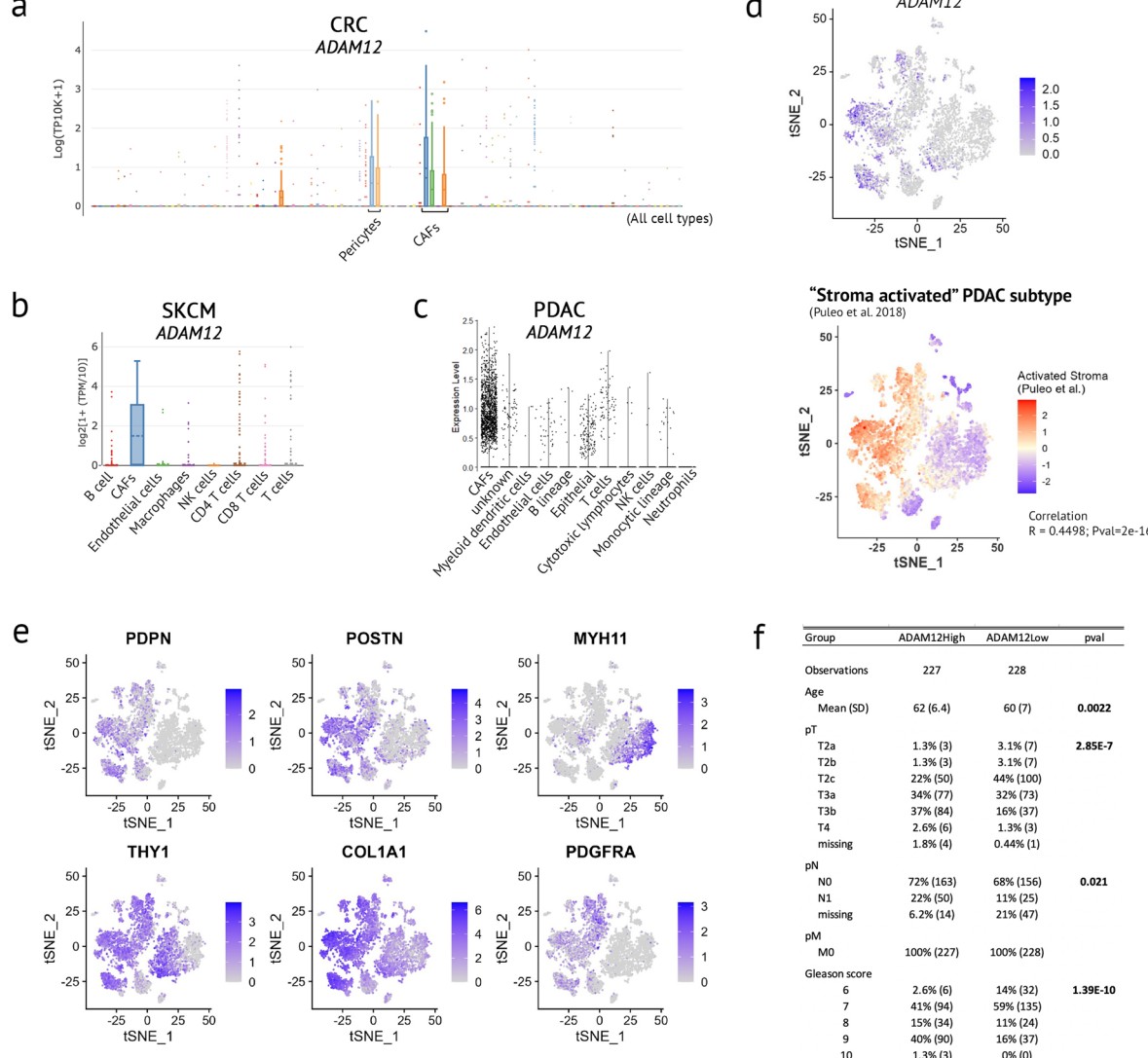

**Extended Data Fig. 8 | Characterization of ADAM12⁺ cells in human tumors.**
**(a)** Expression of *ADAM12* transcript in single-cell RNAseq from human colorectal cancer patients[84] (N = 62 patients, total 371,223 cells). Clustering = All cells (tSNE); On x axis, all cell subsets from the tumors are represented (88), and the major clusters expressing *ADAM12* are highlighted (annotation from Pelka et al.[84]; Pericyte_cS19, max: 3.314; q3:1.08525, median: 0; CAF_cS28, max:4.485, q3:1.50125, median: 0.558). **(b)** Expression of *ADAM12* transcripts in single-cell RNAseq from melanoma patients[85] (N = 31 melanoma tumors, total 39744 cells). Clustering = Non-malignant cells from melanoma tumors (CAF, max: 5.2823, q3: 3.0613, mean: 1.49153, median: 0). **(c)** Expression of *ADAM12* transcripts in single-cell RNAseq from human PDAC[86] (N = 24 primary untreated PDAC tumors, 41986 cells). Clustering= All cells (see Methods). **(d)** tSNE representation of *ADAM12* expression in human PDAC as described in Peng et al.[86] (top panel)

and transcriptional signature of activated stromal PDAC subtype as described in Puleo et al.[65] (lower panel). Two-sided Pearson's Correlation Coefficient (R) was measured between *ADAM12* expression and the activated stromal subtype signature (displayed in the tSNE panel). **(e)** Expression of the indicated transcripts in scRNAseq dataset from d. **(f)** Clinicopathologic characteristics of patients from the TCGA_PRAD project studied in Fig. 6. Patients (N = 455) were grouped according to their expression of *ADAM12*, as described in Methods. The associations were tested using two-sided Pearson chi2 test for categorical variables and the Mann–Whitney U test for continuous variables. Pathological classification: T – Extent of the primary tumor; N – Absence or presence and extent of regional lymph node metastases; M – Absence or presence of distant metastases. In a,b, data were visualized on the Single Cell portal of The Broad Institute of MIT and Harvard (https://singlecell.broadinstitute.org/single_cell).

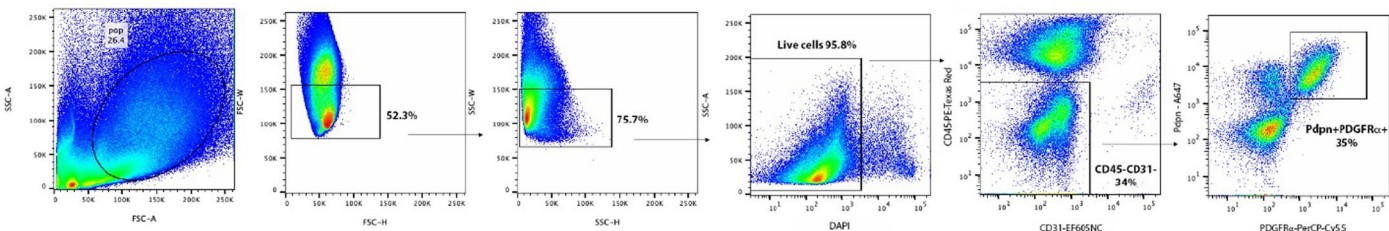

**Extended Data Fig. 9 | Gating strategy for stromal cells.** Flow cytometry gating strategy for PDPN⁺PDGFRα⁺ stromal cells in early stage MO5 tumor.

**Extended Data Table 1 | GFP+ cells in TRAMP and RIPTag tumors**

|      |         | RIPTAG | TRAMP |
|------|---------|--------|-------|
| UP   | *Adam12* | 2,96 | 8,00 |
|      | *Pdgfra* | 4,03 | 2,32 |
|      | *Osmr* | 3,02 | 2,44 |
|      | *Il6st* | 2,12 | 1,63 |
|      | *Il1r1* | 1,11 | 1,82 |
|      | *Csf1* | 3,00 | 3,57 |
|      | *Ccl2* | 2,22 | 3,86 |
|      | *Cxcl12* | 2,36 | 5,66 |
|      | *Has1* | 3,02 | 2,01 |
|      | *Has2* | 3,89 | 4,76 |
|      | *Icam1* | 4,65 | 1,48 |
|      | *Vcam1* | 1,85 | 1,59 |
|      | *Ly6a* | 2,96 | 1,26 |
|      | *Gadd45b* | 3,24 | 2,39 |
|      | *Cxcl1* | 3,52 | 3,98 |
|      | *Il6* | 3,07 | 4,10 |
|      | *Mmp3* | 6,21 | 3,23 |
|      | *Gas6* | 1,27 | 1,19 |
| DOWN | *Cdk1* | 0,79 | 0,43 |
|      | *Mki67* | 0,63 | n.d. in GFP+ |
|      | *Cdkn2a* | 0,72 | n.d. in GFP+ |

Expression of the indicated transcripts, measured by qRT-PCR, in GFP⁺ and GFP⁻ cells isolated from TRAMP and Rip-Tag2 tumors growing in ADAM12-GFP mice (Fold change GFP⁺ vs GFP⁻ cells), from independent experiments (n=3). N.d not detected.

# Reporting Summary

## Statistics

For all statistical analyses, confirm that the following items are present in the figure legend, table legend, main text, or Methods section.

| n/a | Confirmed | |
|---|---|---|
| ☐ | ☒ | The exact sample size (*n*) for each experimental group/condition, given as a discrete number and unit of measurement |
| ☐ | ☒ | A statement on whether measurements were taken from distinct samples or whether the same sample was measured repeatedly |
| ☐ | ☒ | The statistical test(s) used AND whether they are one- or two-sided <br> *Only common tests should be described solely by name; describe more complex techniques in the Methods section.* |
| ☒ | ☐ | A description of all covariates tested |
| ☐ | ☒ | A description of any assumptions or corrections, such as tests of normality and adjustment for multiple comparisons |
| ☐ | ☒ | A full description of the statistical parameters including central tendency (e.g. means) or other basic estimates (e.g. regression coefficient) AND variation (e.g. standard deviation) or associated estimates of uncertainty (e.g. confidence intervals) |
| ☐ | ☒ | For null hypothesis testing, the test statistic (e.g. *F*, *t*, *r*) with confidence intervals, effect sizes, degrees of freedom and *P* value noted <br> *Give P values as exact values whenever suitable.* |
| ☒ | ☐ | For Bayesian analysis, information on the choice of priors and Markov chain Monte Carlo settings |
| ☒ | ☐ | For hierarchical and complex designs, identification of the appropriate level for tests and full reporting of outcomes |
| ☒ | ☐ | Estimates of effect sizes (e.g. Cohen's *d*, Pearson's *r*), indicating how they were calculated |

*Our web collection on statistics for biologists contains articles on many of the points above.*

## Software and code

Policy information about availability of computer code

| Data collection | This study does not report original code. |
|---|---|
| Data analysis | Data acquisition and analysis was performed using the following softwares, as detailed in methods: FlowJo v.10; BD FACSDiva software; ZEN 2 (Zeiss); ImageJ v1.54f; GraphPad Prism 9; Bio-Rad CFX manager 3.1; 2100 Bioanalyzer Expert Software (Agilent); Sequana 0.11.0; Snakemake 6.1.1. |

For manuscripts utilizing custom algorithms or software that are central to the research but not yet described in published literature, software must be made available to editors and reviewers. We strongly encourage code deposition in a community repository (e.g. GitHub). See the Nature Portfolio guidelines for submitting code & software for further information.

## Data

Policy information about availability of data

All manuscripts must include a data availability statement. This statement should provide the following information, where applicable:

- Accession codes, unique identifiers, or web links for publicly available datasets
- A description of any restrictions on data availability
- For clinical datasets or third party data, please ensure that the statement adheres to our policy

The TCGA_PAAD, TCGA_SKCM, TCGA_COAD and TCGA_PRAD datset can be accessed at https://portal.gdc.cancer.gov/. The ICGC_PAN_AU dataset can be accessed at: https://dcc.icgc.org/. scRNAseq data of pancreatic cancer from Peng et al. (Cell Research 2019) are available from the Genome Sequence Archive (project

## Human research participants

Policy information about studies involving human research participants and Sex and Gender in Research.

| | |
|---|---|
| Reporting on sex and gender | N/A |
| Population characteristics | N/A |
| Recruitment | N/A |
| Ethics oversight | N/A |

Note that full information on the approval of the study protocol must also be provided in the manuscript.

# Field-specific reporting

Please select the one below that is the best fit for your research. If you are not sure, read the appropriate sections before making your selection.

☒ Life sciences　　☐ Behavioural & social sciences　　☐ Ecological, evolutionary & environmental sciences

For a reference copy of the document with all sections, see nature.com/documents/nr-reporting-summary-flat.pdf

# Life sciences study design

All studies must disclose on these points even when the disclosure is negative.

| | |
|---|---|
| Sample size | No statistical method was used to predetermine sample size. Sample size was chosen based on prior experience and prior published studies with similar layout. |
| Data exclusions | No data was excluded from the analysis. |
| Replication | The number of replicates for each experiment is indicated in figure legends. Within each experimental group the reproducibility was successful, although a degree of variability was detected due to inter- individual diversity. |
| Randomization | Age and sex-matched non-transgenic littermate mice were used as control for all mice experiments. Mice were randomly assigned in each group. |
| Blinding | The investigators were blinded to group allocation during data collection. |

# Reporting for specific materials, systems and methods

We require information from authors about some types of materials, experimental systems and methods used in many studies. Here, indicate whether each material, system or method listed is relevant to your study. If you are not sure if a list item applies to your research, read the appropriate section before selecting a response.

## Materials & experimental systems

| n/a | Involved in the study |
|---|---|
| ☐ | ☒ Antibodies |
| ☐ | ☒ Eukaryotic cell lines |
| ☒ | ☐ Palaeontology and archaeology |
| ☐ | ☒ Animals and other organisms |
| ☒ | ☐ Clinical data |
| ☒ | ☐ Dual use research of concern |

## Methods

| n/a | Involved in the study |
|---|---|
| ☒ | ☐ ChIP-seq |
| ☐ | ☒ Flow cytometry |
| ☒ | ☐ MRI-based neuroimaging |

# Antibodies

| | |
|---|---|
| Antibodies used | Purified anti-GFP polyclonal antibody, Invitrogen #A11122 |
| | Pe-cy7 anti-CD11b monoclonal antibody, clone M1/70, Invitrogen # 25-0112-82 |
| | FITC anti-MHCII monoclonal antibody, clone M5/114.15.2, Invitrogen #11-5321-82 |
| | APC-eF780 anti-NK1.1 monoclonal antibody, clone PK136, Invitrogen #47-5941-82 |
| | PerCP-Cy5.5 anti-Foxp3 monoclonal antibody, clone FJK-16s, Invitrogen #15-5773-82 |
| | Purified anti-Collagen I polyclonal antibody, Biorad # 2150-1410 |
| | Purified anti-Collagen IV polyclonal antibody, Biorad # 2150-1470 |
| | Purified anti-GFP polyclonal antibody, Abcam #ab13970 |
| | Cy3 anti-aSMA monoclonal antibody, clone 1A4, Sigma #C6198 |
| | Purified anti-NG2 polyclonal antibody, Millipore #AB5320 |
| | APC anti-CD31 monoclonal antibody, clone MEC13.3, BD Bioscience, #17-0311-82 |
| | BV785 anti-CD31 monoclonal antibody, clone 390, BD OptiBuild, # 740879 |
| | Purified anti-CD3 monoclonal antibody, clone 500A2, BD Bioscience #14-0033-85 |
| | Alexa Fluor 700 anti-CD4 monoclonal antibody, clone RM4-5, BD Bioscience # 557956 |
| | V500 anti-CD45.2 monoclonal antibody, clone 104, BD Horizon #562129 |
| | BV605 anti-Ly6C monoclonal antibody, clone AL-21, BD Horizon #563011 |
| | PE-CF594 anti-SiglecF monoclonal antibody, clone E50-2440, BD Horizon #562757 |
| | Biotin anti-PDGFRa monoclonal antibody, clone APA5, Thermo Fisher #13-1401-82 |
| | Biotin anti-PDGFRb monoclonal antibody, clone APB5, Thermo Fisher #13-1402-82 |
| | APC anti-PDGFRa monoclonal antibody, clone APA5, eBioscience #17-1401-81 |
| | eFluor 710 anti-CD8b monoclonal antibody, clone eBioH35-17.2, eBioscience #46-0083-82 |
| | APC anti-F4/80 monoclonal antibody, clone BM8, eBioscience #17-4801-82 |
| | eFluor 660 anti-IFN gamma monoclonal antibody, clone XMG1.2, eBioscience #50-7311-82 |
| | eFluor 660 Isotype control IgG1 monoclonal antibody, clone eBio299Arm, eBioscience #50-4888-80 |
| | PE anti-ICAM1 monoclonal antibody, clone 3E2, BDPharmigen # 5553253 |
| | PerCP-Cy5.5 anti-Ly6G monoclonal antibody, clone 1A8, BDPharmigen # 560602 |
| | BV711 anti-CD3 monoclonal antibody, clone 17A2, Biolegend #100349 |
| | BV785 anti-CD11c monoclonal antibody, clone N418, Biolegend #117335 |
| | PE anti-CD206 monoclonal antibody, clone C068C2, Biolegend #141706 |
| | BV711 Streptavidin, Biolegend # 405241 |
| | PerCP-Cy5.5 Streptavidin, eBioscience # 45-4317-82 |
| | Purified anti-mouse Pdpn antibody, gift from A. Farr (University of Washington, Seattle). |
| | Alexa Fluor 488 anti-chicken IgY, Thermo Fisher #A11039 |
| | Alexa Fluor 488 anti-rabbit IgG, Thermo Fisher #A21441 |
| | Cy3-AffiniPure F(ab')2 fragment anti-syrian hamster IgG, Jackson immune #107-166-142 |
| | Alexa Fluor 488 anti-hamster IgG, Thermo Fisher #A21110 |
| | Biotin anti-CD31 monoclonal antibody, clone MEC13.3, BDPharmigen # 553371 |
| | Alexa Fluor 647 anti-rat IgG, Thermo Fisher # A21247 |
| | FITC anti-CD45.2 monoclonal antibody, clone 104, BDBioscience # 553772 |
| | PE anti-CD11b monoclonal antibody, clone M1/70, BDBioscience # 561689 |
| | FITC anti-F4/80 monoclonal antibody, clone BM8, Biolegend # 123108 |
| | Alexa Fluor 647 anti-hamster IgG, clone A21451, Invitrogen |
| | Polyclonal anti-Axl, R&D # AF854 |
| | Goat IgG control, R&D # AB-108-C |
| | Polyclonal anti-Cleaved Caspase 3, Cell Signaling, # 9661 |
| | Polyclonal anti-Phospho-Axl, R&D, # AF2228 |
| | PE-Cy7 anti-DPPIV/CD26 monoclonal antibody, clone H194-112, Biolegend # 137810 |
| | Biotin anti-FDC monoclonal antibody, clone FDC-M1, BDPharmigen # 551320 |
| | PE anti-MadCAM-1 monoclonal antibody, clone MECA-367 , Biolegend # 120709 |
| | FITC anti-CD34 monoclonal antibody, clone RAM34, eBioscience # 48-0341 |
| | eFluor 660 anti-Ki67 monoclonal antibody, clone SolA15, Invitrogen # 50-5698 |
| | BV510 anti-Ly6C monoclonal antibody, clone HK1.4, Biolegend #128033 |
| | APC-Cy7 anti-PDPN monoclonal antibody, clone 8.1.1, Sony #1237090 |
| | Purified anti-CD8 antibody, Biolegend #100746 |
| | Rat IgG2a control, Biolegend #400544 |
| Validation | All antibody used in this study were commercially available. Validation for FACS, immunofluorescence on frozen sections or in vivo studies is accessible from the supplier website (direct testing or by providing adequate references). We performed addditional antibody validation using isotype controls, streptavidins or secondary antibody only when necessary. We validated GFP staining as positive signal using GFP- tissues to set the negative threshold. |

# Eukaryotic cell lines

Policy information about cell lines and Sex and Gender in Research

| | |
|---|---|
| Cell line source(s) | B16-MO5 (CVCL_WM77) were provided by Claude Leclerc (Institut Pasteur) |
| Authentication | B16-OVA (MO5) were previously validated (Fayolle et al., JI 1999). Cells were maintained in G418 (2 mg/ml) and hygromycin B (0,06 mg/ml) as described in methods. |
| Mycoplasma contamination | Cell lines were negative for mycoplasma and used within 4 passages. |

| Commonly misidentified lines<br>(See ICLAC register) | No misidentified cell lines were used in this study |
| --- | --- |

## Animals and other research organisms

Policy information about studies involving animals; ARRIVE guidelines recommended for reporting animal research, and Sex and Gender in Research

| Laboratory animals | ADAM12-GFP and ADAM12-DTR mice were previously described (Dulauroy et al, Nature Medicine 2012). TGFBR2floxed mice (Strain #012603) and TRAMP mice (Strain #003135) were obtained from Jackson Laboratory. Rip1Tag2 mice were obtained from the NCI Mouse Repository. We used mice from C57Bl/6 background, age 8-12w for MO5 studies, or as indicated in the text for TRAMP and RIPTag models. |
| --- | --- |
| Wild animals | No wild animals were used in this study |
| Reporting on sex | We used age and sex-matched males and females |
| Field-collected samples | None |
| Ethics oversight | Mice experiments were approved by the French Ministère de l'éducation nationale, de l'enseignement supérieur et de la recherche. |

Note that full information on the approval of the study protocol must also be provided in the manuscript.

## Flow Cytometry

### Plots

Confirm that:

☒ The axis labels state the marker and fluorochrome used (e.g. CD4-FITC).

☒ The axis scales are clearly visible. Include numbers along axes only for bottom left plot of group (a 'group' is an analysis of identical markers).

☒ All plots are contour plots with outliers or pseudocolor plots.

☒ A numerical value for number of cells or percentage (with statistics) is provided.

### Methodology

| Sample preparation | Tumors were cut in small pieces and processed in a solution composed of DMEM (Gibco), Liberase TL (0,26 Wunit/mL; Roche) and DNase I (1 U/ml; ThermoFisher) for 30 minutes, with manual dissociation by pipetting every 10 minutes. Cells were filtered through a 100-μm and a 40μm mesh, washed and then processed for cell staining. |
| --- | --- |
| Instrument | Data were analyzed using BD LSRFortessa cytometer. Cells were sorted using FACSARIA III (BD Biosciences) |
| Software | Flow cytometry data were analyzed with Flowjo v.10 software. |
| Cell population abundance | Purity of the samples were determined in preliminary experiments by analyzing the post-sort population by FACS. Additional details are provided in the legends. |
| Gating strategy | For FACS experiments, we used FSC/SSC gates to eliminate debris and discriminate doublets, and excluded dead cells using DAPI (unfixed samples) or Live-dead (fixed samples). For further gating of specific cell populations, boundaries between positive and negative staining were determined using isotype controls, secondary antibody alone or streptavidins alone.  For all FACS experiments of GFP+ cells, a GFP negative sample (originating from a GFP– littermate) was systematically used to define the positive/negative signal.<br>For cell sorting of stromal cells, we first excluded debris, dead cells, doublet cells, hematopoietic (CD45+) and endothelial (CD31+) cells, and then selected for positive stromal cell markers, as indicated in the legends/supp information. For isolation of GFP+ cells from tissues, the positive  gate for GFP+ cells was determined using GFP– littermates with similar treatments. |

☒ Tick this box to confirm that a figure exemplifying the gating strategy is provided in the Supplementary Information.

