## [Peer Review File · Nature Immunology]

Peer Review Information

Journal: Nature Immunology

Manuscript Title: Depletion of slow-cycling PDGFRa+ADAM12+ mesenchymal cells promotes antitumor immunity by restricting macrophage efferocytosis

Corresponding author name(s): Dr. Lucie Peduto

Reviewer Comments & Decisions:

Decision Letter, initial version:
--

28th Jul 2022

Dear Lucie,

Thank you for providing a point-by-point response to the referees' comments on your manuscript entitled "Depletion of slow-cycling PDGFRa+ADAM12+ mesenchymal cells promotes antitumor immunity by restricting macrophage efferocytosis". As mentioned previously, while they find your work of considerable potential interest, they have raised quite substantial concerns that must be addressed. In light of these comments, we cannot accept the current manuscript for publication, but would be very interested in considering a revised version along the lines provided in your response.

We invite you to submit a substantially revised manuscript, however please bear in mind that we will be reluctant to approach the referees again in the absence of major revisions.

Specifically, the revision should include new experiments to address:

- (1) examine TGF- β expression from sorted cell types found in the tumor environment
- (2) perform dose-response experiment for anti-Gas6 neutralization
- (3) re-analyze existing tumor transcriptomic datasets for CAF ADAM12 expression
- (4) examine tumor response upon DT-mediated cell depletion that targets another stromal cell subset other than ADAM12+ fibroblasts
- (5) provide quantitative analysis on the efficiency of DT-mediated cell depletion
- (6) provide more flow cytometric analysis on the CAF tumor populations and a quantitative analysis of the progeny of ADAM12+ CAFs at different time points
- (7) perform transcriptomic analysis of the ADAM12+ cells and compare to their progeny cells
- (8) examine interactions between ADAM12+ CAFs with Axl+ macrophages within the tumor environment
- (9) analyze Casp3+ cells in vicinity of CD31+ blood vessels

- (10) examine effects of depletion of Tgfbr2 on ADAM12+ CAFs in tumor environment
- (11) compare efferocytosis capacity of ADAM12+ and ADAM12- CAFs in vitro, same for macrophages upon depletion of ADAM12+ CAFs in the tumor environment.

Please include the additional textual clarifications as indicated in your response letter.

When you revise your manuscript, please take into account all reviewer and editor comments, please highlight all changes in the manuscript text file in Microsoft Word format.

- * Include a "Response to referees" document detailing, point-by-point, how you addressed each referee comment. If no action was taken to address a point, you must provide a compelling argument. This response will be sent back to the referees along with the revised manuscript.
- * If you have not done so already please begin to revise your manuscript so that it conforms to our Article format instructions at <http://www.nature.com/ni/authors/index.html>. Refer also to any guidelines provided in this letter.
- * Include a revised version of any required reporting checklist. It will be available to referees (and, potentially, statisticians) to aid in their evaluation if the manuscript goes back for peer review. A revised checklist is essential for re-review of the paper.

The Reporting Summary can be found here:

When submitting the revised version of your manuscript, please pay close attention to our [href="https://www.nature.com/nature-portfolio/editorial-policies/image-integrity">Digital Image Integrity Guidelines. and to the following points below:](https://www.nature.com/nature-portfolio/editorial-policies/image-integrity)

[REDACTED]

If you wish to submit a suitably revised manuscript we would hope to receive it within 6 months. If you cannot send it within this time, please let us know. We will be happy to consider your revision so long as nothing similar has been accepted for publication at Nature Immunology or published elsewhere.

Nature Immunology is committed to improving transparency in authorship. As part of our efforts in this direction, we are now requesting that all authors identified as 'corresponding author' on published papers create and link their Open Researcher and Contributor Identifier (ORCID) with their account on the Manuscript Tracking System (MTS), prior to acceptance. ORCID helps the scientific community achieve unambiguous attribution of all scholarly contributions. You can create and link your ORCID from the home page of the MTS by clicking on 'Modify my Springer Nature account'. For more information please visit www.springernature.com/orcid.

Thank you for the opportunity to review your work.

Kind regards,

Laurie

Laurie A. Dempsey, Ph.D.
Senior Editor
Nature Immunology
l.dempsey@us.nature.com
ORCID: 0000-0002-3304-796X

Referee expertise:

Referee #1: cancer microenvironment

Referee #2: Cancer-associated fibroblasts

Referee #3: Efferocytosis

Reviewers' Comments:

Reviewer #1:

Remarks to the Author:

Di Carlo et al. describe a small subset of ADAM12+PDGFRa+ mesenchymal stem cell in early tumor development in mouse models of melanoma, pancreatic cancer, and prostate cancer. This population

produces important microenvironment factors including Gas6, Lgals3, Has2, and Csf1 and is induced by TGF-beta. These cells appear to have low proliferative index and expression of mesenchymal progenitor cells (Ngfr, sca-1, Vcam1). Genetic depletion of these cells alters the microenvironment to be permissive of T-cells. Correlation of ADAM12 in human cancers is also demonstrated. Overall, this is an interesting mouse model with a variety of correlative analyses, but the mechanistic underpinnings of ADAM12+ mesenchymal cells or clear human translational studies is lacking. The Axl experiments are probably the most convincing, but a little more depth on this pathway is needed.

Comments/Questions:

- 1) Abstract "In human melanoma, pancreatic ductal adenocarcinoma (PDAC), prostate and colon cancer, ADAM12 stratifies patients with high levels of hypoxia and innate resistance mechanisms including AXL." Pancreatic is misspelled and might want to reword In human melanoma and cancers of pancreatic, prostate, and colon origin, ADAM12 stratifies...
- 2) Figure 1D – what is the * signify and the error bars for the animal study? Should be in the legend
- 3) Figure 2I – the second scatter plot is not clarified in the legend. Is this the width of the vessel?
- 4) The finding that TGF-beta induces ADAM12+ cells is interesting and the authors allude to macrophage derived TGF-beta. But were there other CAFs or tumor cells that produce TGF-beta in their model?
- 5) Figure 4G - the Efferocytosis test in BMDM should have more clear details of the mouse. Axlfl/fl vs LysMCreAxlfl/fl should be explained in the text or legend. The effects are quite modest and the effect of CM from GFP- fibroblasts does look like there is some induction indicating that the induction is likely not from a factor that is unique to ADAM12+ fibroblasts. Probably the best experiment would be to use a neutralizing antibody to Gas6 in a dose dependent manner to abrogate the efferocytosis.
- 6) Figure 6 – human tissue correlatives in databases are supportive, but can the authors demonstrate what cells ADAM12 is expressed in human tumors and if the presence of these cells is associated with outcomes in a different cohort of patients outside of re-analysis of the consortium databases?

Reviewer #2:

Remarks to the Author:

The role of cancer-associated fibroblasts (CAFs) remains a controversial topic. Whereas expression of CAF signatures predicts poor prognosis and immunotherapy failure in multiple cancer types, experiments of genetic cell ablation in cancer mouse models have failed to demonstrate a role for CAFs in tumor progression and immune evasion. Their heterogeneity and origin also remain under intense scrutiny. Peduto and colleagues shed light on some of these questions. Using a genetic cell ablation strategy in melanoma models, they provide evidence that a slow-cycling CAF subset labeled by the marker gene ADAM12 is responsible for T cell exclusion and for inhibiting anti-tumor macrophage activity. Through lineage tracing analyses, authors also show that ADAM12+ fibroblasts are recruited to the tumor microenvironment early during tumorigenesis and produce long-lasting progeny that differentiates into alpha-SMA+ CAFs at the tumor periphery. I found the study interesting and relevant, but the data are too preliminary and must be reinforced to a large extent. Besides, there are a number of technical and conceptual caveats that make me somewhat skeptical about the authors' conclusions. I detail my criticisms below:

- 1) A large part of the authors' conclusions is based on the results of ADAM12+ CAF ablation experiments. Three aspects that must be properly controlled:

- DTR-mediated cell ablation creates inflammation, which may exacerbate macrophage activity, T cell infiltration and facilitate anti-tumor responses. Thus, the authors must show that the reported effects are specific to the ablation of the ADAM12+ fibroblast population.

- Authors should provide evidence that ablation targets the ADAM12+ population but spare other CAFs in the TME.

- Tumor draining lymph nodes play a central role in anti-tumor responses. They contain heterogeneous fibroblast populations. Therefore, authors should analyze if ADAM12-DTR+ cell ablation affects Fibroblast Reticular Cells composition and, possibly, the activation of CD8+ T cells in draining LNs. This may explain why early ablation does not exert therapeutic responses.

2) The authors' experiments focus on the ADAM12+ CAFs, but a description of the other fibroblast populations in the TME is largely missing. The study will gain priority if the authors could perform scRNAseq of the TME and assess CAF diversity before and after ADAM12+ cell ablation. This analysis may also help clarify the specificity of the genetic ablation strategy and strengthen the lineage relationships between ADAM12+ fibroblasts and the rest of the CAF subtypes/states.

3) Do ADAM12+ CAFs play a role in the TRAMP and/or RIP-TAG tumor model? Could you please repeat the cell ablation experiments in these models?

4) Lineage tracing analyses included in the manuscript are not quantitative. Authors should include information on clone size and number over time. It is unclear if ADAM12+ cells keep producing progeny over tumor evolution or only at the onset of the process.

5) Along the lines above, the authors must investigate the kinetics of ADAM12+ cell division and differentiation. Authors propose that the frequency of ADAM12-GFP+ fibroblasts gradually decreases over time based on the reduction of the GFP+ cell proportion. It is unclear if these changes are the result of switching off GFP expression, an active reduction of the cell population, or a dilution of the population due to the expansion of other fibroblast subsets in TME. Authors may consider quantifying absolute numbers of GFP+ cells at different time points. I also advise to re-analyze the scRNAseq dataset from Davidson et al (Cell Reports, 2020), focusing on the evolution of ADAM12 levels and ADAM12-GFP+ cell gene expression signature.

6) Authors suggest that ADAM12-GFP+ cells are "persistent" slow proliferative cells based on their gene profile. They also provide EdU incorporation data. However, they do not study proliferation (or EdU incorporation) in other fibroblast populations (such as PDPN+ or aSMA+ CAFs). This may help confirm that slow proliferation is a specific feature of ADAM12-GFP+ cells.

7) Some aspects of the role and origin of CAFs must be clarified. In the first part of the study, the authors provide evidence supporting a role of ADAM12+ CAFs in immune evasion. In the second part, lineage tracing data indicates that ADAM12+ fibroblasts differentiate into other CAFs subsets. Which is the effector population in terms of immune evasion? The ADAM12+ progenitor subset or its differentiated progeny? Besides, lineage tracing experiments show that Adam12+ cells only contribute a small subset of fibroblasts in the TME (<12%), implying that they do not represent the major CAF source in tumors. Authors must clarify these aspects.

8) ADAM12+ stromal cells localized near CD206+ TAMs and the authors infer an important role for

this interaction. Yet, they fail to show if other CAF populations present at the tumor margin are in close contact with Axl1+ TAMs.

9) The evidence for ADAM12+ mesenchymal cell in controlling macrophage behavior, particularly efferocytosis, is largely based on in vitro observations and differences between experimental conditions are significant but marginal (Fig 4G). Authors should provide additional in vivo evidence supporting this effect.

Reviewer #3:

Remarks to the Author:

Di Carlo et al. establish the relevance of a previously described but poorly understood population of ADAM12+PDGFR α + mesenchymal cells in immunity to tumors. The authors show how tumor conditions, especially macrophage-derived soluble factors, reactivate a developmental program in stromal cells characterized by the expression of ADAM12, a hypo proliferative state, and the production of factors that promote macrophage efferocytosis. The authors identify that these ADAM12+ stromal cells contribute to the immunosuppressive environment of tumors by promoting hypoxia and inducing macrophages to acquire a less inflammatory phenotype. The use of multiple models (both transplantable and orthotopic) is a strength of the paper that serves to bolster the author's description of ADAM12+ stromal cells' role in tumor immunity. They identify several factors, including TGF- β , IL-1 β , and OSM that contribute to acquisition of the ADAM12+ phenotype by stromal cells, and use fate-mapping techniques to demonstrate that ADAM12+ progenitors presage a peritumoral population of stromal cells that is maintained throughout the course of tumor progression.

The impact of this work is clear; the means by which stromal cells instruct macrophage (and subsequently T cell) function in tumors is poorly understood, and this study contributes to the field by identifying the instructive role of a unique stromal cell subset. However, some methodological concerns and unsupported claims dampen the enthusiasm of this reviewer.

- Evidence of the functional role of ADAM12+ mesenchymal cells relies heavily on the finding that tumor growth is inhibited in ADAM12-DTR mice that lack ADAM12+ stromal cells upon treatment with diphtheria toxin. Attributing the entirety of the tumor growth phenotype to the activity of ADAM12+ stromal cells requires absolute specificity of ADAM12-DTR expression in stromal cells. This reviewer finds that the specificity of ADAM12 expression in ADAM12-DTR mice is incompletely explored in this study. While Fig. 1C does show that there is no expression of ADAM12-GFP in any CD45+ or CD31+ cells, this seems to potentially conflict with previous reports demonstrating ADAM12+ expression on T cells, including Tregs (i.e.: PMID 32572163, PMID 32572163, as well as examination of available of sequencing datasets). A more detailed gating strategy in the supplement with technical controls (i.e. fluorescence minus one control for GFP) would be helpful in this regard.

The lack of T cell depletion, and specifically Treg depletion, observed upon administration of diphtheria toxin in Fig. S1D would somewhat ameliorate these concerns, except that the proportion of Tregs among CD45+ tumor-infiltrating cells is extremely surprising and inconsistent with previous reports. Fig. S1D indicates that while ~10% of all CD45+ cells are CD4+ T cells, only ~0.3% of all CD45+ cells are FoxP3+ Tregs. This implies that only ~3% of all CD4+ T cells are FoxP3+ in tumors, which is at least an order of magnitude lower than what has been previously reported in B16 or B16-OVA tumors (Klages et al., Cancer Research, 2010; Magnuson et al., PNAS, 2018). Providing the gating strategy for flow cytometric analysis of these cells might help explain this discrepancy.

- Additionally, the authors do not report on the efficiency of the depletion of ADAM12+ cells in their DTR model. This data is essential for evaluating the efficacy of the model and determining whether the observed changes in tumor growth are dependent on depletion of the cells of interest.

- One potential confounding factor of this model is the induction of stromal cell death by administration of DT. The authors demonstrate that ADAM12+ stromal cells were located adjacent to peritumoral blood vessels; thus, administration of DT and subsequent cell death is likely to alter the vasculature in a manner that is not necessarily dependent on the identity of the dying cell. It is plausible that the sensing of cell death increases permeability of peritumoral blood vessels and permits infiltration of T cells into the tumor as observed in Fig. 2A. For example, the upregulation of ICAM-1 that is attributed to normalization of vasculature in the absence of ADAM12+ stromal cells could also be attributed to inflammation secondary to death of perivascular cells. Thus, the contributions of cell death induced by DT vs. the lack of ADAM12+ stromal cells to the observed phenotype are extremely difficult to separate in this model.

If this is indeed the case, then it is difficult to determine whether ADAM12+ stromal cells are indeed a causative determinant of immunity to tumors, or rather a correlative indicator of an ongoing anti-tumor immune response. While the authors demonstrate that ADAM12+ stromal cells produce factors that alter macrophage phenotype and anti-tumor activity, they also demonstrate that they are also activated by IL-1 β and OSM and thus raise the possibility that their induction may be downstream of macrophage activation. Thus, careful consideration on the interpretation of the findings is merited.

- While the growth-inhibitory role of TGF- β on stromal cells has been well-established, the mechanism of its specific induction of the development of ADAM12+ cells requires further investigation. The authors claim that the specific upregulation of Tgfbr3, but not Tgfbr1 or Tgfbr2, in ADAM12+ cells acts to enhance their TGF- β signaling and induce acquisition of the ADAM12+ phenotype. This claim needs to be validated by multiple methods beyond RNA sequencing results, including at the protein level.

- The authors go on to show expression of Axl in tumor-associated macrophages in close proximity to ADAM12+ stromal cells in Fig. 4, and infer a functional relationship between these cells. The specificity of Axl staining should also be explored here. Axl is also expressed by fibroblasts, and its signaling may directly affect fibroblast phenotype in addition to its role in macrophage-mediated efferocytosis. Given the results observed upon anti-Axl treatment later in the figure, Axl expression should be evaluated in non-macrophage populations, and a functional role for Axl in fibroblasts' acquisition of the ADAM12+ phenotype should also be explored.

The authors subsequently claim in Fig. 4G that Axl-sufficient, but not Axl-deficient, BMDMs exhibit increased efferocytic capacity in the presence of conditioned media from GFP+, stromal cells. The stated increase in efferocytosis of apoptotic MO5 cells by Axl-sufficient BMDM is unconvincing despite the observed statistical significance, especially given that there is no indication that this experiment was performed multiple times with similar results. Statistical analysis with correction for multiple comparisons is appropriate in this case, and results may no longer be significant when the appropriate statistical test is applied.

Furthermore, there is no indication in the methods section or otherwise as to how this efferocytosis experiment was performed. The method of inducing apoptosis in MO5 cells, the ratio of BMDM to apoptotic MO5 cells, the fluorescent marker used to measure efferocytosis, and the baseline for the fold increase on the y-axis of Fig. 4G are all critical pieces of information in evaluating the validity of this experiment, and are indicated nowhere in the text.

In addition to efferocytosis by macrophages, stromal cells have also been demonstrated to have

efferocytic functions. The efferocytosis of apoptotic cargo by stromal cells should also be evaluated in this context; the relative efferocytic activity of ADAM12⁺ vs. ADAM12⁻ cells may be instructive in their acquisition of their respective phenotypes.

- The authors report that cleaved caspase-3-expressing cells are more abundant in the tumors of ADAM12-DTR mice treated with DT, and conclude that this is a result of decreased efferocytosis in the absence of ADAM12⁺ stromal cells. However, increased cleaved caspase-3 staining may also be explained by increased killing of tumor cells in DT-treated mice independent of any effect on efferocytosis. Thus, this finding may simply be correlate of a generally enhanced anti-tumor immune response in ADAM12-DTR mice upon treatment with DT.

The dependency of the observed anti-tumor effect of ADAM12⁺ cell depletion on decreased efferocytosis was evaluated using concurrent treatment of ADAM12-DTR mice with an anti-Axl agonist antibody. The authors conclude that depletion of ADAM12⁺ cells promotes tumor immunity through Axl-dependent effect on efferocytosis because of the observed restoration of tumor growth in DT-treated mice also treated with the anti-Axl antibody. However, signaling through Axl directly on tumor cells has also been demonstrated to elicit potent pro-tumorigenic effects. Thus, the dependency of the observed phenotype on the inhibition of efferocytosis is still unclear. Treatment of ADAM12-DTR mice with anti-Axl in the absence of DT administration will be a critical control to resolve this question.

- This reviewer does not have the expertise to evaluate the claims of clinical relevance in Figure 6. However, it should be noted that this analysis does not distinguish between ADAM12 expression on stromal cells vs. tumor cells, which complicates interpretation of these results.

Rigor

This manuscript requires a much more detailed description of methods in many cases, and the lack of methodological details often obscures interpretation of the data. For example, several figures reference the tumor margin and quantify infiltration across the margin, but there is no description of how the margin is determined.

Representation of tumor growth data should also be more comprehensive. Fig. 1D shows average growth plots, but spider plots of tumor growth for each individual mouse should also be represented so that variability can be adequately assessed. Both average growth and individual spider plots should also be represented alongside Fig. 4L. Additionally, tumor growth inhibition (TGI) score is shown in Fig. 1E and Fig. 4L, but there is no reference to or description of how this metric is calculated or any statistics to ascertain significance. Furthermore, tumor growth is evaluated in Fig. S1E using a different metric than the TGI score shown in Fig. 1E and Fig. 4L. Tumor growth in Fig. S1E appears to be evaluated at a much later timepoint than Fig. 1D-E, as the tumor volume of the control + DT group is approximately double the average at Day 18 in Fig. 1D. The day at which this analysis was performed is not stated in the text or figure. These metrics should remain consistent throughout the manuscript to facilitate comparison across figures, and to evaluate the claim of dependence on CD8⁺ T cells in Fig. S1E.

The lack of EdU incorporation by ADAM12⁺ stromal cells in Fig. 3E is supported by RNA sequencing results, but a side-by-side comparison of EdU incorporation of Pdpn⁺ADAM12⁺ vs. Pdpn⁺ADAM12⁻ stromal cells would be the best validation of their slow-cycling phenotype. This can be calculated from the existing images in Fig. 3E.

Figure 4B indicates CD11b staining in red, GFP expression in green, and F4/80 staining in blue; however, the inset shows P-Axl expression in red. This appears to be an inadvertent substitution of CD11b for P-Axl staining in red.

Statistical analysis also lacks rigor in many cases; every comparison utilizes unpaired student's t-test with no correction for multiple comparisons even though it is an inadequate test in several panels.

Author Rebuttal to Initial comments

See inserted PDF

Reviewer #1

(Remarks to the Author)

Di Carlo et al. describe a small subset of ADAM12+PDGFRa+ mesenchymal stem cell in early tumor development in mouse models of melanoma, pancreatic cancer, and prostate cancer. This population produces important microenvironment factors including Gas6, Lgals3, Has2, and Csf1 and is induced by TGF-beta. These cells appear to have low proliferative index and expression of mesenchymal progenitor cells (Ngfr, sca-1, Vcam1). Genetic depletion of these cells alters the microenvironment to be permissive of T-cells. Correlation of ADAM12 in human cancers is also demonstrated. Overall, this is an interesting mouse model with a variety of correlative analyses, but the mechanistic underpinnings of ADAM12+ mesenchymal cells or clear human translational studies is lacking. The Axl experiments are probably the most convincing, but a little more depth on this pathway is needed.

Comments/Questions:

1) Abstract “In human melanoma, pancreatic ductal adenocarcinoma (PDAC), prostate and colon cancer, ADAM12 stratifies patients with high levels of hypoxia and innate resistance mechanisms including AXL.” Pancreatic is misspelled and might want to reword In human melanoma and cancers of pancreatic, prostate, and colon origin, ADAM12 stratifies...

We thank the reviewer for all comments/questions. We have corrected as suggested.

2) Figure 1D – what is the * signify and the error bars for the animal study? Should be in the legend

We have now added the missing information in the legend. For the tumor growth curve (average tumor volume) in Fig 1d, we assessed statistical significance using an ordinary two-way ANOVA, and obtained $p=0.0298$ (d15) and $p=0.0053$ (d18) (more experiments were now included due to revisions). Values are mean and SD (standard deviation). For more clarity, we now also show the individual animal growth curves in Extended Fig. 1h.

3) Figure 2I – the second scatter plot is not clarified in the legend. Is this the width of the vessel?

Yes it's the width of the vessel, we have now added the missing information on the scatter plot in Figure 2i.

4) The finding that TGF-beta induces ADAM12+ cells is interesting and the authors allude to macrophage derived TGF-beta. But were there other CAFs or tumor cells that produce TGF-beta in their model?

To further investigate this point, we have now analyzed Tgf-beta expression by qRT-PCR in several cell types isolated from melanomas, including tumor cells and CAFs. We show that tumor macrophages are the highest expressor of Tgfb1 in advanced tumors, although other cell types including T cells, tumor cells and endothelial cells express some levels of Tgfb1 (Extended Data Fig 4f). To have a more comprehensive view of the different immune and fibroblastic cell types within the TME, and define their contribution at earlier stages of tumorigenesis, we have further analyzed Tgfb1 expression in scRNAseq from murine

melanoma tumors at different tumor stages (Davidson, Cell Reports 2020). These data confirmed that, although at later tumor stages macrophages are a major source for Tgfb1, several cell types produce Tgfb1 in the TME at early stages of tumorigenesis (now shown in Extended Data Fig. 4e). These data suggest that ADAM12+ stromal cells are induced by several cell types producing TGF-beta upon tumorigenesis.

5) Figure 4G - the Efferocytosis test in BMDM should have more clear details of the mouse. Axlfl/fl vs LysMCreAxlfl/fl should be explained in the text or legend. The effects are quite modest and the effect of CM from GFP- fibroblasts does look like there is some induction indicating that the induction is likely not from a factor that is unique to ADAM12+ fibroblasts. Probably the best experiment would be to use a neutralizing antibody to Gas6 in a dose dependent manner to abrogate the efferocytosis.

We have now added in the text details for Axlfl/fl vs LysMCreAxlfl/fl mice model used in efferocytosis assay. We also performed additional experiments *in vitro* and *in vivo* to further investigate efferocytosis induction by ADAM12+ cells.

The effect of CM from GFP- cells is coherent with the observation that some levels of Gas6 are present in the CM of GFP- cells, although lower than in the CM of GFP+ cells (previous Fig. 4F). As the CM was collected after a few days in culture, we asked whether culture conditions might have affected this response. Indeed, we observed that Gas6 expression in stromal cells *in vitro* is increased in reduced serum conditions (Extended Data Fig 4d; note that in the first version the graph labels were inverted). By collecting CM in complete medium rather than in 2% serum, we observed that GFP+ cells produced similarly high levels of Gas6, but levels of Gas6 in GFP- cells were reduced (Fig. 4e), suggesting that they were artificially induced by culture conditions. By repeating the efferocytosis assay in these conditions, which are more representative of the unaltered state of GFP+ and GFP- cells *in vivo*, we observed improved specificity of GFP+ induction of efferocytosis compared to GFP- cells (new Fig 4f; statistical significance determined by two-way ANOVA, p=0,0086). Further supporting a specific role for GFP+ cells, we now show that the CM from GFP+ cells, but not the CM from GFP- cells, induce immunosuppressive / proangiogenic genes in efferocytic BMDM (Extended Data Fig. 5d), consistent with the *in vivo* data (Fig 4j).

We further attempted to abrogate efferocytosis using anti-Gas6 neutralizing Abs, however the experiment was not conclusive due to high variability obtained with the IgG control as well. Nevertheless, induction of efferocytosis by the CM of GFP+ cells was abrogated in BMDM lacking Axl (Fig 4f, p<0.001), showing that induction of efferocytosis by GFP+ cells requires Axl. These data support the hypothesis that GFP+ cells produce high levels of Axl ligands such as Gas6, and promote macrophage efferocytosis *in vitro* in an Axl-dependent way.

To further strengthen this claim, we performed additional experiments *in vivo*. We now show that Gas6 expression is significantly decreased in tumors depleted from ADAM12+ cells, compared to WT tumors, showing that ADAM12+ cells are a major source for Gas6 within the TME *in vivo* (Extended Data Fig. 5g). We further show that macrophages isolated from tumors lacking ADAM12+ cells (which contained less Gas6 as shown in Extended Data Fig. 5g) have decreased engulfing ability (Extended Data Fig. 5f), which is in line with their shift toward M1 phenotype (Fig. 4i,j).

6) Figure 6 – human tissue correlatives in databases are supportive, but can the authors demonstrate what cells ADAM12 is expressed in human tumors and if the presence of these cells is associated with outcomes in a different cohort of patients outside of re-analysis of the consortium databases?

We have now analyzed *ADAM12* expression in single cells RNAseq of human melanoma, pancreatic cancer and colorectal cancer, which show high expression of *ADAM12* preferentially in stromal populations (CAFs/tumor pericytes) within the TME (Extended Data Fig. 8a-e). In Extended Data Fig 8a and 8c, tumor cells are included in the scRNAseq dataset, showing very low expression of *ADAM12* compared to stromal cells. Expression of *ADAM12* in malignant melanoma cells (from scRNAseq dataset shown in Extended Data Fig 8b) are shown in the boxplot below. These data further show that out of 14 tumors, only one express significant level of *ADAM12* in melanoma cells (graph below). Overall, these data show that, for a majority of tumors, *ADAM12* expression in tumor cells is rather low compared to stromal cells within the TME.

Consistent with this, we now show that *ADAM12* expression correlates with the “activated stroma” subtype of human PDAC, which is associated with a severe prognosis (as described in Puleo et al, Gastroenterology 2018), and that *ADAM12*+ stromal cells in human PDAC are mainly found within the PDPN+PDGFRA+ populations (Extended Data Fig. 8d,e), as observed in mice models. These data are consistent with previous literature, including at the single cells level, showing that *ADAM12* is mainly expressed by stromal cells in several human tumors including liver, PDAC and colorectal cancer (Le Pabic, Hepatology 2003; Dominguez, Cancer Discovery 2020; Hoorn, BMC Cancer 2022).

ADAM12 expression has been correlated to unfavorable outcome in several human cancers, including in colorectal cancer, PDAC, gastric cancer and estrogen receptor-positive breast cancer (all references are now cited in the corresponding sections). We now further show that stratification according to *ADAM12* expression in human prostate cancer correlates with the Gleason score, which identifies high-grade tumors at high risk of recurrence (Extended Data Fig. 8f). These data are consistent with our previous report showing that *ADAM12*, highly expressed by CAFs, was required for prostate tumor progression in the TRAMP model (Peduto et al., Oncogene 2006). *ADAM12* has been identified as a marker for stromal activation and early detection of human prostate cancer and as diagnostic marker in metastatic prostate carcinoma (Bilgin Dogru 2014; Bacalod et al., 2021; Wilkinson et al., 2013). *ADAM12* has been shown to be useful as a marker for stroma activation with prognostic value in pancreatic cancer (Veenstra et al., 2018) and several other cancers, as well as to predict PD(L)-1 blockade benefits across solid tumors (Tomlins et al, 2023). Additional references are detailed in the introduction and discussion part. We have now integrated and discuss these points.

Reviewer #2

(Remarks to the Author)

The role of cancer-associated fibroblasts (CAFs) remains a controversial topic. Whereas expression of CAF signatures predicts poor prognosis and immunotherapy failure in multiple cancer types, experiments of genetic cell ablation in cancer mouse models have failed to demonstrate a role for CAFs in tumor progression and immune evasion. Their heterogeneity and origin also remain under intense scrutiny. Peduto and colleagues shed light on some of these questions. Using a genetic cell ablation strategy in melanoma models, they provide evidence that a slow-cycling CAF subset labeled by the marker gene ADAM12 is responsible for T cell exclusion and for inhibiting anti-tumor macrophage activity. Through lineage tracing analyses, authors also show that ADAM12+ fibroblasts are recruited to the tumor microenvironment early during tumorigenesis and produce long-lasting progeny that differentiates into alpha-SMA+ CAFs at the tumor periphery. I found the study interesting and relevant, but the data are too preliminary and must be reinforced to a large extent. Besides, there are a number of technical and conceptual caveats that make me somewhat skeptical about the authors' conclusions. I detail my criticisms below:

1) A large part of the authors' conclusions is based on the results of ADAM12+ CAF ablation experiments. Three aspects that must be properly controlled:

We thank the reviewer for all comments/questions. We now provided additional experiments to strengthen these different aspects, as detailed below.

- DTR-mediated cell ablation creates inflammation, which may exacerbate macrophage activity, T cell infiltration and facilitate anti-tumor responses. Thus, the authors must show that the reported effects are specific to the ablation of the ADAM12+ fibroblast population.

ADAM12+ cells are a small subset of stromal cells (< 5-8% of CD45-CD31- stromal cells,). As the targeted population is very small (< 1% of the total tumor mass), we did not expect major inflammation due to cell ablation. Nevertheless, to further address this concern, we now performed *in vivo* depletion experiments of a different stromal subset using different mice models generated in the lab (Jacob et al., Cell Stem Cell 2022). In this model, the DTR is under control of lymphotoxin beta receptor (*Ltbr*), expressed by CAFs in the murine melanoma (Figure 1A below shows *Ltbr* expression in scRNAseq from Davidson et al, Cell reports 2020). Using a reporter LTBR-GFP model, we confirmed that LTBR+ stromal cells have a similar localization at the margin of MO5 tumors (Fig. 1B below, LTBR+ cells are in green (GFP) and stromal cells are in red). We reasoned that if DT-induced death of stromal cells is sufficient to induce antitumor responses, irrespective of the nature of the targeted stroma, we should obtain a similar decrease in tumor growth in this model. In contrast to depletion of ADAM12+ stromal cells, we observed that depletion of LTBR+ stromal cells did

Figure 1

not decrease MO5 tumor growth (Fig. 1C), showing that specificities of the targeted stroma determine the outcome. These data are consistent with previous reports showing that ablation of other subsets of stromal cells within the TME, notably expressing α SMA or CCL19 (which is regulated by LTBR signaling), do not decrease tumor growth or induce anti-tumor responses (Ozdemir, 2014; Cheng 2018).

- Authors should provide evidence that ablation targets the ADAM12+ population but spare other CAFs in the TME.

We previously reported specificity of ablation of ADAM12+ cells in our DTR depletion model. We used the lineage tracing model (ADAM12-tTA-Cre^{YFP} line, a scheme is provided in Fig 5d) as it expresses the DTR transgene and permanently marks all progeny as YFP. When depleting ADAM12+ cells by injecting DT in this model, we showed that the ADAM12-DTR model is efficient at > 97% to deplete ADAM12+ cells (as measured by loss of the progeny YFP+) while Pdpn+ YFP- stroma was not affected (Fig 4b in Dulauroy et al., Nature Medicine 2012).

To further investigate this point, we have analyzed expression of human *DTR* (*HBEGF*) by qPCR in MO5 tumors growing in ADAM12-DTR mice that are treated with DT or not, compared to WT tumors (DTR-). This approach is based on the fact that only ADAM12+ cells carry the human *DTR* transgene, while WT cells express the murine *Dtr*. We observed that, when ADAM12-DTR+ mice were treated with DT to deplete ADAM12+ cells, expression levels of human *DTR* in tumors was drastically reduced (similar to tumors in DTR- littermate) (Extended Data Fig.1g), confirming high efficiency of depletion.

Concerning the other CAFs, we show that the frequency and absolute numbers of total CAFs do not decrease after depletion of ADAM12+ cells (Fig 2c and Extended Data Fig 2b). These data are consistent with our observation that ADAM12+ cells represent a small subset of the total CAFs. To further investigate this last point, we have now performed a detailed analysis of the abundance of different CAFs subsets when depleting ADAM12+ cells (see point 2 below).

- Tumor draining lymph nodes play a central role in anti-tumor responses. They contain heterogeneous fibroblast populations. Therefore, authors should analyze if ADAM12-DTR+ cell ablation affects Fibroblast Reticular Cells composition and, possibly, the activation of CD8+ T cells in draining LNs. This may explain why early ablation does not exert therapeutic responses.

To address this point, we have performed additional experiments in vivo. These data show that there is no difference of CD8+ T cells activation in the draining LNs after depletion of ADAM12+ cells (Extended Data Fig. 1k). Consistent with these results, we did not observe significant differences in Fibroblastic Reticular Cells composition of the draining LNs after depletion of ADAM12+ cells (Extended Data Fig. 1l, gates for FRC populations are defined in methods).

2) The authors' experiments focus on the ADAM12+ CAFs, but a description of the other fibroblast populations in the TME is largely missing. The study will gain priority if the authors could perform scRNAseq of the TME and assess CAF diversity before and after ADAM12+ cell ablation. This analysis may also help clarify the specificity of

the genetic ablation strategy and strengthen the lineage relationships between ADAM12+ fibroblasts and the rest of the CAF subtypes/states.

To investigate how depletion of ADAM12+ cells affect the other CAFs, we have now performed several additional experiments *in vivo*. Characterization of CAFs heterogeneity in a similar murine melanoma model at different tumor stages has been previously reported (Davidson et al, Cell Reports 2020). By scRNAseq analysis, Davidson et al reported 3 CAF subsets (S1, S2, S3), which evolve over time. S1 is mostly found at early tumor stages (day 5) while S2 and S3 are more abundant at later stages. CAFs heterogeneity at our time of analysis is therefore quite low. By FACS, we identified S1, S2, and S3 based on markers as described in Davidson et al.: S1 (PDGFRa^{High} Pdpn^{High} CD34^{High} DPP4^{High} Ly6c^{High}), S2 (PDGFRa⁺ Pdpn⁺ CD34^{Mid/Low} DPP4^{Low/-}) and S3 (PDGFRa^{Mid} Pdpn^{Low} CD34^{Low} Dpp4⁻). The gating strategy and marker genes for S1, S2, S3 are now shown in Extended Data Fig. 3a,b. By isolating S1, S2 and S3 from MO5 tumors, we confirmed a similar gene expression by qPCR as reported in the scRNAseq from Davidson et al (Extended Data Fig. 3c). In tumors depleted from ADAM12+ cells, we observed a decrease in the frequency of S1, while S2 and S3 increased (Extended Data Fig. 3d). As S2 and S3 express higher levels of pericyte markers including *Rgs5* and *Cspg4* (coding for NG2) (Extended Data Fig. 3b,c), these data are in line with increased pericyte coverage observed in tumors depleted from ADAM12+ cells (Fig. 2i).

We further investigated the mechanism. The decrease in S1 was unlikely due to direct ablation of ADAM12+ cells, as ADAM12+ cells were mainly DPP4^{Low/-} CD34^{Mid/Low}, consistent with previous scRNAseq data¹² (Extended Data Fig. 3e,f), and represented a small percentage of total stromal cells. In figure 2, we have shown that the major difference in the TME when ablating ADAM12+ cells is decrease in tumor hypoxia, a consequence of vascular normalization (Fig. 2f, i-k). We therefore asked whether hypoxia might affect CAF differentiation toward S1. Accordingly, we observed that hypoxia induced upregulation of *Cd34*, *Dpp4*, *C3*, *Il6ra* and *Il6st* (marker genes of S1 stromal population, Extended Data Fig. 3b) on stromal cells *in vitro*, while expression of broad fibroblast markers such as *Pdgfra* and *Pdpn* were not affected (Extended Data Fig. 3g). These data are consistent with previous reports showing that inflammatory CAFs are enriched in tumor hypoxic regions^{57, 58}, and further suggest that depletion of ADAM12+ cells normalize the other CAFs subsets by decreasing tumor hypoxia.

To investigate lineage relationships of ADAM12+ cells, we have performed lineage tracing experiments *in vivo*. Our data indicate that ADAM12+ cells generated a small fraction of the total CAFs (<12%). Quantifications and detailed characterization of the progeny of ADAM12+ cells (YFP+) in melanoma, RIPTag and TRAMP models are now shown in Fig. 5 and Extended Data Fig. 7b-e. We show that YFP+ cells localized both within the stroma around tumoral glands and close to blood vessels, and differentiated toward cells expressing varying levels of Acta2 (aSMA), *Cspg4* (NG2), and *Pdgfrb* (Extended Data Fig 7b-d). resembling S3 stromal populations. Although in proximity to blood vessels, YFP+ cells were localized outside the vascular basement membrane (unlike healthy pericytes), consistent with detached pericytes or perivascular aSMA^{mid} myofibroblasts (Fig. 5f and Extended Data Fig. 7d). These data are in line with previous reports showing that S3 populations represent a small percentage of total CAFs (from 1-10% depending on the tumor stage) containing pericytes and detached perivascular fibroblasts like cells (Davidson et al., Cell Reports 2020). We now discuss these points.

3) Do ADAM12+ CAFs play a role in the TRAMP and/or RIP-TAG tumor model? Could you please repeat the cell ablation experiments in these models?

We choose to use the RIPTag and TRAMP models as they develop spontaneous tumors, with tumor stages similar to human cancer, from hyperplasia (at 4-6 weeks) to carcinoma stages/metastasis. These characteristics were key to obtain insightful information on the induction and fate of the ADAM12 lineage during tumor progression (Fig. 5e-j) and differentiation of the lineage over several months within its own niche *in vivo* (Extended Data Fig. 7c-e). Indeed, tumors in these models grow very slowly, up 3-4 months for the RIP-Tag and 8-9 months for the TRAMP model.

Our data indicated that ADAM12+ cells are induced in RIPTag and TRAMP tumors and have a similar gene expression and perivascular localization as in the melanoma (Fig 5a-c), showing that similar cells develop in spontaneous tumors. To further investigate the role of ADAM12+ CAFs in a spontaneous tumor model, we have now performed additional *in vivo* experiments. Due to time constraint and technical reasons (we do not have the TRAMP model anymore and TRAMP tumors growth over 8-9 months), we performed the depletion experiments in the RIPTag tumor model, which has a faster tumor growth rate compared to the TRAMP model. We generated RIP-Tag2 x ADAM12-DTR+ mice and littermate RIP-Tag mice (DTR-). In line with our observations in the melanoma, we observed that ablation of ADAM12+ cells from week 11-14 in RIP-Tag:ADAM12-DTR mice induced a significant decrease in tumor size compared to RIP-Tag WT littermate mice (Extended Data Fig. 7g), and tumors displayed increased infiltration of CD3+ T cells (Extended Data Fig. 7h). These data show that ADAM12+ cells play a pro-tumor role in a spontaneous tumor model.

Concerning human prostate cancer, ADAM12 has been identified as a marker for stromal activation and early detection, as well as a diagnostic marker in metastatic prostate carcinoma (Bilgin Dogru 2014; Bacalod et al., 2021; Wilkinson et al., 2013). We have now further analyzed ADAM12 expression in human tumors, and show that it correlates with high Gleason score, which identifies aggressive high-grade tumors at high risk of recurrence (Extended Data Fig. 8a). These data are consistent with our previous report showing that ADAM12, which was specifically expressed by CAFs, was required for prostate tumor progression in the TRAMP model (Peduto et al., Oncogene 2006). We have further discuss these points.

4) Lineage tracing analyses included in the manuscript are not quantitative. Authors should include information on clone size and number over time. It is unclear if ADAM12+ cells keep producing progeny over tumor evolution or only at the onset of the process.

To address this point, we now provide quantifications of lineage tracing experiments (YFP+ cells) in melanoma, RIPTag and TRAMP tumors (frequency and absolute numbers for the three models are now shown in Fig. 5g and Extended Data Fig. 7b,e).

To further determine whether ADAM12+ cells keep producing progeny over time, or only at the onset of tumorigenesis, we have performed inducible lineage tracing of ADAM12+ cells at different time of tumorigenesis *in vivo*. This lineage tracing model is “tet-off”, removing dox will start the fate mapping and generate YFP+ cells if a cell is expressing ADAM12 at

that time. We focused on RIPTag and TRAMP models, as tumors develop at puberty and progress slowly through well-defined tumor stages similar to human disease (experimental strategy is shown in Fig. 5e). Our data show that ADAM12+ cells produce the major part of their progeny at the onset of tumor growth, although a small percentage is produced at later tumor stages. The frequency and absolute numbers of YFP+ cells in these different settings are shown in Fig. 5g and Extended Data Fig. 7e (“Early” and “Late” tumors, as defined in Fig 5e).

We further show that YFP+ cells localize at specific places within the TME (Fig. 5f). These data suggest that one or several clones of ADAM12+ cells expanded locally. We observed that ADAM12+ cells rapidly downregulated *Adam12* and *Pdgfra* and differentiated when cultured *in vitro* (figure below), therefore we could not investigate their progenitor function *in vitro*. These data therefore suggest that essential factors of the *in vivo* niche of ADAM12+ cells (which are not known yet) play an essential role in their maintenance, and most likely regulate the clone size over time.

5) Along the lines above, the authors must investigate the kinetics of ADAM12+ cell division and differentiation. Authors propose that the frequency of ADAM12-GFP+ fibroblasts gradually decreases over time based on the reduction of the GFP+ cell proportion. It is unclear if these changes are the result of switching off GFP expression, an active reduction of the cell population or a dilution of the population due to the expansion of other fibroblast subsets in TME. Authors may consider quantifying absolute numbers of GFP+ cells at different time points. I also advise to re-analyze the scRNAseq dataset from Davidson et al (Cell Reports, 2020), focusing on the evolution of ADAM12 levels and ADAM12-GFP+ cell gene expression signature

To address this question, we have now quantified absolute numbers of ADAM12-GFP+ cells by FACS at different time points. We show that ADAM12+ cells are not present in the normal skin, increase gradually in the first days after tumor injection, peak at around 10 days, then decrease at later time point, both in frequency and absolute numbers (Fig 1c and Extended Data Fig. 1d). It is therefore a reduction of the GFP population and not a dilution among other fibroblasts subsets of the TME.

The ADAM12-GFP+ population could switch off ADAM12 expression and nevertheless expand within the TME. To further investigate this point, we quantified the lineage traced ADAM12 population (YFP+ cells) in tumors. Our data show that the progeny of ADAM12+ cells (including cells that may have switched off ADAM12 expression) is reduced in frequency and absolute numbers in late tumors compared to early tumors (Fig. 5g and Extended Data Fig 7e).

Our data are in line with the scRNAseq dataset of murine melanoma, which report no *Adam12* expression in normal skin, and increased *Adam12* expression in CAFs from day 5 to 11 (Davidson et al., Cell reports 2020; and Extended Data Fig. 3e). As Davidson and colleagues did not report analysis of CAFs after day 11, which is when we observe a significant reduction in the frequency and absolute numbers of ADAM12-GFP+ (Day 14-17, Fig. 1c and Extended Data Fig. 1d), it prevented analysis of ADAM12-GFP+ cell gene expression signature in CAFs when the ADAM12+ population is reduced.

Overall, these data show that ADAM12-GFP+ cells and their progeny expand in the early phases of tumorigenesis and undergo an active reduction of the population at later tumor stages.

6) Authors suggest that ADAM12-GFP+ cells are “persistent” slow proliferative cells based on their gene profile. They also provide EdU incorporation data. However, they do not study proliferation (or EdU incorporation) in other fibroblast populations (such as PDPN+ or aSMA+ CAFs). This may help confirm that slow proliferation is a specific feature of ADAM12-GFP+ cells.

We have now investigated proliferation in other fibroblasts populations by performing additional experiments of Edu incorporation. Our data show that both Pdpn+ and aSMA+ CAFs proliferate significantly more than ADAM12+ cells, with aSMA+ CAFs being the most proliferative population (Fig 3e, $p < 0.0001$ compared to ADAM12+ cells). These data are in line with previous reports identifying aSMA+ CAFs as the most proliferative CAF subset in melanoma (Davidson et al., Cell reports 2020) and confirm that ADAM12+ cells are slow proliferating cells within the TME.

7) Some aspects of the role and origin of CAFs must be clarified. In the first part of the study, the authors provide evidence supporting a role of ADAM12+ CAFs in immune evasion. In the second part, lineage tracing data indicates that ADAM12+ fibroblasts differentiate into other CAFs subsets. Which is the effector population in terms of immune evasion? The ADAM12+ progenitor subset or its differentiated progeny? Besides, lineage tracing experiments show that Adam12+ cells only contribute a small subset of fibroblasts in the TME (<12%), implying that they do not represent the major CAF source in tumors. Authors must clarify these aspects.

To investigate this question, we have further characterized the progeny of ADAM12+ cells (YFP+). We now show that, compared to ADAM12+ cells (GFP), YFP+ cells downregulated several factors including *Ccl2*, *Csf1*, *Gas6*, *Cxcl12*, *Il6* and *Angpt1*, while upregulating *Car9* and heterogeneous levels of *Angpt2*, *Acta2* and *Pdgfrb* (Extended Data Fig. 7c). As *Car9* and *Angpt2* have essential roles in blocking anti-tumor immunity^{85, 51, 52}, these data suggest that both ADAM12+ cells and their progeny promote immunosuppression, although through different mechanisms. This is coherent with the global anti-tumor effect we observe when depleting ADAM12+ cells and their progeny.

Inducible lineage tracing experiments indicated that ADAM12+ cells generate <12% of CAFs within the TME. We show that YFP+ cells localized at the tumor margins, both within

the stroma and around blood vessels (although outside the vascular basement membrane, in contrast to healthy pericytes), and express varying levels of NG2 and aSMA (Extended Data Fig. 7b,d). Based on their gene expression and histological analysis, the progeny of ADAM12+ cells resemble the S3 populations described by Davidson et al. (Cell Reports 2020). Therefore, our data are in line with previous reports showing that S3 represent a small percentage of total CAFs (from 1-10% depending on the tumor stage) containing pericytes and detached perivascular fibroblasts-like cells (Davidson et al., Cell Reports 2020). Although representing a minor fraction of total CAFs, stromal cells of the perivascular niche have major roles in TME alterations and immunosuppression, notably by affecting perivascular TAMs and vascular function^{54,96}, which we now discuss.

These data are in line with previous reports showing that targeting major CAFs populations or common stromal pathways, such as aSMA, type I collagen or Hh signaling, actually promotes tumor growth (Rhim, Cancer Cell 2014; Ozdemir, Cancer Cell 2014; Lee, PNAS 2014; Jiang, JCI 2020; Chen, Cancer Cell 2021), as these cells have also homeostatic and protective roles essential for organ function. Importantly, ADAM12 expression in the stroma is specifically induced by the tumor, it is not present in the homeostatic stroma present in non-tumoral organs (shown in Fig. 1c and Fig. 5b).

8) ADAM12+ stromal cells localized near CD206+ TAMs and the authors infer an important role for this interaction. Yet, they fail to show if other CAF populations present at the tumor margin are in close contact with Axl+ TAMs.

We have shown that >80% of ADAM12+ cells are closely adjacent to P-Axl+ TAMs. We now analyzed the proximity of other CAF populations to P-Axl+ TAMs. Our data show that around 40% of DPP4+ CAFs (a marker for S1, Extended Data Fig 3b) and 20 % of aSMA+ CAFs can be found in proximity to Axl+ TAMs. These data show that although other CAFs populations have some level of proximity to P-Axl+ TAMs, it is significantly less compared to ADAM12+ cells (Extended Data Fig. 5a, $p < 0.0001$).

Importantly, we performed functional experiments to show that the interaction with ADAM12+ cells has an important role, as shown by their impact on TAMs polarization *in vivo* (Fig 4i,j), on Axl-dependent efferocytosis *in vitro* (Fig. 4f), and as the phenotype induced by depletion of ADAM12+ cells can be rescued by activating Axl (Fig 4k,l). We also show now that ADAM12+ cells are a major source for Gas6 in the tumor (Extended Data Fig 5g). Overall, these data are, in addition to co-localization, supporting an important role for ADAM12+ cells in TAMs function.

9) The evidence for ADAM12+ mesenchymal cell in controlling macrophage behavior, particularly efferocytosis, is largely based on in vitro observations and differences between experimental conditions are significant but marginal (Fig 4G). Authors should provide additional in vivo evidence supporting this effect.

To further investigate this question, we have performed several additional experiments, both *in vitro* and *in vivo*.

By optimizing conditions of the efferocytosis assay *in vitro* (as detailed in point 5, reviewer 1, page 2), we now obtained more clear differences between the different conditions, showing

that the CM from GFP+ cells promote efferocytosis in an Axl-dependent way, an effect that is not observed with the CM from GFP- cells (now in Fig 4f). We now further show that the CM from GFP+ cells induces expression of *Vegfa*, *Tgfb1* and *Il10* in macrophages undergoing efferocytosis *in vitro*, an effect that is not observed with the CM from GFP- cells (Extended Data Fig 5d). This is in line with the decreased expression of these genes *in vivo* in macrophages sorted from tumors depleted from ADAM12+ cells (Fig 4i,j).

We show additional *in vivo* evidence. Our current data show that macrophages isolated from tumors lacking ADAM12+ cells have decreased engulfing ability compared to macrophages isolated from WT tumors (Extended Data Fig 5f). We also show that the level of expression of *Gas6* in the tumor *in vivo* is significantly decreased when ablating ADAM12+ cells compared to WT tumors, showing that ADAM12+ cells are a major source for *Gas6* (Extended Data Fig 5g). Finally, activating the Axl pathway *in vivo* rescued the phenotype induced by depletion of ADAM12+ cells (Fig. 4k,l), and depletion of ADAM12+ cells induced a decrease in Axl-P TAMs and increase in *cas3*+ cells (Fig 4g,h). Overall, these data support the hypothesis that ADAM12+ cells have an impact on tumor macrophages and efferocytosis.

Reviewer #3

(Remarks to the Author)

Di Carlo et al. establish the relevance of a previously described but poorly understood population of ADAM12+PDGFR α + mesenchymal cells in immunity to tumors. The authors show how tumor conditions, especially macrophage-derived soluble factors, reactivate a developmental program in stromal cells characterized by the expression of ADAM12, a hypo proliferative state, and the production of factors that promote macrophage efferocytosis. The authors identify that these ADAM12+ stromal cells contribute to the immunosuppressive environment of tumors by promoting hypoxia and inducing macrophages to acquire a less inflammatory phenotype. The use of multiple models (both transplantable and orthotopic) is a strength of the paper that serves to bolster the author's description of ADAM12+ stromal cells' role in tumor immunity. They identify several factors, including TGF- β , IL-1 β , and OSM that contribute to acquisition of the ADAM12+ phenotype by stromal cells, and use fate-mapping techniques to demonstrate that ADAM12+ progenitors presage a peritumoral population of stromal cells that is maintained throughout the course of tumor progression.

The impact of this work is clear; the means by which stromal cells instruct macrophage (and subsequently T cell) function in tumors is poorly understood, and this study contributes to the field by identifying the instructive role of a unique stromal cell subset. However, some methodological concerns and unsupported claims dampen the enthusiasm of this reviewer.

1) Evidence of the functional role of ADAM12⁺ mesenchymal cells relies heavily on the finding that tumor growth is inhibited in ADAM12-DTR mice that lack ADAM12⁺ stromal cells upon treatment with diphtheria toxin. Attributing the entirety of the tumor growth phenotype to the activity of ADAM12⁺ stromal cells requires absolute specificity of ADAM12-DTR expression in stromal cells. This reviewer finds that the specificity of ADAM12 expression in ADAM12-DTR mice is incompletely explored in this study. While Fig. 1C does show that there is no expression of ADAM12-GFP in any CD45⁺ or CD31⁺ cells, this seems to potentially conflict with previous reports demonstrating ADAM12⁺ expression on T cells, including Tregs (i.e.: PMID 32572163, PMID 32572163, as well as examination of available of sequencing datasets). A more detailed gating strategy in the supplement with technical controls (i.e. fluorescence minus one control for GFP) would be helpful in this regard.

We thank the reviewer for all comments/questions. We agree that specificity of the transgene expression is key. To discriminate autofluorescence from GFP signal, we included non-transgenic GFP⁻ littermates in all our FACS experiments to correctly set up the positive gate for GFP⁺ cells. For more clarity, we now show gates for immune cells and endothelial cells with technical controls (GFP⁺ compared to GFP⁻) (Extended Data Fig 1e). Additional technical gates for GFP expression within T cell subsets are shown below.

These data confirm that we do not detect GFP expression in CD45⁺ cells in our model. Several reasons could account for the discrepancy relative to some reports in the literature, including different techniques of detection. As ADAM12 exist in a soluble form, which can bind on the cell surface notably by interaction with integrins (Eto, JBC 2000), it could be detected on cells that do not produce it. It is also possible that a certain level of ADAM12 is required to induce GFP expression in our model. In this case, our model would not target cells that express very low levels of ADAM12. In line with this hypothesis, we measured barely detectable levels of *Adam12* in CD45⁺ cells as compared to PDGFRa⁺ stromal cells in our model (qPCR for *Adam12* in the indicated population isolated from MO5 tumors is shown on the right). As CD45⁺ cells are not carrying the transgene in our model, it represents

an opportunity to target ADAM12+ stromal cells specifically. These data are consistent with our FACS data showing that effector T cells increase after depletion of ADAM12+ cells (Fig. 1f), and that Tregs remain unaffected (Extended Data Fig. 1j).

The lack of T cell depletion, and specifically Treg depletion, observed upon administration of diphtheria toxin in Fig. S1D would somewhat ameliorate these concerns, except that the proportion of Tregs among CD45+ tumor-infiltrating cells is extremely surprising and inconsistent with previous reports. Fig. S1D indicates that while ~10% of all CD45+ cells are CD4+ T cells, only ~0.3% of all CD45+ cells are FoxP3+ Tregs. This implies that only ~3% of all CD4+ T cells are FoxP3+ in tumors, which is at least an order of magnitude lower than what has been previously reported in B16 or B16-OVA tumors (Klages et al., Cancer Research, 2010; Magnuson et al., PNAS, 2018). Providing the gating strategy for flow cytometric analysis of these cells might help explain this discrepancy.

We thank the reviewer for pointing out this discrepancy. FACS analysis were performed at day14-15, when activated T cells infiltrate the tumor. As Tregs accumulate over tumor progression, we now analyzed FoxP3+ Tregs at later time points (d17-18) (this information has now been added in the method section). In these settings, we observed that up to 18% of CD4+ T cells are FoxP3+ in MO5 tumors (Extended Data Fig 1j, the gating strategy is shown below). Although additional variables can account for some variability compared to previous reports (such as tumor size, staining methods or use of reporter mice for FoxP3), these data confirmed that no significant differences in Tregs were observed in tumors depleted from ADAM12+ cells compared to WT tumors.

2) Additionally, the authors do not report on the efficiency of the depletion of ADAM12+ cells in their DTR model. This data is essential for evaluating the efficacy of the model and determining whether the observed changes in tumor growth are dependent on depletion of the cells of interest.

We previously reported specificity of ablation of ADAM12+ cells in our DTR depletion model. We used the lineage tracing model (ADAM12-tTA-Cre^{YFP} line, a scheme is provided in Fig 5d) as it expresses the DTR transgene and permanently marks all progeny as YFP. When depleting ADAM12+ cells by injecting DT in this model, we showed that the ADAM12-DTR model is efficient at > 97% to deplete ADAM12+ cells (as measured by loss of the progeny YFP+) while Pdpn+ YFP- stroma was not affected (Fig 4b in Dulauroy et al., Nature Medicine 2012).

To further investigate this point, we have analyzed expression of human *DTR* (*HBEGF*) by qPCR in MO5 tumors growing in ADAM12-DTR mice that are treated with DT or not, compared to WT tumors (DTR-). This approach is based on the fact that only ADAM12+ cells carry the human *DTR* transgene, while WT cells express the murine *Dtr*. We observed that, when ADAM12-DTR+ mice were treated with DT to deplete ADAM12+ cells, expression levels of human *DTR* in tumors was drastically reduced (similar to tumors in DTR- littermate) (Extended Data Fig.1g), confirming high efficiency of depletion.

3) One potential confounding factor of this model is the induction of stromal cell death by administration of DT. The authors demonstrate that ADAM12+ stromal cells were located adjacent to peritumoral blood vessels; thus, administration of DT and subsequent cell death is likely to alter the vasculature in a manner that is not necessarily dependent on the identity of the dying cell. It is plausible that the sensing of cell death increases permeability of peritumoral blood vessels and permits infiltration of T cells into the tumor as observed in Fig. 2A. For example, the upregulation of ICAM-1 that is attributed to normalization of vasculature in the absence of ADAM12+ stromal cells could also be attributed to inflammation secondary to death of perivascular cells. Thus, the contributions of cell death induced by DT vs. the lack of ADAM12+ stromal cells to the observed phenotype are extremely difficult to separate in this model.

We have now better described this point. Although ADAM12+ cells are perivascular, they are specifically found around blood vessels of the tumor margin (as shown in the high magnification image in Fig 1b). To clarify this point, we now provide a low magnification of tumors showing specific localization of ADAM12-GFP+ cells in the tumor margin (Extended Data Fig. 1b). Normalization of the vessels occurs throughout the tumor, in particular inside the tumor which is highly hypoxic when blood vessels are abnormal. To better show this, we have now added small magnifications of tumors with abnormal (Ctrl) or normalized blood vessels (DTR) in Extended Data Fig 2e (high magnifications of blood vessels in these conditions are shown in Fig. 2i,j). To further address this concern, we have performed immunofluorescence analysis of cleaved caspase 3 and the pericyte marker PDGFRb in tumors depleted of ADAM12+ cells. These data confirmed that casp3+ cells are not found in proximity to perivascular PDGFRb+ cells within the tumor when ADAM12+ cells are depleted (Extended Data Fig 5e, right panel).

The restricted localization of GFP+ cells in the tumor margin is a major argument pointing to the involvement of macrophages in the crosstalk. Macrophages accumulate at the tumor margins but also migrate inside the tumor. After depletion of ADAM12+ cells, we show that macrophages overexpress factors that normalize vessels, such as *Light*, and express less *Vegfa*, a major factor inducing abnormal vessels in tumors (Fig 4 j). Overall, this suggests that the crosstalk between ADAM12+ cells and TAMs is essential to relay signals from the margin to the total tumor. We now discuss these additional points.

If this is indeed the case, then it is difficult to determine whether ADAM12+ stromal cells are indeed a causative determinant of immunity to tumors, or rather a correlative indicator of an ongoing anti-tumor immune response. While the authors demonstrate that ADAM12+ stromal cells produce factors that alter macrophage phenotype and anti-tumor activity, they also demonstrate that they are also activated by IL-1 β and OSM and thus raise the possibility that their induction may be downstream of

macrophage activation. Thus, careful consideration on the interpretation of the findings is merited.

To address this concern, we performed additional experiments. These data show that the inflammatory cytokines Il1b and OSM, which are mostly produced by TAMs in our model, do not induce *Adam12* expression (Extended Data Fig 4c), in contrast to TGF-beta (Fig. 3g). This shows that, although Il1b and OSM modulate the phenotype of ADAM12⁺ cells (Fig. 3h), they are not sufficient to induce ADAM12⁺ cells. These data are consistent with previous reports showing that TGF-beta is the major inducer of *Adam12* (Atfi et al., 2007; Dulauroy et al., 2012; Veenstra et al., 2018). Our observation that several cell types within the TME, including non-immune cells, express TGF-beta (Extended Data Fig 4e,f) upon tumorigenesis further argues against induction of ADAM12⁺ cells downstream of macrophages activation. We propose that induction of ADAM12⁺ cells at early stages of tumorigenesis is a fundamental tissue repair mechanism initiated by cytostatic TGF-beta and modulated by inflammation, to promote angiogenesis and anti-inflammatory responses in coordination with macrophages.

4) While the growth-inhibitory role of TGF-β on stromal cells has been well-established, the mechanism of its specific induction of the development of ADAM12⁺ cells requires further investigation. The authors claim that the specific upregulation of Tgfbr3, but not Tgfbr1 or Tgfbr2, in ADAM12⁺ cells acts to enhance their TGF-β signaling and induce acquisition of the ADAM12⁺ phenotype. This claim needs to be validated by multiple methods beyond RNA sequencing results, including at the protein level.

We have now re-phrased our claim, and performed several additional experiments *in vivo* to clarify this point. TGFBR3 is a proteoglycan that has no kinase activity, although it can bind TGF-β ligands and may function as a TGF-β co-receptor to sequester and present TGF-β ligands to TGFBR2, or interacts with scaffold proteins (Blobe, 2001; Chen, 2003; Vander Ark, 2018). TGFBR3 can therefore modulate TGF-β signaling and responsiveness in the microenvironment. As it is expressed at high levels in ADAM12⁺ cells, we suggested that it may regulate TGF-β signaling locally. However, TGFBR3 is not the signaling receptor for TGF-β, and is unlikely responsible for ADAM12⁺ cells induction or phenotype.

We have now performed additional experiments *in vivo* to further investigate this point. Canonical TGF-β signaling occurs when TGF-β ligands binds to TGFBR2, which recruits and phosphorylates TGFBR1, activating downstream TGF-β signaling pathway. We show that ADAM12⁺ cells express both *Tgfbr1* and *Tgfbr2* (Extended Data Fig 4g,h), and the interaction between ADAM12 and TGFBR2 has been described *in vitro* (Atfi 2007; Gruel 2009). To investigate *in vivo* the role of *Tgfbr2* in our model, we have generated a tetracycline-regulated ablation model of *Tgfbr2* in ADAM12⁺ cells by crossing ADAM12-tTA mice to tet-controlled Cre (LC-1 mice) and *Tgfbr2*^{loxP/loxP} mice (ADAM12-tTA-Cre^{Tgfbr2} mice, Extended Data Fig. 4i). As TGFβ is produced at the onset of tumorigenesis (Extended Data Fig. 4e), we started ablating *Tgfbr2* in ADAM12⁺ cells at initiation of tumorigenesis in the MO5 model, and analyzed tumors after two weeks (experimental set-up is shown in Extended Data Fig. 4j). In these settings, we observed a significant decrease in MO5 tumor growth, as well as increased T cells infiltration, in ADAM12-tTA-Cre^{Tgfbr2} mice compared to WT littermate mice (Extended Fig. 4k,l), showing that TGFBR2 is required for the pro-tumorigenic role of ADAM12⁺ cells. Macrophages isolated from MO5 tumors from

ADAM12-tTA-Cre^{Tgfr2} mice expressed lower levels of *Vegfa*, a major inducer of leaky vessels, and tumor vessels had increased pericyte coverage (Extended Data Fig. 4m,n). Consistent with blood vessels normalization, stromal cells isolated from tumors growing in ADAM12-tTA-Cre^{Tgfr2} mice expressed higher levels of *Angpt1* and *Pdgfrb*, which stabilize the vasculature through pericyte-endothelial cells interaction⁷², and lower levels of *Car9*, induced by hypoxia⁵⁰, compared to stromal cells isolated from WT tumors (Extended Data Fig. 4o). Altogether, these data demonstrate that TGFBR2 signaling in ADAM12⁺ cells is required for their pro-tumorigenic function and TME alterations *in vivo*.

5) The authors go on to show expression of Axl in tumor-associated macrophages in close proximity to ADAM12⁺ stromal cells in Fig. 4, and infer a functional relationship between these cells. The specificity of Axl staining should also be explored here. Axl is also expressed by fibroblasts, and its signaling may directly affect fibroblast phenotype in addition to its role in macrophage-mediated efferocytosis. Given the results observed upon anti-Axl treatment later in the figure, Axl expression should be evaluated in non-macrophage populations, and a functional role for Axl in fibroblasts' acquisition of the ADAM12⁺ phenotype should also be explored.

Part of the question concerning the proximity of ADAM12⁺ cells to P-Axl⁺ macrophages and functional relationship has been answered in point 8 from reviewer 2 (p.10).

Concerning Axl staining: in images and quantifications reported in Fig. 4b,h,l, we have stained specifically for phospho-Axl (Y779), which was mostly detected in F480⁺ macrophages (>98%), not in fibroblasts. Therefore, Axl expressed on fibroblasts does not seem to be phosphorylated in our conditions. Nevertheless, to further investigate Axl signaling in fibroblast's phenotype, we have isolated by FACS fibroblasts from MO5 tumors and activated Axl *in vitro*. Results obtained by qPCR are shown in the graph below.

Although we observed increased expression of genes regulating proliferation, consistent with previous reports (Axelrod et al, 2014), we did not detect significant differences on expression of genes regulating efferocytosis and macrophages, including *Csf1*, *Gas6*, *Prosl*, *Lgals3*, or *Adam12* and other genes highly expressed by ADAM12⁺ cells. These data support the hypothesis of a direct effect of Axl activating antibodies on macrophages, and further argue against a functional role for Axl in fibroblasts acquisition of ADAM12⁺ phenotype. This is in line with the observation that ADAM12⁺ and ADAM12⁻ stromal cells express similar levels of Axl (graph on the right shows qPCR of *Axl* in ADAM12⁺ and ADAM12⁻ cells isolated from melanomas).

Overall, these data are consistent with the control experiment *in vivo* now added in Fig.4k (see also point 8 below). Indeed, the new data shows that anti-Axl treatment in control mice +MO5 does not have a pro-tumor effect by itself (Fig. 4k,l, Ctrl +a-Axl,) in contrast to the results obtained upon anti-Axl treatment in mice depleted from ADAM12+ cells (Fig. 4k,l, DTR +a-Axl).

6) The authors subsequently claim in Fig. 4G that Axl-sufficient, but not Axl-deficient, BMDMs exhibit increased efferocytic capacity in the presence of conditioned media from GFP+, stromal cells. The stated increase in efferocytosis of apoptotic MO5 cells by Axl-sufficient BMDM is unconvincing despite the observed statistical significance, especially given that there is no indication that this experiment was performed multiple times with similar results.

We apologize for the lack of details for the efferocytosis experiment, we unintentionally sent a version of methods/legends that was not fully updated. Both methods and legend are now updated and include all experimental details as required. The experiment was performed three times with similar results.

As detailed in previous comment, we have now optimized conditions for the efferocytosis experiments, leading to improved significance (new Fig. 4f). The effect of CM from GFP- cells is coherent with the observation that some levels of Gas6 are present in the CM of GFP- cells, although lower than in the CM of GFP+ cells (previous Fig. 4F). As the CM was collected after a few days in culture, we asked whether culture conditions might have affected this response. Indeed, we observed that Gas6 expression in stromal cells *in vitro* is rapidly increased in reduced serum conditions (Extended Data Fig 4d, note that in the first version the graph labels were inverted). By collecting CM in complete medium rather than in 2% serum, we now observed that GFP+ cells produced similarly high levels of Gas6, but level of Gas6 in GFP- cells were reduced (Fig. 4e), suggesting that they were artificially induced by culture conditions. By repeating the efferocytosis assay in these conditions, which are more representative of the unaltered state of GFP+ and GFP- cells *in vivo*, we observed improved specificity of GFP+ induction of efferocytosis compared to GFP- cells (new Fig 4f; statistical significance determined by two-way ANOVA, $p=0,0086$). Further supporting a specific role for GFP+ cells, we now show that the CM from GFP+ cells, but not the CM from GFP- cells, induce immunosuppressive / proangiogenic genes in efferocytic BMDM (Extended Data Fig. 5d), consistent with the *in vivo* data (Fig. 4j). Induction of efferocytosis by the CM of GFP+ cells was abrogated in BMDM lacking Axl (Fig 4f, $p<0.001$), showing that induction of efferocytosis by GFP+ cells requires Axl. We now discuss that GFP+ cells produce high levels of Axl ligands such as Gas6, and promote macrophage efferocytosis in an Axl-dependent way.

To further strengthen this claim, we performed additional experiments *in vivo*. We now show that Gas6 expression is significantly decreased in tumors depleted from ADAM12+ cells, compared to WT tumors, showing that ADAM12+ cells are a major source for Gas6 within the TME *in vivo* (Extended Data Fig. 5g). We further show that macrophages isolated from tumors lacking ADAM12+ cells (which contained less Gas6 as shown in Extended Data Fig. 5g) have decreased engulfing ability (Extended Data Fig. 5f), which is in line with their shift toward M1 phenotype (Fig. 4i,j).

Statistical analysis with correction for multiple comparisons is appropriate in this case, and results may no longer be significant when the appropriate statistical test is applied.

Statistical analysis on the revised version has been performed using a two-way ANOVA test, ($p < 0.001$ for GFP+ vs control and $p < 0.01$ for GFP+ vs GFP-). The complete description for statistical analysis is now provided in the legend and methods.

Furthermore, there is no indication in the methods section or otherwise as to how this efferocytosis experiment was performed. The method of inducing apoptosis in MO5 cells, the ratio of BMDM to apoptotic MO5 cells, the fluorescent marker used to measure efferocytosis, and the baseline for the fold increase on the y-axis of Fig. 4G are all critical pieces of information in evaluating the validity of this experiment, and are indicated nowhere in the text.

We apologize for the lack of details for the efferocytosis experiment, we unintentionally sent a version of methods/legends that was not fully updated. Both methods and legend are now updated and include all experimental details as required (detailed in the methods p.32-33)

In addition to efferocytosis by macrophages, stromal cells have also been demonstrated to have efferocytic functions. The efferocytosis of apoptotic cargo by stromal cells should also be evaluated in this context; the relative efferocytic activity of ADAM12+ vs. ADAM12- cells may be instructive in their acquisition of their respective phenotypes.

We have now evaluated the efferocytic capacity of stromal cells isolated from tumors and tested in the same conditions than BMDM, or macrophages isolated from tumors. The data show that, in contrast to BMDM and macrophages isolated from MO5 tumors and incubated with apoptotic cells, total stromal cells isolated from MO5 tumors have no efferocytic capacity (Extended Data Fig. 5c). Efferocytosis is therefore unlikely determinant in the acquisition of ADAM12+ and ADAM12- respective stromal phenotype in this model.

7) The authors report that cleaved caspase-3-expressing cells are more abundant in the tumors of ADAM12-DTR mice treated with DT, and conclude that this is a result of decreased efferocytosis in the absence of ADAM12+ stromal cells. However, increased cleaved caspase-3 staining may also be explained by increased killing of tumor cells in DT-treated mice independent of any effect on efferocytosis. Thus, this finding may simply be correlate of a generally enhanced anti-tumor immune response in ADAM12-DTR mice upon treatment with DT.

We agree that staining for casp3 alone might not discriminate between these two situations. To further investigate whether macrophages have decreased efferocytosis *in vivo*, which would result in accumulation of casp3+ cells, we have now performed additional experiments. We now report that macrophages isolated from tumors lacking ADAM12+ cells have decreased engulfing capacity compared to macrophages isolated from WT tumors (Extended Data Fig. 5f). Consistent with this observation, we observed that tumors lacking ADAM12+ cells have decreased levels of Gas6 compared to WT tumors (Extended Data Fig. 5g), further showing that ADAM12+ cells are an important source for Gas6 in the tumor. Finally, we have performed additional experiments/quantifications for cleaved-caspase3

staining, which confirmed our initial observation. Altogether, these additional data support the hypothesis that ADAM12⁺ cells play an important role in macrophage efferocytosis.

8) The dependency of the observed anti-tumor effect of ADAM12⁺ cell depletion on decreased efferocytosis was evaluated using concurrent treatment of ADAM12-DTR mice with an anti-Axl agonist antibody. The authors conclude that depletion of ADAM12⁺ cells promotes tumor immunity through Axl-dependent effect on efferocytosis because of the observed restoration of tumor growth in DT-treated mice also treated with the anti-Axl antibody. However, signaling through Axl directly on tumor cells has also been demonstrated to elicit potent pro-tumorigenic effects. Thus, the dependency of the observed phenotype on the inhibition of efferocytosis is still unclear. Treatment of ADAM12-DTR mice with anti-Axl in the absence of DT administration will be a critical control to resolve this question.

To investigate whether Axl signaling has direct pro-tumorigenic effects, we have now treated Ctrl mice + DT with anti-Axl agonist antibody. Note that to be comparable with the control condition lacking anti-Axl treatment (and be consistent throughout the article), we injected DT in Ctrl mice (DTR-), rather than not administering DT in ADAM12-DTR mice. We now confirmed that activating Axl alone has no pro-tumorigenic effect when ADAM12⁺ cells are not depleted in our model (Extended Data Fig 4k, Ctrl+Axl).

9) This reviewer does not have the expertise to evaluate the claims of clinical relevance in Figure 6. However, it should be noted that this analysis does not distinguish between ADAM12 expression on stromal cells vs. tumor cells, which complicates interpretation of these results.

To strengthen this point, we have now analyzed *ADAM12* expression in single cells RNAseq of human melanoma, pancreatic cancer and colorectal cancer, which showed high expression of *ADAM12* preferentially in stromal populations within the TME (Extended Data Fig. 8a-e). In Extended Data Fig 8a and 8c, tumor cells are included in the scRNAseq dataset, showing very low expression of *ADAM12* in tumor cells compared to stromal cells. Expression of *ADAM12* in malignant melanoma cells (from scRNAseq dataset shown in Extended Data Fig 8b) are shown in the boxplot below. These data further show that out of 14 tumors, only one express significant level of *ADAM12* in melanoma cells (graph below). Overall, these data show that, for a majority of tumors, *ADAM12* expression in tumor cells is rather low compared to stromal cells within the TME.

Consistent with this, we now show that *ADAM12* expression correlates with the “activated stroma” subtype of human PDAC, which is associated with a severe prognosis (as described

in Puleo et al, Gastroenterology 2018), and that ADAM12+ stromal cells in human PDAC are mainly found within the PDPN+PDGFRA+ populations (Extended Data Fig. 8d,e), as observed in mice models. These data are consistent with previous literature, including at the single cells level, showing that ADAM12 is mainly expressed by stromal cells in several human tumors including liver, PDAC and colorectal cancer (Le Pabic, Hepatology 2003; Dominguez, Cancer Discovery 2020; Hoorn, BMC Cancer 2022). ADAM12 has also been reported as a marker for stromal activation in several cancers, including human prostate cancer and pancreatic cancer (Veenstra et al., 2018; Bilgin Dogru 2014; Bacalod et al., 2021; Wilkinson et al., 2013), is and associated to an unfavorable outcome. We now added this additional information in the article.

Rigor

This manuscript requires a much more detailed description of methods in many cases, and the lack of methodological details often obscures interpretation of the data. For example, several figures reference the tumor margin and quantify infiltration across the margin, but there is no description of how the margin is determined.

We thank the reviewer for pointing out the need for additional methodological details. We have now added more information in the legends and in the method section in several cases. Tumor margin was identified as a peritumoral zone with high density of Pdpn⁺ or aSMA⁺ stromal cells (as shown in Extended Data Fig 6a,b, Extended Data Fig 1b and Fig 5f). This information is now added in the method/legends.

Representation of tumor growth data should also be more comprehensive. Fig. 1D shows average growth plots, but spider plots of tumor growth for each individual mouse should also be represented so that variability can be adequately assessed.

We now have represented spider plots of tumor growth for each individual mouse alongside growth curves, in Fig 1d as well as in all other graphs representing tumor growth, so to better assess variability.

Both average growth and individual spider plots should also be represented alongside Fig. 4L. Additionally, tumor growth inhibition (TGI) score is shown in Fig. 1E and Fig. 4L, but there is no reference to or description of how this metric is calculated or any statistics to ascertain significance.

We now have added individual spider plots alongside growth curves in the current Fig 4k, and, to standardize the graphs, also in all other figures representing tumor growth curves, which are now shown in Extended Data Fig 1h,i,m and Fig 4k. As they do not provide any additional information, we have now removed TGI graphs.

Furthermore, tumor growth is evaluated in Fig. S1E using a different metric than the TGI score shown in Fig. 1E and Fig. 4L. Tumor growth in Fig. S1E appears to be evaluated at a much later timepoint than Fig. 1D-E, as the tumor volume of the control + DT group is approximately double the average at Day 18 in Fig. 1D. The day at which this analysis was performed is not stated in the text or figure. These metrics should

remain consistent throughout the manuscript to facilitate comparison across figures, and to evaluate the claim of dependence on CD8+ T cells in Fig. S1E.

To facilitate comparison across figures, we now show anti-CD8 treatment at a similar day of analysis/tumor volume (Extended Data Fig. 1m), and represented both average tumor volume and individual animal growth curve, as shown in Fig 1.

The lack of EdU incorporation by ADAM12+ stromal cells in Fig. 3E is supported by RNA sequencing results, but a side-by-side comparison of EdU incorporation of Pdpn+ADAM12+ vs. Pdpn+ADAM12- stromal cells would be the best validation of their slow-cycling phenotype. This can be calculated from the existing images in Fig. 3E.

We now quantified proliferation in Pdpn+ADAM12+ cells, Pdpn+ADAM12- cells, and aSMA+ADAM12- cells. The data confirmed that both ADAM12- stromal populations proliferate significantly more compared to ADAM12+ cells ($p < 0.01$ and $p < 0.0001$ for Pdpn+ and aSMA+ cells, respectively, now shown in Fig 3e)

Figure 4B indicates CD11b staining in red, GFP expression in green, and F4/80 staining in blue; however, the inlet shows P-Axl expression in red. This appears to be an inadvertent substitution of CD11b for P-Axl staining in red.

They were indeed two different staining. To improve clarity, we show now both staining in different panels in Fig. 4b, and the quantification of GFP+ distance to macrophages is now done to P-Axl+ macrophage, as described in the legend.

Statistical analysis also lacks rigor in many cases; every comparison utilizes unpaired student's t-test with no correction for multiple comparisons even though it is an inadequate test in several panels.

We have now corrected when necessary and confirmed that statistical significance is maintained. All details for statistical tests are now indicated in the corresponding legend, as well as in the method section.

Decision Letter, first revision:

6th Jun 2023

Dear Lucy,

Thank you for providing a point-by-point response to the remaining comments voiced by referee #2 on your manuscript entitled, "Depletion of slow-cycling PDGFRa+ADAM12+ mesenchymal cells promotes antitumor immunity by restricting macrophage efferocytosis". As noted previously, referees #1 and #3 were largely satisfied by the revised manuscript. We are very interested in the possibility of publishing your study in Nature Immunology, but would like to have you add the additional clarifications and data that are already in hand.

We invite you to submit a revised manuscript along the lines indicated in your response rebuttal.

Specifically, the revision should include the experimental data to address:

- (1) qPCR of Adam12 in GFP+/- cells
- (2) include the phenotypic analysis for pericyte and mural cells in the CAF2/3 cell subsets
- (3) please examine ADAM12 expression by histology & examine if colocalized with pericytes
- (4) stain for Adam12 expression by histology and/orRNAscope in DTR-treated tumors
- (5) include as a discussion point Adam12+ vs Lrrc15+ CAFs
- (6) discuss CAF abundance in late-stage tumors
- (7) Clarify Ltrb expression in CAFs

Please include the additional textual clarifications as indicated in your response letter.

When you revise your manuscript, please take into account all reviewer and editor comments, please highlight all changes in the manuscript text file in Microsoft Word format.

* If you have not done so already please begin to revise your manuscript so that it conforms to our

Article format instructions at <http://www.nature.com/ni/authors/index.html>. Refer also to any guidelines provided in this letter.

* Please include a revised version of any required reporting checklist. It will be available to referees to aid in their evaluation of the manuscript goes back for peer review. They are available here:

Reporting summary:

[REDACTED]

We hope to receive your revised manuscript within four weeks. If you cannot send it within this time, please let us know. We will be happy to consider your revision so long as nothing similar has been accepted for publication at Nature Immunology or published elsewhere.

Nature Immunology is committed to improving transparency in authorship. As part of our efforts in this direction, we are now requesting that all authors identified as 'corresponding author' on published papers create and link their Open Researcher and Contributor Identifier (ORCID) with their account on the Manuscript Tracking System (MTS), prior to acceptance. ORCID helps the scientific community achieve unambiguous attribution of all scholarly contributions. You can create and link your ORCID from the home page of the MTS by clicking on 'Modify my Springer Nature account'. For more information please visit www.springernature.com/orcid.

Kind regards,

Laurie

Laurie A. Dempsey, Ph.D.
Senior Editor
Nature Immunology
l.dempsey@us.nature.com
ORCID: 0000-0002-3304-796X

Reviewers' Comments:

Reviewer #1:

Remarks to the Author:

Di Carlo et al. describe a small subset of ADAM12+PDGFRa+ mesenchymal stem cell in early tumor development in mouse models of melanoma, pancreatic cancer, and prostate cancer. This population produces important microenvironment factors including Gas6, Lgals3, Has2, and Csf1 and is induced by TGF-beta. In this revision the authors have done considerable additional work and I have no further comments/questions.

Reviewer #2:

Remarks to the Author:

The manuscript incorporates significant improvements, with the authors including new data, enhanced images, and better-quality figures. The newly presented data convincingly support the notion that Adam12-DTR+ CAFs, rather than FRCs in draining lymph nodes, impact tumor growth, T cell recruitment, and macrophage biology. However, there are several unresolved questions that hinder definitive conclusions regarding whether these functions are specific to Adam12+ CAFs or if they can be exerted by other CAF populations.

Firstly, further clarification is needed regarding the characterization of CAF subsets and the analysis of the efficiency and specificity of the Adam12-DTR strain. Although the authors utilized the dataset from Davidson et al., which demonstrated Adam12 mRNA expression in nearly all cells within the CAF2 and CAF3 subsets at both day 5 and day 11 (Extended Data Fig. 3e), FACS analysis of Adam12-GFP expression revealed low targeting of these populations (5.2%)(Extended Data Fig. 3f). Additionally, it is puzzling that the proportions of the two targeted clusters increase when Adam12-DTR+ cells are ablated (Extended Data Fig. 3e), raising questions about cell cluster definitions. Moreover, the observation of increased pericyte abundance is somewhat problematic as pericytes also appear to express Adam12 mRNA. Another issue is that the authors did not use an equivalent FACS approach during the lineage tracing experiment to elucidate the ontogenetic relationship between CAF2 and CAF3 (Extended Data Fig. 7b).

The authors should perform single-cell RNA sequencing (scRNAseq) of the tumor microenvironment (TME) and assess CAF diversity before and after ADAM12+ cell ablation to demonstrate the strain's efficiency and specificity, as I previously requested. Given the similarities between this manuscript and

a recently published paper by Krishnamurty et al. (Nature 2022), which describes the role of Lrrc15+ CAFs and TGF-beta signaling in tumor growth and T cell infiltration, these aspects are particularly relevant. Additionally, it remains unclear why the ADAM12-GFP+ population disappears, especially if the population remains present in late-stage tumors but is not targeted by this murine strain. To address this question, the authors should consider performing scRNAseq on late-stage tumors since the dataset from Davidson et al. does not include those samples.

The inclusion of new experiments using LTBR-DTR contributes to supporting some of the conclusions (Fig. 1 for the reviewers). However, the data actually shows that LTBR is expressed by the same CAF clusters as the Adam12-DTR, yet no effect on tumor growth is detected. My concern here is once again the limited definition of the cell targeting of these strains, which precludes robust conclusions about the mechanism exerted by Adam12+ cells to regulate the immune environment. Another caveat is the difference in the experimental setup used in LTBR-DTR mice compared to the Adam12-DTR strain; it seems that the mice are treated only on day 10 instead of daily from day 10 to day 18.

The authors have also confirmed the effect of Adam12+ ablation in one of the spontaneous models, showing a reduction in tumor growth and T cell recruitment. However, they do not clarify if there is a common mechanism with the melanoma model. Specifically, they should investigate whether pericytes and/or macrophages are affected in this model as well.

In their attempt to identify the effector population responsible for immune evasion, the authors compared GFP-positive progenitor cells and their differentiated YFP progeny. They argue that both ADAM12+ progenitors and their differentiated progeny promote immunosuppression. However, most of the factors analyzed exhibit higher expression in GFP+ progenitor cells (Ext. Data Fig. 7c), implying a key role for this population during immune evasion. To address this concern, the authors should provide a more detailed profile of each cell population, ideally by performing RNAseq on YFP and GFP-sorted CAFs.

Minor comments:

- Authors have provided data on total number of targeted cells in the lineage tracing analysis, but they haven't included measurements of clone sizes.

Reviewer #3:

Remarks to the Author:

Di Carlo et al convincingly demonstrate a critical role for ADAM12+ stromal cells in driving tumor immunity. Through mechanistically exploring their development and localization, describing their unique properties in relation to ADAM12- stromal cells, and detailing the functional consequences of their interactions with macrophages and other immune cells in the context of cancer, the authors underscore the importance of a thorough understanding of stromal cells in cancer biology. While the manuscript was strong upon its initial submission, clarification of key conclusions and some methodological concerns have noticeably strengthened the authors' claims.

It should be noted that there are still a few errors in the manuscript.

- In the phagocytosis assay in Fig. 4F, the graph legend denotes "CM GFP+ cells" in blue and "CM

GFP- cells" in green. Based on the description in the text and the elevated Gas6 expression in GFP+ cells demonstrated in Fig. 4d-e, phagocytosis should be elevated upon exposure to CM from GFP+ cells, but not GFP- cells. It is likely these labels have been switched by mistake. It might help further clarify if the same color of bars were used for GFP+ and GFP- for each panel in Fig. 4d-f.

- There are several typos and misspellings/grammatical errors throughout the manuscript:
 - o Line 72- "... has been associated to resistance to chemotherapies..."
 - o Line 165- "... single cells RNAseq..."
 - o Line 448- "Trough engulfment of apoptotic cells..."
 - o Line 452- "... capacity to phagocyte and expressed..."
 - o Line 453- "...as well as proangiogenic vegfa..."

Author Rebuttal, first revision:

See inserted PDF

Reviewer #1

(Remarks to the Author)

Di Carlo et al. describe a small subset of ADAM12+PDGFRA+ mesenchymal stem cell in early tumor development in mouse models of melanoma, pancreatic cancer, and prostate cancer. This population produces important microenvironment factors including Gas6, Lgals3, Has2, and Csf1 and is induced by TGF-beta. In this revision the authors have done considerable additional work and I have no further comments/questions.

Thank you for all comments/questions.

Reviewer #2

(Remarks to the Author)

The manuscript incorporates significant improvements, with the authors including new data, enhanced images, and better-quality figures. The newly presented data convincingly support the notion that Adam12-DTR+ CAFs, rather than FRCs in draining lymph nodes, impact tumor growth, T cell recruitment, and macrophage biology. However, there are several unresolved questions that hinder definitive conclusions regarding whether these functions are specific to Adam12+ CAFs or if they can be exerted by other CAF populations.

Firstly, further clarification is needed regarding the characterization of CAF subsets and the analysis of the efficiency and specificity of the Adam12-DTR strain. Although the authors utilized the dataset from Davidson et al., which demonstrated Adam12 mRNA expression in nearly all cells within the CAF2 and CAF3 subsets at both day 5 and day 11 (Extended Data Fig. 3e), FACS analysis of Adam12-GFP expression revealed low targeting of these populations (5.2%)(Extended Data Fig. 3f).

Thank you for all comments/questions. To clarify this point, we performed detailed analysis of ADAM12 expression. At day 5, the number of CAF3 analyzed in the dataset from

Davidson et al. is very low (12 cells), rising questions about significance. At day 8, which is closer to our analysis time, only a fraction of CAF2 and CAF3 expressed *Adam12*, as at day 11 (Extended data Fig 3e). By re-analyzing our FACS data with a gating strategy more similar to Davidson et al (notably by gating specifically on S1-2 and S3), we obtain a percentage of GFP+ cells comparable with Davidson dataset (FACS plots on the left). Further differences

in cell isolation protocol and viability might account for additional discrepancies (as Davidson et al report low recovery of stromal cells compared to our protocol).

To further address this concern, we analyzed expression of *Adam12* by qPCR in GFP+ compared to GFP- stromal cells. If the reporter mice were not targeting all ADAM12 expressing cells, the GFP- cells would be expected to have high level of *Adam12* expression, which is not the case (now shown in Extended Data Fig 1e).

Additionally, it is puzzling that the proportions of the two targeted clusters increase when Adam12-DTR+ cells are ablated (Extended Data Fig. 3e), raising questions about cell cluster definitions. Moreover, the observation of increased pericyte abundance is somewhat problematic as pericytes also appear to express Adam12 mRNA

We defined cell clusters according to expression of surface markers based on several published reports. Furthermore, we confirmed cluster definition by performing qPCR of key marker genes on CAF1, CAF2 and CAF3 isolated by FACS from the tumor (Extended Data Fig 3a and 3c). The relative increase in CAF2 and CAF3 by FACS is consistent with increased pericyte coverage, confirmed by histology as well as at the functional level (increased perfusion), that we observed in melanoma after ADAM12+ cells depletion. We now show data confirming a similar phenotype in spontaneous RIP-Tag tumors after depletion of ADAM12+ cells, further supporting this data (Extended Data Fig. 7l-n).

Mural cells in tumors are known to be heterogeneous, and include “healthy” pericytes (which are embedded within the vascular basement membrane and therefore functional) as well as “pathological” pericyte-like cells that are detached from the blood vessels. Analysis of the dataset from Davidson et al confirmed some heterogeneity within CAF2 and CAF3

populations with respect to pericyte markers (figure above). However, whether these populations are functional or not (healthy vs pathological) cannot be discriminated in scRNAseq data.

To further investigate the relationship of ADAM12+ cells and pericytes, we performed detailed immunofluorescence analysis on tumor sections. This data shows that ADAM12+ cells are localized outside of the vascular basement membrane (ColIV+), in contrast to normal pericytes (Extended Data Fig 1h,i). Overall, our data are consistent with ablation of “pathological”- pericyte-like cells, which are detached from vessels, but not “healthy” pericytes (ADAM12⁻) which are required for vascular normalization.

Finally, the relative increase in CAF2 and CAF3 is consistent with the observed decrease in CAF1, due to restored normoxia following depletion of ADAM12+ cells. This data are in line with recent publications showing that CAF1 accumulate specifically in hypoxic tumor regions (Schwoerer et al, 2023; Mello et al, 2022).

Another issue is that the authors did not use an equivalent FACS approach during the lineage tracing experiment to elucidate the ontogenetic relationship between CAF2 and CAF3 (Extended Data Fig. 7b).

We choose to show histology of YFP+ cells as it is the gold standard to characterize and discriminate between healthy and pathological (“detached”) pericytes. This requires combination of several markers and position to the vascular basement membrane, as pathological pericytes are localized outside the vascular basement membrane. This data confirmed that, although YFP+ cells express some levels of pericyte markers, they are, as GFP+ cells, localized outside the vascular basement membrane. This point is key to our message and cannot be discriminated by FACS.

The authors should perform single-cell RNA sequencing (scRNAseq) of the tumor microenvironment (TME) and assess CAF diversity before and after ADAM12+ cell ablation to demonstrate the strain's efficiency and specificity, as I previously requested.

To demonstrate the strain's efficiency and specificity, we used different approaches and experimental settings, including restoration of WT levels of DTR (Extended Data Fig1j), loss of lineage tracing after depletion (Dulauroy, Nature Medicine 2012), and provided evidence that other CAFs populations are still present in depleted tumors. CAF diversity before and after ADAM12+ cell depletion was addressed by FACS, as previous reports indicated low CAF diversity in this same model at our time of analysis, and as several surface markers were available to identify these populations. Importantly, we further confirmed identity of the CAF subsets by transcriptome analysis on sorted cell populations (Extended Data Fig. 3).

To further address this concern, and to formally show depletion of ADAM12+ cells in the DTR model after depletion, we now report analysis of *Adam12* expression by RNAscope in tumors before and after depletion. This experiment demonstrates lack of *Adam12* expression in depleted tumors compared to WT tumors (Extended Data Fig. 1k, $p < 0.001$), confirming that ADAM12 expressing cells are not found within the TME after depletion using the DTR model. Concerning the reporter mice, we now show specificity of *Adam12* expression in GFP+ cells compared to GFP- stromal cells (now shown in Extended Data Fig 1e), and re-analysis by FACS of GFP+ cells (reported in page 1). Technical controls for GFP were shown previously (Extended Data Fig 1f, and data shown page 12 from previous point by point response). Altogether, these data point to high efficiency and specificity of the strains used.

Given the similarities between this manuscript and a recently published paper by Krishnamurty et al. (Nature 2022), which describes the role of Lrrc15+ CAFs and TGF-beta signaling in tumor growth and T cell infiltration, these aspects are particularly relevant.

We further investigated this point. Our analysis indicates that LRRC15+ cells are quite rare in melanoma (see tSNE plot below) as well as in prostate cancer, in contrast to ADAM12+

cells. In pancreatic cancer scRNAseq data, *Adam12* is not expressed in the *Lrrc15*+ CAF subset (Dominguez et al, Cancer Discov 2020), consistent with the observation that GFP+ cells and LRRC15+ cells express different markers (Krishnamurti et al, Nature 2022; Dominguez et al, Cancer Discov 2020). Confirming this data, we now show that GFP- CAFs express high levels of *Lrrc15* compared to GFP+ cells (Extended data Fig 4a).

To further address this point, we now also report that ADAM12+ cells are not localized close to T cells (shown in Extended Data Fig 5a). This is in contrast to *Lrrc15*+ CAFs, which were described to be in close proximity to tumour-infiltrating T cells, and to suppress tumor immunity by direct inhibition of T cells (Krishnamurti et al, Nature 2022). Overall, this data suggests that LRRC15+ and ADAM12+ are different stromal subsets promoting tumorigenesis through different mechanisms, although both dependent on TGF-beta. We now discuss this point.

Additionally, it remains unclear why the ADAM12-GFP+ population disappears, especially if the population remains present in late-stage tumors but is not targeted by this murine strain. To address this question, the authors should consider performing scRNAseq on late-stage tumors since the dataset from Davidson et al. does not include those samples.

We show that ADAM12+ cells are abundant in early-stage tumors (Fig 1c, Fig 5g), and further demonstrates by inducible fate mapping that the ADAM12-GFP+ population is still present in late-stage tumors (Fig 5e and 5g, condition “late”), which is consistent with previous reports.

The inclusion of new experiments using LTBR-DTR contributes to supporting some of the conclusions (Fig. 1 for the reviewers). However, the data actually shows that LTBR is expressed by the same CAF clusters as the Adam12-DTR, yet no effect on tumor

growth is detected. My concern here is once again the limited definition of the cell targeting of these strains, which precludes robust conclusions about the mechanism exerted by Adam12+ cells to regulate the immune environment.

In contrast to *Adam12*, *Ltbr* is expressed by all CAF subsets, and notably by CAF1 which shows almost no expression of *Adam12*. This is a major difference between these two stromal clusters. Although further investigations are required to understand differences between LTBR+ and ADAM12+ stromal subsets (notably as CAF clusters seem to be quite heterogeneous), our results are consistent with previous data showing that depletion of a stromal cells expressing *Ccl19* (which is regulated by LTBR) does not decrease tumor growth (Cheng et al., 2018).

Another caveat is the difference in the experimental setup used in LTBR-DTR mice compared to the Adam12-DTR strain; it seems that the mice are treated only on day 10 instead of daily from day 10 to day 18.

We apologize for the lack of information on the figure. LTBR-DTR were treated exactly as ADAM12-DTR, DT injected daily from day 10-18.

The authors have also confirmed the effect of Adam12+ ablation in one of the spontaneous models, showing a reduction in tumor growth and T cell recruitment. However, they do not clarify if there is a common mechanism with the melanoma model. Specifically, they should investigate whether pericytes and/or macrophages are affected in this model as well.

To investigate this point, we have performed additional analysis, that are now shown in Extended Data Fig 7h-n. Our data show that pancreatic tumors in the spontaneous RipTag model have improved perfusion, better infiltration of T cells and of Pdpn+ stroma within the tumor center, as well as normalized blood vessels with increased pericyte coverage and ICAM1 expression after ADAM12+ cells depletion compared to WT tumors. Overall, this data supports the hypothesis of a common mechanism in the spontaneous tumor model with the melanoma model.

In their attempt to identify the effector population responsible for immune evasion, the authors compared GFP-positive progenitor cells and their differentiated YFP progeny. They argue that both ADAM12+ progenitors and their differentiated progeny promote immunosuppression. However, most of the factors analyzed exhibit higher expression in GFP+ progenitor cells (Ext. Data Fig. 7c), implying a key role for this population during immune evasion. To address this concern, the authors should provide a more detailed profile of each cell population, ideally by performing RNAseq on YFP and GFP-sorted CAFs.

We performed RNAseq on GFP+ sorted cells (Fig. 3). As the point was to compare GFP+ to YFP+ cells, we performed qPCR on YFP+ cells for most genes that were upregulated in GFP+ cells. To strengthen this point, we now show additional gene analysis in Extended Data Fig. 7c. We fully agree that most immune mediators are highly expressed in GFP+ cells compared to YFP+ cells, showing that GFP+ cells are a key population in immune evasion. In addition, YFP+ cells express higher levels of other molecules involved in

immunosuppression as well, such as Car9 and Angpt2, overall suggesting a “broad” immunosuppressive phenotype for the ADAM12 lineage. We discuss this point.

Minor comments:

Authors have provided data on total number of targeted cells in the lineage tracing analysis, but they haven't included measurements of clone sizes.

Unfortunately, identification of clone size was not feasible *in vitro* as cells rapidly lose their progenitor potential (as detailed in point 4, page 8 in the previous point-by-point response). Furthermore, it was not possible to discriminate between different clones *in vivo* in this model as all clones are YFP+.

Reviewer #3

(Remarks to the Author)

Di Carlo et al convincingly demonstrate a critical role for ADAM12+ stromal cells in driving tumor immunity. Through mechanistically exploring their development and localization, describing their unique properties in relation to ADAM12- stromal cells, and detailing the functional consequences of their interactions with macrophages and other immune cells in the context of cancer, the authors underscore the importance of a thorough understanding of stromal cells in cancer biology. While the manuscript was strong upon its initial submission, clarification of key conclusions and some methodological concerns have noticeably strengthened the authors' claims.

It should be noted that there are still a few errors in the manuscript.

• In the phagocytosis assay in Fig. 4F, the graph legend denotes “CM GFP+ cells” in blue and “CM GFP- cells” in green. Based on the description in the text and the elevated Gas6 expression in GFP+ cells demonstrated in Fig. 4d-e, phagocytosis should be elevated upon exposure to CM from GFP+ cells, but not GFP- cells. It is likely these labels have been switched by mistake. It might help further clarify if the same color of bars were used for GFP+ and GFP- for each panel in Fig. 4d-f.

Thank you for all comments/questions. We modified the colors as suggested and corrected the switched labels.

• There are several typos and misspellings/grammatical errors throughout the manuscript:

- o Line 72- “... has been associated to resistance to chemotherapies...”**
- o Line 165- “... single cells RNAseq...”**
- o Line 448- “Trough engulfment of apoptotic cells...”**
- o Line 452- “... capacity to phagocyte and expressed...”**
- o Line 453- “...as well as proangiogenic vegfa...”**

Thank you, these errors have been corrected.

Decision Letter, second revision:

2nd Aug 2023

Dear Lucie,

Thank you for submitting your revised manuscript "Depletion of slow-cycling PDGFRa+ADAM12+ mesenchymal cells promotes antitumor immunity by restricting macrophage efferocytosis" (NI-A34258B). As noted in my previous e-mail message, we have been chasing referee #2, but we have not heard back from this individual. Given the length of time they have had access to the revised manuscript and that the other two referees had endorsed publication of the study, we feel confident to go forward without referee #2 final comments. Therefore we'll be happy in principle to publish it in Nature Immunology, pending minor revisions to comply with our editorial and formatting guidelines.

We will now perform detailed checks on your paper and will send you a checklist detailing our editorial and formatting requirements in about a week. Please do not upload the final materials and make any revisions until you receive this additional information from us.

We will need a Word file for the current version of the manuscript to begin the editing process (the source file we have is a PDF); please email that to immunology@us.nature.com at your earliest convenience.

Thank you again for your interest in Nature Immunology. Please do not hesitate to contact me if you have any questions.

Kind regards,

Laurie

Laurie A. Dempsey, Ph.D.
Senior Editor
Nature Immunology
l.dempsey@us.nature.com
ORCID: 0000-0002-3304-796X

Final Decision Letter:

Dear Lucie,

I am delighted to accept your manuscript entitled "Depletion of slow-cycling PDGFRa+ADAM12+ mesenchymal cells promotes antitumor immunity by restricting macrophage efferocytosis" for publication in an upcoming issue of Nature Immunology.

Over the next few weeks, your paper will be copyedited to ensure that it conforms to Nature Immunology style. Once your paper is typeset, you will receive an email with a link to choose the appropriate publishing options for your paper and our Author Services team will be in touch regarding any additional information that may be required.

Please note that *Nature Immunology* is a Transformative Journal (TJ). Authors may publish their research with us through the traditional subscription access route or make their paper immediately open access through payment of an article-processing charge (APC). Authors will not be required to make a final decision about access to their article until it has been accepted. [Find out more about Transformative Journals](https://www.springernature.com/gp/open-research/transformative-journals).

Authors may need to take specific actions to achieve [compliance with funder and institutional open access mandates](https://www.springernature.com/gp/open-research/funding/policy-compliance-faqs). If your research is supported by a funder that requires immediate open access (e.g. according to [Plan S principles](https://www.springernature.com/gp/open-research/plan-s-compliance)) then you should select the gold OA route, and we will direct you to the compliant route where possible. For authors selecting the subscription publication route, the journal's standard licensing terms will need to be accepted, including [self-archiving policies](https://www.springernature.com/gp/open-research/policies/journal-policies). Those licensing terms will supersede any other terms that the author or any third party may assert apply to any version of the manuscript.

Your paper will be published online soon after we receive your corrections and will appear in print in the next available issue. Content is published online weekly on Mondays and Thursdays, and the embargo is set at 16:00 London time (GMT)/11:00 am US Eastern time (EST) on the day of publication. Now is the time to inform your Public Relations or Press Office about your paper, as they might be interested in promoting its publication. This will allow them time to prepare an accurate and satisfactory press release. Include your manuscript tracking number (NI-A34258C) and the name of the journal, which they will need when they contact our office.

About one week before your paper is published online, we shall be distributing a press release to news organizations worldwide, which may very well include details of your work. We are happy for your

institution or funding agency to prepare its own press release, but it must mention the embargo date and Nature Immunology. Our Press Office will contact you closer to the time of publication, but if you or your Press Office have any enquiries in the meantime, please contact press@nature.com.

Also, if you have any spectacular or outstanding figures or graphics associated with your manuscript - though not necessarily included with your submission - we'd be delighted to consider them as candidates for our cover. Simply send an electronic version (accompanied by a hard copy) to us with a possible cover caption enclosed.

If you have not already done so, we strongly recommend that you upload the step-by-step protocols used in this manuscript to the Protocol Exchange. Protocol Exchange is an open online resource that allows researchers to share their detailed experimental know-how. All uploaded protocols are made freely available, assigned DOIs for ease of citation and fully searchable through nature.com. Protocols can be linked to any publications in which they are used and will be linked to from your article. You can also establish a dedicated page to collect all your lab Protocols. By uploading your Protocols to Protocol Exchange, you are enabling researchers to more readily reproduce or adapt the methodology you use, as well as increasing the visibility of your protocols and papers. Upload your Protocols at www.nature.com/protocolexchange/. Further information can be found at www.nature.com/protocolexchange/about .

Please note that we encourage the authors to self-archive their manuscript (the accepted version before copy editing) in their institutional repository, and in their funders' archives, six months after publication. Nature Portfolio recognizes the efforts of funding bodies to increase access of the research they fund, and strongly encourages authors to participate in such efforts. For information about our editorial policy, including license agreement and author copyright, please visit www.nature.com/ni/about/ed_policies/index.html

Kind regards,

Laurie

Laurie A. Dempsey, Ph.D.
Senior Editor
Nature Immunology
l.dempsey@us.nature.com
ORCID: 0000-0002-3304-796X